# Adult stem cell-derived complete lung organoid models emulate lung disease in COVID-19

Courtney Tindle[1,2†], MacKenzie Fuller[1,2†], Ayden Fonseca[1,2†], Sahar Taheri[3†], Stella-Rita Ibeawuchi[4], Nathan Beutler[5], Gajanan Dattatray Katkar[1], Amanraj Claire[1,2], Vanessa Castillo[1], Moises Hernandez[6], Hana Russo[4], Jason Duran[7], Laura E Crotty Alexander[8,9], Ann Tipps[4], Grace Lin[4], Patricia A Thistlethwaite[6], Ranajoy Chattopadhyay[1,2,10]*, Thomas F Rogers[5,11,12]*, Debashis Sahoo[3,13]*, Pradipta Ghosh[1,2,14]*†, Soumita Das[2,4]*†

[1]Department of Cellular and Molecular Medicine, University of California San Diego, San Diego, United States; [2]HUMANOID CoRE, University of California San Diego, San Diego, United States; [3]Department of Computer Science and Engineering, Jacobs School of Engineering, University of California San Diego, San Diego, United States; [4]Department of Pathology, University of California San Diego, San Diego, United States; [5]Department of Immunology and Microbiology, The Scripps Research Institute, La Jolla, United States; [6]Division of Cardiothoracic Surgery, University of California San Diego, San Diego, United States; [7]Division of Cardiology, Department of Internal Medicine, UC San Diego Medical Center, San Diego, United States; [8]Pulmonary Critical Care Section, Veterans Affairs (VA) San Diego Healthcare System, La Jolla, United States; [9]Division of Pulmonary and Critical Care, Department of Medicine, University of California, San Diego, La Jolla, CA, United States; [10]Cell Applications Inc., La Jolla, CA, United States; [11]Division of Infectious Diseases, Department of Medicine, University of California, San Diego, La Jolla, United States; [12]Department of Immunology and Microbiology, The Scripps Research Institute, La Jolla, United States; [13]Department of Pediatrics, University of California, San Diego, La Jolla, CA, United States; [14]Department of Medicine, University of California, San Diego, La Jolla, CA, United States

*For correspondence:
rachatto72@gmail.com (RC);
trogers@health.ucsd.edu (TFR);
dsahoo@ucsd.edu (DS);
prghosh@ucsd.edu (PG);
sodas@ucsd.edu (SD)

†These authors contributed equally to this work

Competing interest: The authors declare that no competing interests exist.

## Abstract

**Background:** SARS-CoV-2, the virus responsible for COVID-19, causes widespread damage in the lungs in the setting of an overzealous immune response whose origin remains unclear.
**Methods:** We present a scalable, propagable, personalized, cost-effective adult stem cell-derived human lung organoid model that is complete with both proximal and distal airway epithelia. Monolayers derived from adult lung organoids (ALOs), primary airway cells, or hiPSC-derived alveolar type II (AT2) pneumocytes were infected with SARS-CoV-2 to create in vitro lung models of COVID-19.
**Results:** Infected ALO monolayers best recapitulated the transcriptomic signatures in diverse cohorts of COVID-19 patient-derived respiratory samples. The airway (proximal) cells were critical for sustained viral infection, whereas distal alveolar differentiation (AT2→AT1) was critical for mounting the overzealous host immune response in fatal disease; ALO monolayers with well-mixed proximodistal airway components recapitulated both.
**Conclusions:** Findings validate a human lung model of COVID-19, which can be immediately utilized to investigate COVID-19 pathogenesis and vet new therapies and vaccines.

**Funding:** This work was supported by the National Institutes for Health (NIH) grants 1R01DK107585-01A1, 3R01DK107585-05S1 (to SD); R01-AI141630, CA100768 and CA160911 (to PG) and R01-AI 155696 (to PG, DS and SD); R00-CA151673 and R01-GM138385 (to DS), R01-HL32225 (to PT), UCOP-R00RG2642 (to SD and PG), UCOP-R01RG3780 (to P.G. and D.S) and a pilot award from the Sanford Stem Cell Clinical Center at UC San Diego Health (P.G, S.D, D.S). GDK was supported through The American Association of Immunologists Intersect Fellowship Program for Computational Scientists and Immunologists. L.C.A's salary was supported in part by the VA San Diego Healthcare System. This manuscript includes data generated at the UC San Diego Institute of Genomic Medicine (IGC) using an Illumina NovaSeq 6000 that was purchased with funding from a National Institutes of Health SIG grant (#S10 OD026929).

## Introduction

SARS-CoV-2, the virus responsible for COVID-19, causes widespread inflammation and injury in the lungs, giving rise to diffuse alveolar damage (DAD) (*Andrea Valeria Arrossi and Farver, 2020*; *Damiani et al., 2021*; *Borczuk et al., 2020*; *Li et al., 2021*; *Roden, 2020*), featuring marked infection and viral burden leading to apoptosis of alveolar pneumocytes (*Hussman, 2020*), along with pulmonary edema (*Bratic and Larsson, 2013*; *Carsana et al., 2020*). DAD leads to poor gas exchange and, ultimately, respiratory failure; the latter appears to be the final common mechanism of death in most patients with severe COVID-19 infection. How the virus causes so much damage remains unclear. A particular challenge is to understand the out-of-control immune reaction to the SARS-CoV-2 infection known as a cytokine storm, which has been implicated in many of the deaths from COVID-19. Although rapidly developed preclinical animal models have recapitulated some of the pathognomonic aspects of infection, for example, induction of disease, and transmission, and even viral shedding in the upper and lower respiratory tract, many failed to develop severe clinical symptoms (*Lakdawala and Menachery, 2020*). Thus, the need for preclinical models remains both urgent and unmet.

To address this need, several groups have attempted to develop human preclinical COVID-19 lung models, all within the last few months (*Duan et al., 2020*; *Mulay et al., 2020*; *Salahudeen et al., 2020*). While a head-to-head comparison of the key characteristics of each model can be found in *Table 1*, what is particularly noteworthy is that most of the models do not recapitulate the heterogeneous epithelial cellularity of both proximal and distal airways, that is, airway epithelia, basal cells, secretory club cells, and alveolar pneumocytes. Although induced pluripotent stem cells (iPSC)-derived AT2 cells be differentiated into proximal and distal cell types, including AT1, ciliated, and club cells (*Kawakita et al., 2020*; *Dye et al., 2015*; *Huang et al., 2020*), these iPSC-derived models lack propagability and cannot be reproducibly generated for biobanking; nor can they be scaled up in cost-effective ways for use in drug screens. More specifically, adult lung organoid models that can be grown in a sustainable mode and are complete with proximo-distal epithelia are yet to emerge. Besides the approaches described so far, there are a few more approaches used for modeling COVID-19: (i) 3D organoids from bronchospheres and tracheospheres have been established before (*Hild and Jaffe, 2016*; *Rock et al., 2009*; *Tadokoro et al., 2016*) and are now used in apical-out cultures for infection with SARS-COV-2 (*Suzuki, 2020*); (ii) the most common model used for drug screening is the air-liquid interphase (ALI model) in which pseudo-stratified primary bronchial or small airway epithelial cells are used to recreate the multilayered mucociliary epithelium (*Mou et al., 2016*; *Randell et al., 2011*); (iii) several groups have also generated 3D airway models from iPSCs or tissue-resident stem cells (*Dye et al., 2015*; *Wong et al., 2012*; *Ghaedi et al., 2013*; *Konishi et al., 2016*; *McCauley et al., 2017*; *Miller et al., 2019*); (iv) others have generated AT2 cells from iPSCs using closely overlapping protocols of sequential differentiation starting with definitive endoderm, anterior foregut endoderm, and distal alveolar expression (*Gotoh et al., 2014*; *Jacob et al., 2017*; *Jacob et al., 2019*; *Yamamoto et al., 2017*; *Chen et al., 2017*; *Huang et al., 2014*); and (v) finally, long-term in vitro culture conditions for pseudo-stratified airway epithelium organoids, derived from healthy and diseased adult humans suitable to assess virus infectivity (*Sachs et al., 2019*; *van der Vaart and Clevers, 2021*; *Zhou et al., 2018*), have been pioneered; unfortunately, these airway organoids expressed virtually no lung mesenchyme or alveolar signature. What remains unclear is if any of these models accurately recapitulate the immunopathological phenotype that is seen in the lungs in COVID-19.

**Table 1.** A comparison of current versus existing lung organoid models available for modeling COVID-19.

| Author | Source of stem cells | Propagability | Cell types | | | | | | SARS-COV-2 infection | Demonstrated reproducibility using more than one patient | Cost-effective (use of conditioned media) | Notes |
|---|---|---|---|---|---|---|---|---|---|---|---|---|
| | | | AT1 | AT2 | Club | Basal | Ciliated | Goblet | | | | |
| Zhou et al PMID: 29891677 | Small pieces of normal lung tissue adjacent to the diseased tissue from patients undergoing surgical resection for clinical conditions. | Long term culture > 1 y | | | | | | | Infection with H1N1 pandemic Influenza virus | | | Proximal differentiation (PD) of human Adult Stem Cell-derived airway organoid (AO) culture. Differentiation conditions (PneumaCult-ALI medium) increase ciliated cells. Serine proteases known to be important for productive viral infection, were elevated after PD. |
| Sachs et al PMID: 30643021 | Generation of normal and tumor organoids from resected surplus lung tissue of patients with lung cancers. | long term culture for over 1 year | | | | | | Not clearly mentioned | | | | airway organoid (AO) expressed **no mesenchyme or alveolar** transcripts. Strongly enriched for bulk lung and small airway epithelial signature limited to basal, club, and ciliated cells Withdrawal of R-spondin terminated AO expansion after 3–4 passages similar to the withdrawal of FGFs |
| Duan et al PMID: 32839764 | hPSC derived lung cells and macrophages | | | | | | | Low | SARS-CoV-2 infection mediated damage onset by macrophages. | | | Co-culture of lung cells and macrophages. Protocol followed enables alveolar differentiation process, although described presence of almost all lung cell types. |
| Salahudeen et al PMID: 33238290 | Cells sorted from human peripheral lung tissues. | Distal Lung organoid with possibility of long-term culture | From differentiation of AT2 | | After diff of basal cells | | | | Infection and presence of dsRNA and nucleocapsid | | | No RNA seq of infected samples to compare with COVID Differentiation to different cell types SARS CoV2 infection in apical-out organoids (not polarized monolayers). The combination of EGF and the Noggin was optimal, without any additional growth-promoting effects of either WNT3A or R-spondin |
| Han et al PMID: 33116299 | hPSC-derived lung organoids | Organoids were generated by 50 days of differentiation | | | | | | | SARS-CoV-2 and SARS-CoV-2-Pseudo-Entry Viruses. | | | AT1, AT2, stromal cells, low number of pulmonary neuroendocrine cells, proliferating cells, and airway epithelial cells were reported. Mostly AT2 based ACE2 receptor was used for virus infection. High throughput screen using hPSC-derived lung organoids identified FDA-approved drug candidates, including imatinib and mycophenolic acid, as inhibitors of SARS-CoV-2 entry. |

*Table 1 continued on next page*

*Table 1 continued*

| Author | Source of stem cells | Propagability | Cell types | | | | | | SARS-COV-2 infection | Demonstrated reproducibility using more than one patient | Cost-effective (use of conditioned media) | Notes |
|---|---|---|---|---|---|---|---|---|---|---|---|---|
| | | | AT1 | AT2 | Club | Basal | Ciliated | Goblet | | | | |
| Youk et al PMID: 33142113 | Adult alveolar stem cells isolated from distal lung parenchymal tissues by collagenase, dispase and sorting | Multiple passages upto 10 months | From AT2; Lost in higher passages | | | | | | In the organoid form | | | Single cell transcriptomic profiling identified two clusters and type I interferon signal pathway are highly elevated at three dpi |
| | Alv organoids with distal lung epithelial cells with lung fibroblast cells | | | | | | | | | | | Infection of AT2 cells trigger apoptosis that may contribute to alveolar injury. Alteration of innate immune response genes from AT2 cells |
| Mulay et al PMID: 32637946 doi: org/10.1101/2020.06.29.174623 | Proximal airway ALI with heterogenous cells | | | | | | | | In the organoid form | | | Infection of ciliated and goblet cells Two separate models for SARS-CoV2 infection |
| Huang J PMID:32979316 | iPSC derived AT2 cell ALI model | | | | | | | | | | | Bulk RNA seq after day 1 and day four infection. The infection induces rapid inflammatory responses. |
| | iPSC derived basal cells as oranoids or 2D ALI | | | | | | | | | | | iPSCs transcripts match human lung better than cancer cell lines. iPSC AT2 cells express host genes mportant for SARS-CoV-2 infection. |
| Abo et al PMID: 32577635 doi: 10.1101/2020.06.03.132639 | iPSC AT2 cells as organoids or 2D ALI | | | | | | | | | | | |
| Rock et al PMID: 19625615 | Bronchospheres were isolated from human lung tissue. | | | | | | | | | | | Bronchospheres derived from human lung can act as stem cells and can differentiate into other cell types. |
| Lamers et al PMID: 33283287 | Lung organoids derived from fetal Lung epithelial bud tips and differentiated ALI model. | 14 passages | | | Detected SCGB3A2(ATII/ club marker) | | | | | 2 subjects were mentioned | | Organoid model derived from fetal lung bud tip tissue consists primarily of SOX2+ SOX9+ progenitor cells. Differentiation under ALI conditions is necessary to achieve mature alveolar epithelium. ALI model was found to contain mostly ATII and ATI cells, with small basal and rare neuroendocrine populations. SFTPC + Alveolar type II like cells were most readily infected by SARS-CoV-2. The infectious virus titer is much higher (five log) compared to other established model. |

*Table 1 continued on next page*

*Table 1 continued*

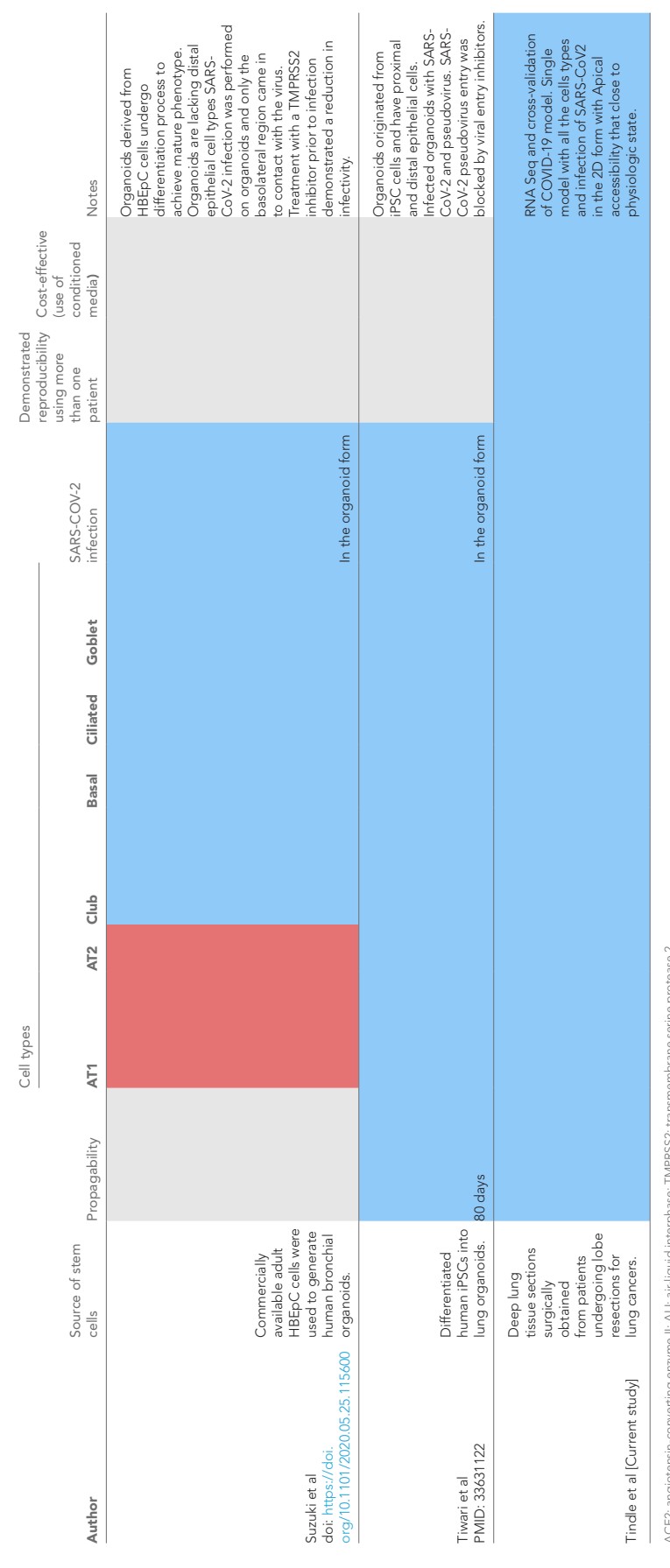

| Author | Source of stem cells | Propagability | Cell types | | | | | | SARS-COV-2 infection | Demonstrated reproducibility using more than one patient | Cost-effective (use of conditioned media) | Notes |
|---|---|---|---|---|---|---|---|---|---|---|---|---|
| | | | AT1 | AT2 | Club | Basal | Ciliated | Goblet | | | | |
| Suzuki et al doi: https://doi.org/10.1101/2020.05.25.115600 | Commercially available adult HBEpC cells were used to generate human bronchial organoids. | | | | | | | | In the organoid form | | | Organoids derived from HBEpC cells undergo differentiation process to achieve mature phenotype. Organoids are lacking distal epithelial cell types SARS-CoV-2 infection was performed on organoids and only the basolateral region came in to contact with the virus. Treatment with a TMPRSS2 inhibitor prior to infection demonstrated a reduction in infectivity. |
| Tiwari et al PMID: 33631122 | Differentiated human iPSCs into lung organoids. | 80 days | | | | | | | In the organoid form | | | Organoids originated from iPSC cells and have proximal and distal epithelial cells. Infected organoids with SARS-CoV-2 and pseudovirus. SARS-CoV-2 pseudovirus entry was blocked by viral entry inhibitors. |
| Tindle et al [Current study] | Deep lung tissue sections surgically obtained from patients undergoing lobe resections for lung cancers. | | | | | | | | | | | RNA Seq and cross-validation of COVID-19 model. Single model with all the cells types and infection of SARS-CoV2 in the 2D form with Apical accessibility that close to physiologic state. |

ACE2: angiotensin-converting enzyme II; ALI: air-liquid interphase; TMPRSS2: transmembrane serine protease 2.

Blue color cells denote the presence of the features.

Red color cells denote the absence of the features.

Grey color cells denote information not found.

**Table 2.** Markers used to identify various cell types in the lung.

| Cell type | Markers |
| --- | --- |
| AT1 | AQP5*$, PDPN*$$, Carboxypeptidase M, CAV-1, CAV-2, HTI56, HOPX, P2$R$ × 4*$$, Na+/K + ATPase$, TIMP3*++, SEMA3F PDPN* AQP5* P2$R$ × 4* TIMP3* SERPINE* |
| AT2 | ABCA3*$$, CC10 (SCGB1A1*)+, CD44v6, Cx32, gp600++, ICAM-1++, KL-6, LAMP3*$$, MUC1, SFTPA1*$$, SFTPB*$, SFTPC*+, SFTPD*, SERPINE1 |
| Club | CC10 (SCGB1A1*)+, CYP2F2*, ITAG6*$$, SCGB3A2*$$, SFTPA1*$$, SFTPB*$, SFTPD* |
| Goblet | CDX-2*, MUC5AC*, MUC5B*, TFF3*, UEA1+ |
| Ciliated | ACT (ACTG2*)$, BTub4 (TUBB4A*), FOXA3*++, FOXJ1*, SNTN* |
| Basal | CD44v6 (CD44*), ITGA6*$$, KRT5*$, KRT13*, KRT14*, p63 (CKAP4*), p75 (NGFR*)$$ |
| Generic Lung Lineage | Cx43 (GJA1*), TTF-1 (TTF1*; Greatest in AT2 & Club), EpCAM (EPCAM*) |

*Markers used for single-cell gating (Figure 1A).
$ denotes markers used in this work for Immunofluorescence (IF).
$$ denotes markers used in this work for qPCR.
+ denotes markers used in both IF and qPCR.
++ denotes obscure markers (Not a lot of research relative to lung).

We present a rigorous transdisciplinary approach that systematically assesses an adult lung organoid model that is propagable, personalized, and complete with both proximal airway and distal alveolar cell types against existing models that are incomplete, and we cross-validate them all against COVID-19 patient-derived respiratory samples. Findings surprisingly show that cellular crosstalk between both proximal and distal components is necessary to emulate how SARS-CoV-2 causes diffuse alveolar pneumocyte damage; the proximal airway mounts a sustained viral infection, but it is the distal alveolar pneumocytes that mount the overzealous host response that has been implicated in a fatal disease.

## Results

### A rationalized approach for creating and validating acute lung injury in COVID-19

To determine which cell types in the lungs might be most readily infected, we began by analyzing a human lung single-cell sequencing dataset (GSE132914) for the levels of expression of angiotensin-converting enzyme II (ACE2) and transmembrane serine protease 2 (TMPRSS2), the two receptors that have been shown to be the primary sites of entry for the SARS-CoV-2 (*Hoffmann et al., 2020*). The dataset was queried with widely accepted markers of all the major cell types (see *Table 2*). Alveolar epithelial type 2 (AT2), ciliated and club cells emerged as the cells with the highest expression of both receptors (*Figure 1A*, *Figure 1—figure supplement 1A*). These observations are consistent with published studies demonstrating that ACE2 is indeed expressed highest in AT2 and ciliated cells (*Mulay et al., 2020*; *Zhao et al., 2020*; *Jia et al., 2005*). In a cohort of deceased COVID-19 patients, we observed by H&E (*Figure 1—figure supplement 1B*) that gas-exchanging flattened AT1 pneumocytes are virtually replaced by cuboidal cells that were subsequently confirmed to be AT2-like cells via immunofluorescent staining with the AT2-specific marker, surfactant protein C (SFTPC; *Figure 1B*, upper panel, *Figure 1—figure supplement 1C*, top). We also confirmed that club cells express ACE2 (*Figure 1—figure supplement 1C*, bottom), underscoring the importance of preserving these cells in any ideal lung model of COVID-19. When we analyzed the lungs of deceased COVID-19 patients, the presence of SARS-COV-2 in alveolar pneumocytes was also confirmed, as determined by the colocalization of viral nucleocapsid protein with SFTPC (*Figure 1B*, lower panel, *Figure 1—figure supplement 1D*). Immunohistochemistry studies further showed the presence of SARS-COV-2 virus in alveolar pneumocytes and in alveolar immune cells (*Figure 1—figure supplement 1E*). These findings are consistent with the gathering consensus that alveolar pneumocytes support the interaction between the epithelial cells and inflammatory cells recruited to the lung; via mechanisms that remain unclear, they are generally believed to contribute to the development of acute lung injury and acute respiratory distress syndrome (ARDS), the severe hypoxemic respiratory failure during COVID-19 (*Hou et al., 2020*; *Spagnolo et al., 2020*). Because prior work has demonstrated that SARS-CoV-2 infectivity in patient-derived airway cells is highest in the proximal airway epithelium compared to the distal alveolar pneumocytes (AT1 and AT2) (*Hou et al., 2020*), and yet, it is the AT2 pneumocytes that harbor the virus, and the AT1 pneumocytes that are ultimately destroyed during DAD, we hypothesized that both proximal airway and distal (alveolar pneumocyte) components might

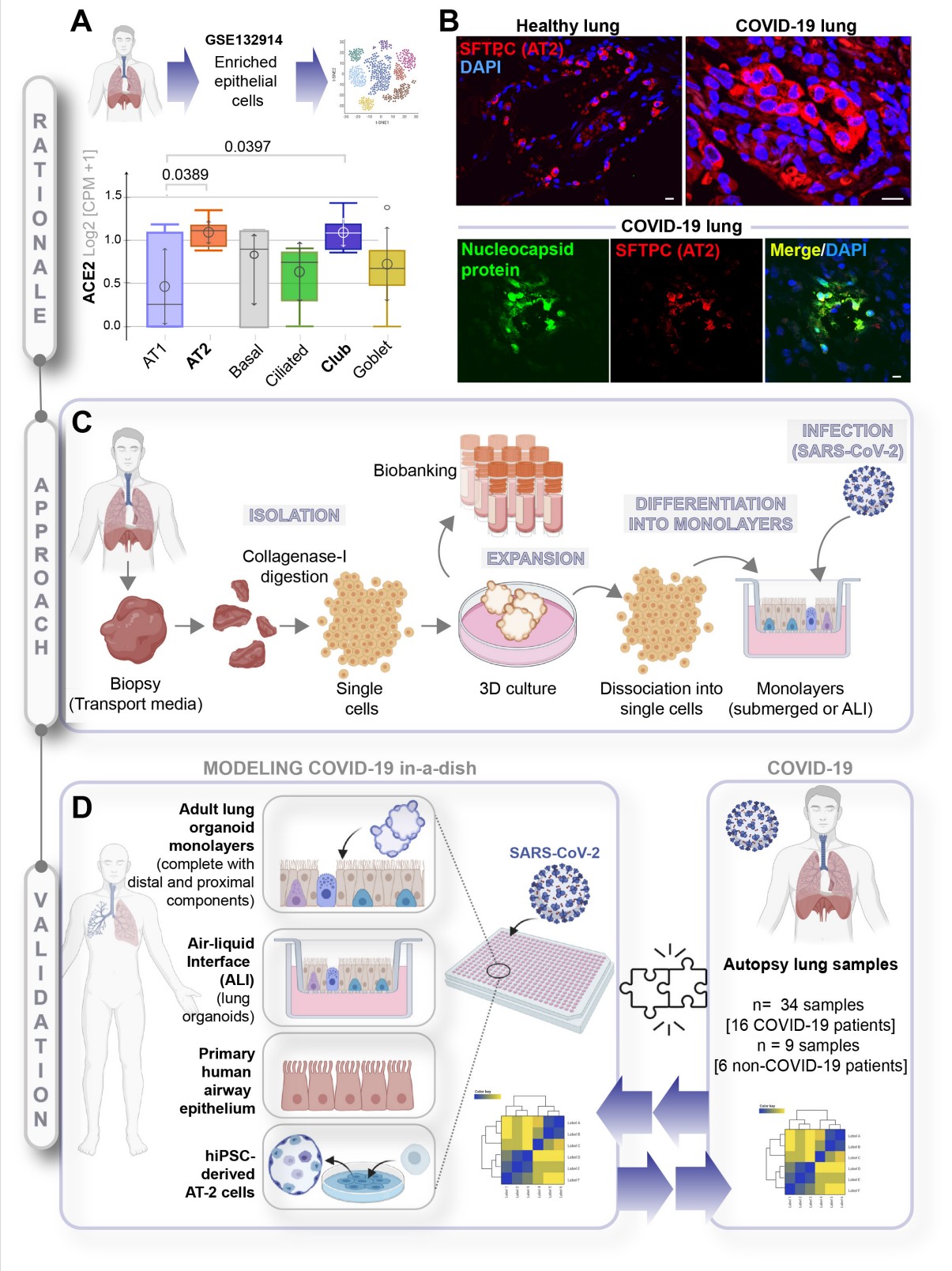

**Figure 1.** A rationalized approach to building and validating human preclinical models of COVID-19. **A**) Whisker plots display relative levels of angiotensin-converting enzyme II (ACE2) expression in various cell types in the normal human lung. The cell types were annotated within a publicly available single-cell sequencing dataset (GSE132914) using genes listed in Table 1. p-values were analyzed by one-way ANOVA and Tukey's post hoc test. (**B**) Formalin-fixed paraffin-embedded sections of the human lung from normal and deceased COVID-19 patients were stained for SFTPC, alone

*Figure 1 continued on next page*

*Figure 1 continued*
or in combination with nucleocapsid protein and analyzed by confocal immunofluorescence. Representative images are shown. Scale bar = 20 μm. (**C**) Schematic showing key steps generating an adult stem cell-derived, propagable, lung organoid model, complete with proximal and distal airway components for modeling COVID-19-in-a-dish. See Materials and methods for details regarding culture conditions. (**D**) A transcriptome-based approach is used for cross-validation of in vitro lung models of SARS-CoV-2 infection (left) versus the human disease, COVID-19 (right), looking for a match in gene expression signatures.

The online version of this article includes the following figure supplement(s) for figure 1:

**Figure supplement 1.** Alveolar type II pneumocyte hyperplasia is a pathognomonic feature of lung injury in COVID-19.

play distinct roles in the respiratory system to mount the so-called viral infectivity and host immune response phases of the clinical symptoms observed in COVID-19 (*Chen and Li, 2020*).

Because no existing lung model provides such proximodistal cellular representation (*Table 1*), and hence, may not recapitulate with accuracy the clinical phases of COVID-19, we first sought to develop a lung model that is complete with both proximal and distal airway epithelia using adult stem cells that were isolated from deep lung biopsies (i.e., sufficient to reach the bronchial tree). Lung organoids were generated using the work flow outlined in *Figure 1C* and a detailed protocol that had key modifications from previously published (*Sachs et al., 2019*; *Zhou et al., 2018*) methodologies (see Materials and methods). Organoids grown in 3D cultures were subsequently dissociated into single cells to create 2D monolayers (either maintained submerged in media or used in ALI model) for SARS-CoV-2 infection, followed by RNA seq analysis. Primary airway epithelial cells and hiPSC-derived alveolar type II (AT2) pneumocytes were used as additional models (*Figure 1D*, left panel). Each of these transcriptomic datasets was subsequently used to cross-validate our ex vivo lung models of SARS-CoV-2 infection with the human COVID-19 autopsy lung specimens (*Figure 1D*, right panel) to objectively vet each model for their ability to accurately recapitulate the gene expression signatures in the patient-derived lungs.

## Creation of a lung organoid model, complete with both proximal and distal airway epithelia

Three lung organoid lines were developed from deep lung biopsies obtained from the normal regions of lung lobes surgically resected for lung cancer; both genders, smokers and non-smokers, were represented (*Figure 2—figure supplement 1A*; *Table 3*). Three different types of media were compared (*Figure 2—figure supplement 1B*); the composition of these media was inspired either by their ability to support adult-stem cell-derived mixed epithelial cellularity in other organs (like the gastrointestinal [GI] tract [*Miyoshi and Stappenbeck, 2013*; *Sato et al., 2009*; *Sayed et al., 2020c*]) or rationalized based on published growth conditions for proximal and distal airway components (*Gotoh et al., 2014*; *Sachs et al., 2019*; *van der Vaart and Clevers, 2021*). A growth condition that included conditioned media from L-WRN cells that express Wnt3, R-spondin, and Noggin, supplemented with recombinant growth factors, which we named as '*lung organoid expansion media*,' emerged as superior compared to alveolosphere media-I and II (*Jacob et al., 2019*; *Yamamoto et al., 2017*) (details in Materials and methods), based on its ability to consistently and reproducibly support the best morphology and growth characteristics across multiple attempts to isolate organoids from lung tissue samples. Three adult lung organoid lines (ALO1-3) were developed using the expansion media, monitored for their growth characteristics by brightfield microscopy and cultured with similar phenotypes until P10 and

**Table 3.** Characteristics of patients enrolled into this study for obtaining lung tissues to serve as source of stem cells to generate lung organoids.

| Name | Date of surgery | Age | Sex | Smoking history | Reason for surgery | Histology |
|---|---|---|---|---|---|---|
| ALO1 | 4/17/2020 | 64 | Male | Current, chronic smoker Packs/day: 0.50 Years: 53 Pack years: 26.5 | Lung carcinoma | Invasive squamous cell carcinoma, non-keratinizing |
| ALO2 | 4/17/2020 | 59 | Male | Non-smoker | Lung carcinoma | Invasive adenocarcinoma |
| ALO3 | 7/7/2020 | 46 | Female | Non-smoker | Left lower lobe nodule | Invasive adenocarcinoma |

beyond (*Figure 2—figure supplement 1C* and D). The 3D morphology of the lung organoid was also assessed by H&E staining of slices cut from formalin-fixed paraffin-embedded (FFPE) cell blocks of *HistoGel*-emb`edded ALO1-3 (*Figure 2—figure supplement 1E*).

To determine if all the six major lung epithelial cells (illustrated in *Figure 2A*) are present in the organoids, we analyzed various cell-type markers by qRT-PCR (*Figure 2B–H* and *Figure 2—figure supplement 2A-H*). All three ALO lines had a comparable level of AT2 cell surfactant markers (compared against hiPSC-derived AT2 cells as positive control) and a significant amount of AT1, as determined using the marker AQP5. ALOs also contained basal cells (as determined by the marker ITGA6, p75/NGFR, TP63), ciliated cells (as determined by the marker FOXJ1), and club cells (as determined by the marker SCGB1A1). As expected, the primary normal human bronchial epithelial cells (NHBE) had significantly higher expression of basal cell markers than the ALO lines (hence, served as a positive control), but they lacked stemness and club cells (hence, served as a negative control).

The presence of all cell types was also confirmed by assessing protein expression of various cell types within organoids grown in 3D cultures. Two different approaches were used—(i) slices cut from FFPE cell blocks of *HistoGel*-embedded ALO lines (*Figure 2I and J*) or (ii) ALO lines grown in 8-well chamber slides were fixed in Matrigel (*Figure 2K*), stained, and assessed by confocal microscopy. Such staining not only confirmed the presence of more than one cell type (i.e., mixed cellularity) of proximal (basal-KRT5) and distal (AT1/AT2 markers) within the same ALO line, but also, in some instances, demonstrated the presence of mixed cellularity within the same 3D structure. For example, AT2 and basal cells, marked by SFTPB and KRT5, respectively, were found in the same 3D structure (*Figure 2J*, interrupted curved lines). Similarly, ciliated cells and goblet cells stained by Ac-Tub and Muc5AC, respectively, were found to coexist within the same structure (*Figure 2J*, interrupted box; *Figure 2K*, arrow). Intriguingly, we also detected 3D structures that co-stained for CC10 and SFTPC (*Figure 2J*, bottom panel) indicative of mixed populations of club and AT2 cells. Besides the organoids with heterogeneous makeup, each ALO line also showed homotypic organoid structures that were relatively enriched in one cell type (*Figure 2J*, arrowheads pointing to two adjacent structures that are either KRT5- or SFTPB-positive). Regardless of their homotypic or heterotypic cellular organization into 3D structures, the presence of mixed cellularity was documented in all three ALO lines (see multiple additional examples in *Figure 2—figure supplement 2I*). It is noteworthy that the coexistence of proximal and distal epithelial cells in lung organoids has been achieved in one another instance prior; Lamers et al. showed such mixed cellular composition in fetal lung bud tip-derived organoids *Lamers et al., 2021*. However, their model lacked ciliated and goblet cells (*Lamers et al., 2021*), something that we could readily detect in our 3D organoids.

Finally, using qRT-PCR of various cell-type markers as a measure, we confirmed that the ALO models overall recapitulated the cell-type composition in the adult lung tissues from which they were derived (*Figure 2—figure supplement 3*) and retained such composition in later passages without significant notable changes in any particular cell type (*Figure 2—figure supplement 4*). The mixed proximal and distal cellular composition of the ALO models and their degree of stability during in vitro culture was also confirmed by flow cytometry (*Figure 2—figure supplement 5*).

## Organoid cellularity resembles tissue sources in 3D cultures but differentiates in 2D cultures

To model respiratory infections such as COVID-19, it is necessary for pathogens to be able to access the apical surface. It is possible to microinject into the lumens of 3D organoids, as done previously with pathogens in the case of gut organoids (*Engevik et al., 2015*; *Forbester et al., 2015*; *Leslie et al., 2015*; *Williamson et al., 2018*), or FITC-dextran in the case of lung organoids (*Porotto et al., 2019*), or carry out infection in apical-out 3D lung organoids with basal cells (*Salahudeen et al., 2020*). However, the majority of the researchers have gained apical access by dissociating 3D organoids into single cells and plating them as 2D-monolayers (*Duan et al., 2020*; *Mulay et al., 2020*; *Huang et al., 2020*; *Sachs et al., 2019*; *Zhou et al., 2018*; *Han et al., 2020a*; *Han et al., 2020b*. As in any epithelium, the differentiation of airway epithelial cells relies upon dimensionality (apicobasal polarity); because the loss of dimensionality can have a major impact on cellular proportions and impact disease-modeling in unpredictable ways, we assessed the impact of the 3D-to-2D conversion on cellularity by RNA seq analyses. Two commonly encountered methods of growth in 2D monolayers were tested: (i) monolayers polarized on trans-well inserts but submerged in growth media (*Figure 3A*

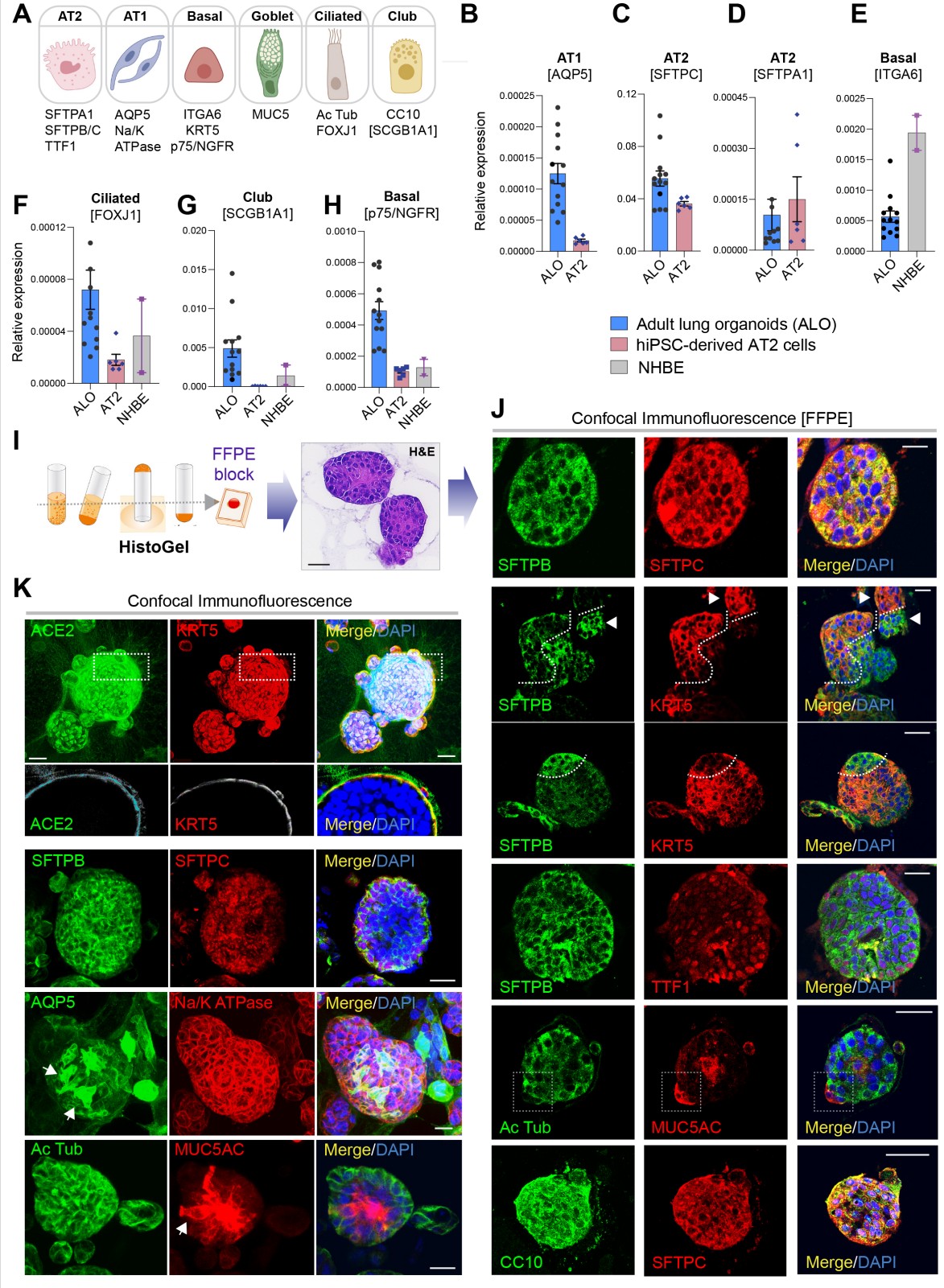

**Figure 2.** Adult stem cell-derived lung organoids are propagatable models with both proximal and distal airway components. (**A**) Schematic lists the various markers used here for qPCR and immunofluorescence to confirm the presence of all cell types in the 3D lung organoids here and in 2D monolayers later (in **Figure 3**). (**B–H**) Bar graphs display the relative abundance of various cell-type markers (normalized to 18S) in adult lung organoids (ALO), compared to the airway ( normal human bronchial epithelial cell [NHBE]) and/or alveolar (AT2) control cells, as appropriate. p-values were

*Figure 2 continued on next page*

*Figure 2 continued*

analyzed by one-way ANOVA. Error bars denote SEM; n = 3–6 datasets from three independent ALOs and representing early and late passages. See also *Figure 2—figure supplement 2* for individual ALOs. (**I, J**). H&E-stained cell blocks were prepared using *HistoGel* (**I**). Slides were stained for the indicated markers and visualized by confocal immunofluorescence microscopy. Representative images are shown in (**J**). Scale bar = 50 μm. (**K**) 3D organoids grown in 8-well chamber slides were fixed, immunostained, and visualized by confocal microscopy as in (**J**). Scale bar = 50 μm. See also *Figure 2—figure supplement 2*. Top row (ACE2/KRT5-stained organoids) displays the single and merged panels as max projections of z-stacks (top) and a single optical section (bottom) of a selected area. For the remaining rows, the single (red/green) channel images are max projections of z-stacks; however, merged panels are optical sections to visualize the centers of the organoids. All immunofluorescence images showcased in this figure were obtained from ALO lines within passage #3–6. See also *Figure 2—figure supplements 3–5* for additional evidence of mixed cellularity of ALO models, their similarity to lung tissue of origin, and stability of cellular composition during early (#1–8) and late (#8–15) passages, as determined by qPCR and flow cytometry.

The online version of this article includes the following figure supplement(s) for figure 2:

**Figure supplement 1.** Lung organoids are reproducibly established from three different donors and propagated in each case over 10 passages.

**Figure supplement 2.** Adult stem cell-derived lung organoids are propagatable models with both proximal and distal airway components.

**Figure supplement 3.** Adult stem cell-derived lung organoids (ALO) generally recapitulate cell-type-specific gene expression patterns observed in the adult lung tissue (ALT) from which they originate.

**Figure supplement 4.** Adult stem cell-derived lung organoids (ALO) generally maintain their cellular composition from early (E) to late (L) passages, as determined by cell-type-specific gene expression by qPCR.

**Figure supplement 5.** Adult stem cell-derived lung organoids (ALO) comprised both proximal and distal airway epithelial population and generally maintain such diversity from early (E) to late (L) passages, as determined by FACS.

and *Figure 3—figure supplement 1A-D*) and (ii) monolayers were grown at the air-liquid interface (popularly known as the 'ALI model'; *Prytherch et al., 2011*; *Dvorak et al., 2011*) for 21 days to differentiate into the mucociliary epithelium (*Figure 3A* and *Figure 3—figure supplement 1E-G*). The submerged 2D monolayers had several regions of organized vacuolated-appearing spots (*Figure 3—figure supplement 1D*, arrow), presumably due to morphogenesis and cellular organization even in 2D. Consistent with this morphological appearance, the epithelial barrier formed in the submerged condition was leakier, as determined by relatively lower transepithelial electrical resistance (TEER; *Figure 3—figure supplement 1B*) and the flux of FITC-dextran from apical to basolateral chambers (*Figure 3—figure supplement 1C*), and corroborated by morphological assessment by confocal immunofluorescence of localization of occludin, a bona fide TJ marker. We chose occludin because it is a shared and constant marker throughout the airway that stabilizes claudins and regulates their turnover *McGowan, 2014* and plays an important role in maintaining the integrity of the lung epithelial barrier *Liu et al., 2014*. Junction-localized occludin was patchy in the monolayer, despite the fact that the monolayer was otherwise intact, as determined by phalloidin staining (*Figure 3—figure supplement 1H* and I). Our finding that ALO 3D organoids differentiating into monolayers in submerged cultures (where alveolar differentiation and cell flattening happens dynamically as progenitor cells give rise to AT1/2 cells) are leaky is in keeping with prior work demonstrating that the TJs are rapidly remodeled as alveolar cells mature *Schlingmann et al., 2015*; *Yang et al., 2016*. By contrast, and as expected *Rayner et al., 2019*, the ALI monolayers formed a more effective epithelial barrier, as determined by TEER (*Figure 3—figure supplement 1F*), and appeared to be progressively hazier with time after air-lift, likely due to the accumulation of secreted mucin (*Figure 3—figure supplement 1G*).

RNA seq datasets were analyzed using the same set of cell markers, as we used in *Figure 1A* (listed in *Table 2*). Consistent with our morphological, gene expression, and FACS-based studies showcased earlier (*Figure 2* and *Figure 2—figure supplements 2–5*), cell-type deconvolution of our transcriptomic dataset using CIBERSORTx (https://cibersortx.stanford.edu/runcibersortx.php) confirmed that cellular composition in the human lung tissues was reflected in the 3D ALO models and that such composition was also relatively well-preserved over several passages (*Figure 3B*, left); both showed a mixed population of simulated alveolar, basal, club, ciliated, and goblet cells. When 3D organoids were dissociated and plated as 2D monolayers on transwells, the AT2 signatures were virtually abolished with a concomitant and prominent emergence of AT1 signatures, suggesting that growth in 2D monolayers favors differentiation of AT2 cells into AT1 cells *Shami and Evans, 2015* (*Figure 3B*, middle). A compensatory reduction in proportion was also observed for the club, goblet, and ciliated cells. The same organoids, when grown in long-term 2D culture conditions in the ALI model, showed a strikingly opposite pattern; alveolar signatures were almost entirely replaced by a concomitant increase

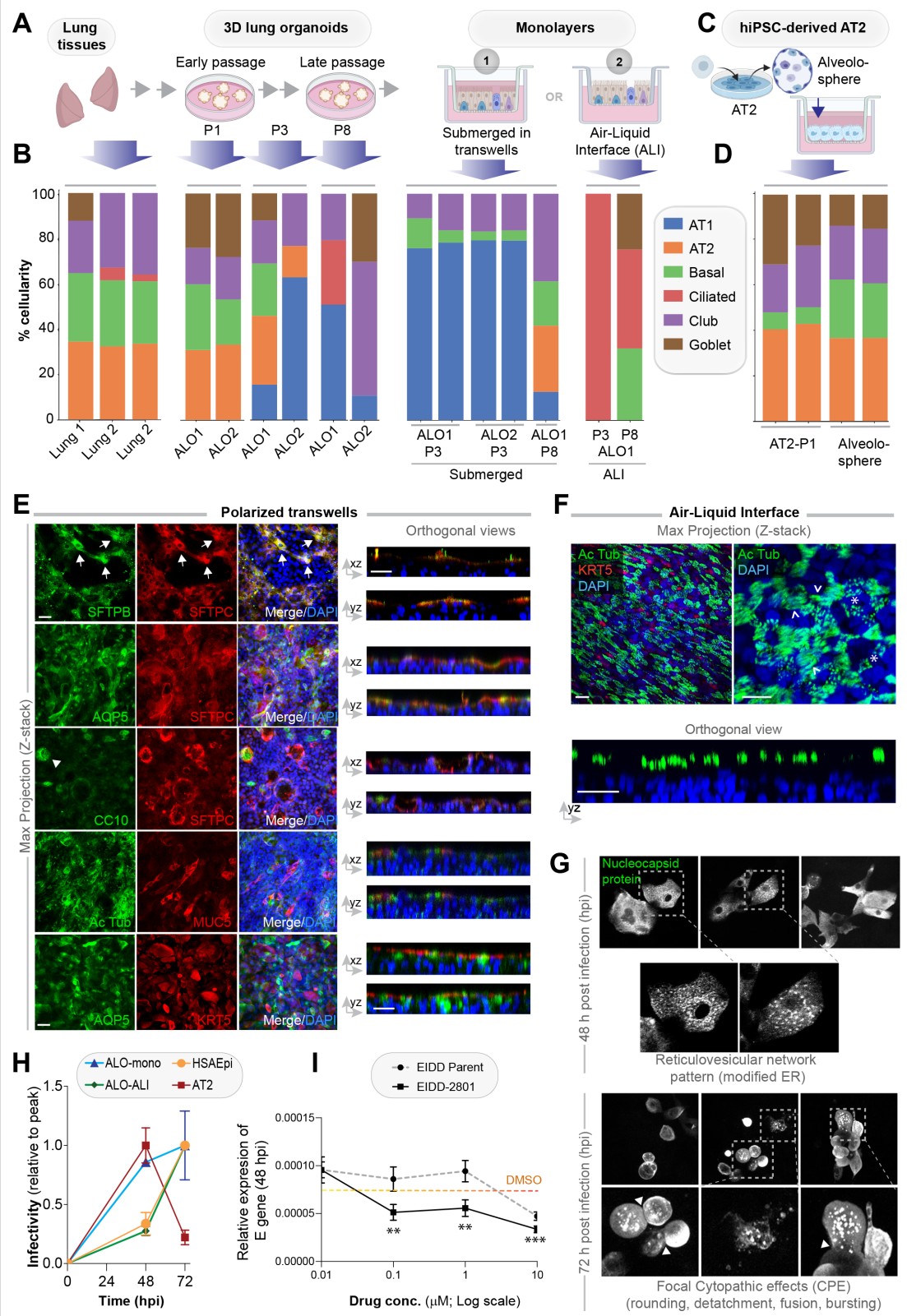

**Figure 3.** Monolayers derived from lung organoids differentiate into proximal and distal airway components. (**A, B**) Samples collected at various steps of lung organoid isolation and expansion in culture, and from the two types of monolayers prepared using the lung organoids were analyzed by bulk RNA seq and the datasets were compared for % cellular composition using the deconvolution method, CYBERSORTx. Schematic in (**A**) shows the workflow steps, and bar plots in (**B**) show the relative proportion of various lung cell types. (**C, D**) hiPSC-derived AT2 cells and alveolospheres (**C**) were

*Figure 3 continued on next page*

*Figure 3 continued*

plated as monolayers and analyzed by RNA seq. Bar plots in (**D**) show % cellular composition. (**E, F**) Submerged adult lung organoids (ALO) monolayers in transwells (**E**) or monolayers were grown as air-liquid interphase (ALI) models (**F**) were fixed and stained for the indicated markers and visualized by confocal immunofluorescence microscopy. The representative max projected z-stack images (left) and the corresponding orthogonal images (right) are displayed. Arrows in (**E**) indicate AT2 cells; arrowheads in (**E**) indicate club cells; asterisk in (**F**) indicates bundles of cilia standing perpendicular to the plane of the ALI monolayers; arrowheads in (**F**) indicate bundles of cilia running parallel to the plane of the ALI monolayers. Scale bar = 20 µm. (**G**) Monolayers of ALO1-3 were challenged with SARS-CoV-2 for indicated time points prior to fixation and staining for KRT5, SARS-COV2 viral nucleocapsid protein and DAPI and visualized by confocal microscopy. A montage of representative images are shown, displaying reticulovesicular network patterns and various cytopathic effects. Scale bar = 15 µm. (**H**) Monolayers of ALO, hiPSC-derived AT2 cells, and other alternative models (see *Figure 3—figure supplements 1–2*) were infected or not with SARS-CoV-2 and analyzed for infectivity by qPCR (targeted amplification of viral envelope, E gene). See also *Figure 3—figure supplement 3B, C* for comparison of the degree of peak viral amplification across various models. (**I**) ALO monolayers pretreated for 4 hr with either vehicle (DMSO) control or EIDD-parent (NHC) or its metabolite EIDD-2801/MK-4482 were infected with SARS-CoV-2 and assessed at 48 hpi for infectivity as in (**H**). Line graphs display the relative expression of E gene. Error bars display SEM. p value **<0.01; ***<0.001.

The online version of this article includes the following figure supplement(s) for figure 3:

**Figure supplement 1.** Monolayers derived from adult lung organoids (ALO) can form an epithelial barrier.

**Figure supplement 2.** Alternative models of lung epithelial cells used in this work for modeling SARS-CoV-2 infection and/or as a control for gene expression studies.

**Figure supplement 3.** Proof of SARS-CoV-2 infectivity.

in ciliated and goblet cells (*Figure 3B*, right). These findings are consistent with the well-established notion that ALI conditions favor growth as pseudo-stratified mucociliary epithelium *Prytherch et al., 2011*; *Dvorak et al., 2011*. As an alternative model for use as monolayers for viral infection, we developed hiPSC-derived AT2 cells and alveolospheres (*Figure 3C*), using established protocols *Huang et al., 2020*. Because they were grown in the presence of CHIR99021 (an aminopyrimidine derivate that is a selective and potent Wnt agonist) *Jacob et al., 2019*; *Yamamoto et al., 2017*; *Abdelwahab et al., 2019*, which probably inhibits the AT2→AT1 differentiation, these monolayers were enriched for AT2 and devoid of AT1 cells (*Figure 3D*).

The multicellularity of lung organoid monolayers was also confirmed by immunofluorescence staining and confocal microscopy of the submerged and ALI monolayers, followed by the visualization of cell markers in either max-projected z-stacks (*Figure 3E*, left) or orthogonal views of the same (*Figure 3E*, right). As expected, markers for the same cell type (i.e., SFTPB and SFTPC, both AT2 markers) colocalize, but markers for different cell types do not. Submerged monolayers showed the prominent presence of both AT1 (AQP5-positive) and AT2 cells. Compared to the 3D organoids, the 2D organoid cultures, especially the ALI model, showed a significant increase in ciliated structures, as determined by acetylated tubulin (compare Ac Tub-stained panels in *Figure 2J and K* with *Figure 3E and F*). The observed progressive prominence of ciliary structures from 3D to 2D models is in keeping with the fact that 3D ALOs that are yet to form lumen represent the least differentiated state, whereas 2D submerged monolayers are intermediate and the 2D ALI monolayers are maximally differentiated; differentiation is known to establish apicobasal polarity, which is essential for the emergence of mature cilia on the apical surface. This increase in ciliated epithelium was associated with a concomitant decrease in KRT5-stained basal cells (*Figure 3F*). Such loss of the basal cell marker KRT5 between submerged monolayers and the ALI model can be attributed to and the expected conversion of basal cells to other cell types (i.e., ciliated cells) *Gras et al., 2017*; *Khelloufi et al., 2018*. The presence of AT2 cells, scattered amidst the ciliated cells in these ALI monolayers, was confirmed by co-staining them for SFTPC and Ac-Tub (*Figure 3—figure supplement 1J*).

Finally, we sought to confirm that the epithelial barrier that is formed by the submerged monolayers derived from ALO is responsive to infections. To this end, we simulated infection by challenging ALO monolayers with LPS. Compared to unchallenged controls, the integrity of the barrier was impaired by LPS, as indicated by a significant drop in the TEER (*Figure 3—figure supplement 1K* and L), which is in keeping with the known disruptive role of LPS on the respiratory epithelium *Kalsi et al., 2020*.

Taken together, the immunofluorescence images are in agreement with the RNA seq dataset; both demonstrate that the short-term submerged monolayer favors distal differentiation (AT2→AT1), whereas the 21-day ALI model favors proximal mucociliary differentiation. It is noteworthy that these distinct differentiation phenotypes originated from the same 3D organoids despite the seeding of cells in the same basic media composition (i.e., PneumaCult) prior to switching over to an ALI maintenance

media for the prolonged growth at ALI; the latter is a well-described methodology that promotes differentiation into ciliated and goblet cells *Rayner et al., 2019*.

## Differentiated 2D monolayers show that SARS-CoV-2 infectivity is higher in proximal than distal epithelia

Because the lung organoids with complete proximodistal cellularity could be differentiated into either distal-predominant monolayers in submerged short-term cultures or proximal-predominant monolayers in long-term ALI cultures, this provided us with an opportunity to model the respiratory tract and assess the impact of the virus along the entire proximal-to-distal gradient. We first asked if ALO monolayers are permissive to SARS-CoV-2 entry and replication and support sustained viral infection. Confocal imaging of infected ALO monolayers with anti-SARS-COV-2 nucleocapsid protein antibody showed that submerged ALO monolayers did indeed show progressive changes during the 48–72 hr window after infection (*Figure 3G*): by 48 hpi, we observed the formation of 'reticulovesicular patterns' that are indicative of viral replication within modified host endoplasmic reticulum (*Knoops et al., 2008*; *Figure 3G*, left), and by 72 hpi we observed focal cytopathic effect (CPE) (*Kaye, 2006*) such as cell-rounding, detachment, and bursting of virions (*Figure 3G*, right, *Figure 3—figure supplement 3A*).

We next asked how viral infectivity varies in the various lung models. Because multiple groups have shown the importance of the ciliated airway cells for infectivity (i.e., viral entry, replication, and apical release [*Hou et al., 2020*; *Milewska et al., 2020*; *Zhu et al., 2020*; *Hui et al., 2020*]), as positive controls, we infected monolayers of human airway epithelia (see the legend, *Figure 3—figure supplement 2A-D*. AT2 cells, which express high levels of viral entry receptors ACE2 and TMPRSS2 (*Figure 1A*, *Figure 1—figure supplement 1A*), have been shown to be proficient in the viral entry but are least amenable to sustained viral release and infectivity (*Hou et al., 2020*; *Hui et al., 2020*). To this end, we infected monolayers of hiPSC-derived homogeneous cultures of AT2 cells as secondary controls (see the legend, *Figure 3—figure supplement 2E-G*). Infection was carried out using the Washington strain of SARS-CoV-2, USA-WA1/2020 (BEI Resources NR-52281 *Rogers et al., 2020*). As expected, the 2D lung monolayers we generated, both the submerged and the ALI models, were readily infected with SARS-CoV-2 (*Figure 3—figure supplement 3B*), as determined by the presence of the viral envelope gene (E gene; *Figure 3H*); however, the kinetics of viral amplification differed. When expressed as levels of E gene normalized to the peak values in each model (*Figure 3H*), the kinetics of the ALI monolayer model mirrored that of the primary airway epithelial monolayers; both showed slow beginning (0–48 hpi) followed by an exponential increase in E gene levels from 48 to 72 hpi. The submerged monolayer model showed sustained viral infection during the 48–72 hpi window (*Figure 3—figure supplement 3B*, left). In the case of AT2 cells, the 48–72 hpi window was notably missing in monolayers of hiPSC-derived AT2 cells (*Figure 3H* and *Figure 3—figure supplement 3B*, right). When we specifically analyzed the kinetics of viral E gene expression during the late phase (48–72 hpi window), we found that proximal airway models (human bronchial airway epi [HBEpC]) showed high levels of sustained infectivity than distal models (human small airway epi [HSAEpC] and AT2) to viral replication (*Figure 3—figure supplement 3C*); the ALO monolayers showed intermediate sustained infectivity (albeit with variability). All models showed extensive cell death and detachment by 96 hr and, hence, were not analyzed. Finally, using the E gene as a readout, we asked if ALO models could be used as platforms for preclinical drug screens. As a proof of concept, we tested the efficacy of nucleoside analog N$^4$-hydroxycytidine (NHC; EIDD-parent) and its derivative pro-drug, EIDD-2801; both have been shown to inhibit viral replication, in vitro and in SARS-CoV-2-challenged ferrets (*Cox et al., 2021*; *Sheahan et al., 2020*). ALO monolayers plated in 384-wells were pretreated for 4 hr with the compounds or DMSO (control) prior to infection and assessed at 48 hpi for the abundance of E gene in the monolayers. Both compounds effectively reduced the viral titer in a dose-dependent manner (*Figure 3I*), and the pro-drug derivative showed a better efficacy, as shown previously.

Taken together, these findings show that sustained viral infectivity is best simulated in monolayers that resemble the proximal mucociliary epithelium, that is, 2D monolayers of lung organoids grown as ALI models and the primary airway epithelia. Because prior studies conducted in patient-derived airway cells (*Hou et al., 2020*) mirror what we see in our monolayers, we conclude that proximal airway cells within our mixed-cellular model appear to be sufficient to model viral infectivity in COVID-19.

Findings also validate optimized protocols for the adaptation of ALO monolayers in miniaturized 384-well formats for use in high-throughput drug screens.

## Differentiated 2D monolayers show that host immune response is higher in distal than proximal epithelia

Next, we asked if the newly generated lung models accurately recapitulate the host immune response in COVID-19. To this end, we analyzed the infected ALO monolayers (both the submerged and ALI variants) as well as the airway epithelial (HSAEpC) and AT2 monolayers by RNA seq and compared them all against the transcriptome profile of lungs from deceased COVID-19 patients. We did this analysis in two steps of reciprocal comparisons: (i) The actual human disease-derived gene signature was assessed for its ability to distinguish infected from uninfected disease models (in *Figure 4*). (ii) The ALO model-derived gene signature was assessed for its ability to distinguish healthy from diseased patient samples (in *Figure 5*). A publicly available dataset (GSE151764) *Nienhold et al., 2020*, comprising lung transcriptomes from victims deceased either due to noninfectious causes (controls) or due to COVID-19, was first analyzed for differentially expressed genes (DEGs; *Figure 4A and B*). This cohort was chosen as a test cohort over others because it was the largest one available at the time of this study with appropriate postmortem control samples. DEGs showed an immunophenotype that was consistent with what is expected in viral infections (*Figure 4C*, *Table 4*, and *Figure 4—figure supplement 1*) and showed overrepresentation of pathways such as interferon, immune, and cytokine signaling (*Figure 4D*, *Table 5*, and *Figure 4—figure supplement 2*). DEG signatures and reactome pathways that were enriched in the test cohort were fairly representative of the host immune response observed in patient-derived respiratory samples in multiple other validation cohorts; the signature derived from the test cohort could consistently classify control (normal) samples from COVID-19-samples (receiver operating characteristics area under the curve [ROC AUC] 0.89–1.00 across the board; *Figure 4E*). The most notable finding is that the patient-derived signature was able to perfectly classify the EpCAM-sorted epithelial fractions from the bronchoalveolar lavage fluids of infected and healthy subjects (ROC AUC 1.00; GSE145926-Epithelium *Liao et al., 2020*), suggesting that the respiratory epithelium is a major site where the host immune response is detected in COVID-19. When compared to existing organoid models of COVID-19, we found that the patient-derived COVID-19-lung signature was able to perfectly classify infected vs. uninfected late passages (>50) of hiPSC-derived AT1/2 monolayers (GSE155241) *Han et al., 2020a* and infected vs. uninfected liver and pancreatic organoids (*Figure 4F*). The COVID-19-lung signatures failed to classify commonly used respiratory models, for example, A549 cells and bronchial organoids, as well as intestinal organoids (*Figure 4F*). A similar analysis on our own lung models revealed that the COVID-19-lung signature was induced in submerged monolayers with distal-predominant AT2→AT1 differentiation, but not in the proximal-predominant ALI model (ROC AUC 1.00 and 0.50, respectively; *Figure 4G*). The ALI model and the small airway epithelia, both models that mimic the airway epithelia (and lack alveolar pneumocytes; see *Figure 3B*), failed to mount the patient-derived immune signatures (*Figure 4H*, left). These findings suggested that the presence of alveolar pneumocytes is critical for emulating host response. To our surprise, induction of the COVID-19-lung signature also failed in hiPSC-derived AT2 monolayers (*Figure 4H*, right), indicating that AT2 cells are unlikely to be the source of such host response. These findings indicate that both proximal airway and AT2 cells, when alone, are insufficient to induce the host immune response that is encountered in the lungs of COVID-19 patient.

Next, we analyzed the datasets from our ALO monolayers for DEGs when challenged with SARS-COV-2 (*Figure 5A,B*). Genes and pathways upregulated in the infected lung organoid-derived monolayer models (*Figure 5—figure supplements 1–2*) overlapped significantly with those that were upregulated in the COVID-19 lung signature (compare *Figure 4C,D* with *Figure 5C,D*, *Table 6*, *Table 7*, *Table 8*). We observed only a partial overlap (ranging from ~22–55% across various human datasets; *Figure 5—figure supplement 3*) in upregulated genes and no overlaps among downregulated genes between model and disease (COVID-19; *Figure 5E*). Because the degree of overlap was even lesser (ranging from ~10 to 25% across various human datasets; *Figure 5—figure supplement 3*) in the case of another publicly released model (GSE160435) (*Mulay et al., 2020*), these discrepancies between the model and the actual disease likely reflect the missing stromal and immune components in our organoid monolayers. Regardless of these missing components, the model-derived DEG signature was sufficient to consistently and accurately classify diverse cohorts of patient-derived respiratory

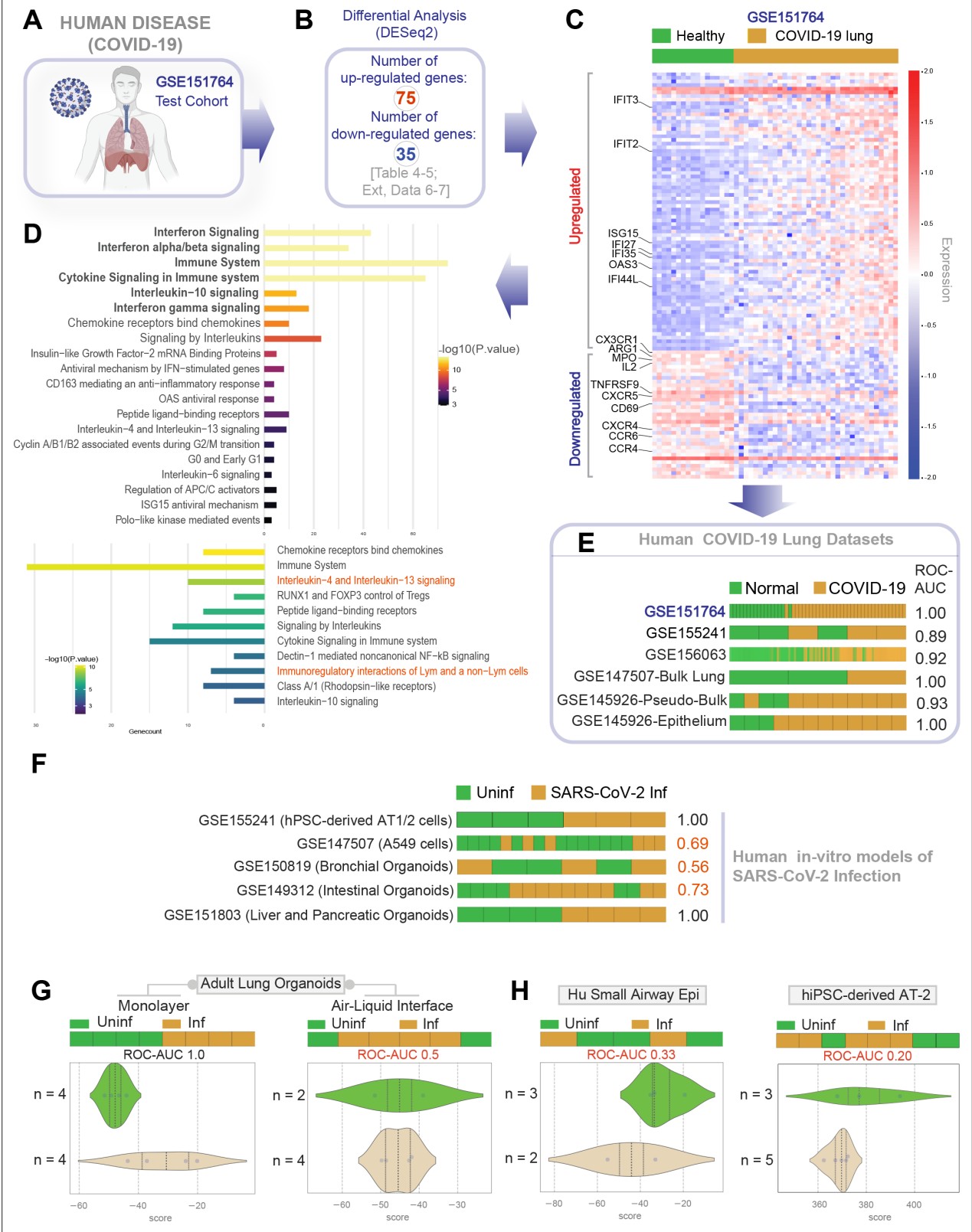

**Figure 4.** Gene expression patterns in the lungs of patients with COVID-19 (actual disease) are recapitulated in lung organoid monolayers infected with SARS-CoV-2 (disease model). (**A–C**) Publicly available RNA seq datasets (GSE151764) of lung autopsies from patients who were deceased due to COVID-19 or noninfectious causes (healthy normal control) were analyzed for differential expression of genes (**B**). The differentially expressed genes (DEGs) are displayed as a heatmap labeled with selected genes in (**C**). See also *Figure 4—figure supplement 1* for the same heatmap with all genes

*Figure 4 continued on next page*

*Figure 4 continued*

labeled. (**D**) Reactome-pathway analysis shows the major pathways up- or downregulated in the COVID-19-afflicted lungs. See also ***Figure 4—figure supplement 2*** for visualization as hierarchical ReacFoam. (**E**) Bar plots display the ability of the DEGs in the test cohort (GSE151764) to classify human COVID-19 respiratory samples from four other independent cohorts. (**F**) Bar plots display the ability of the DEGs in the test cohort (GSE151764) to classify published in vitro models for SARS-CoV-2 infection where RNA seq datasets were either generated in this work or publicly available. (**G, H**) Bar (top) and violin (bottom) plots compare the relative accuracy of disease modeling in four in vitro models used in the current work, as determined by the induction of COVID-19 lung signatures in each model. (**G**) Monolayer (left) and air-liquid interphase (ALI) models (right) prepared using adult lung organoids (ALOs). (**H**) Primary human small airway epithelium (left) and hiPSC-derived AT2 monolayers (right). Table 6 lists details regarding the patient cohorts/tissue or cell types represented in each transcriptomic dataset.

The online version of this article includes the following figure supplement(s) for figure 4:

**Figure supplement 1.** Differential expression analysis of RNA seq datasets from lung autopsies (normal vs. COVID-19).

**Figure supplement 2.** Reactome-pathway analysis of differentially expressed genes in lung autopsies (normal vs. COVID-19).

samples (ROC AUC ranging from 0.88 to 1.00; ***Figure 5F***); the model-derived DEG signature was significantly induced in COVID-19 samples compared to normal controls (***Figure 5G,H***). Most importantly, the model-derived DEG signature was significantly induced in the epithelial cells recovered from bronchoalveolar lavage (***Figure 5I***).

Taken together, these cross-validation studies from disease to model (***Figure 4***) and vice versa (***Figure 5***) provide an objective assessment of the match between the host response in COVID-19 lungs and our submerged ALO monolayers. Such a match was not seen in the case of the other models, for example, the proximal airway-mimic ALI model, HSAEpC monolayer, or hiPSC-derived AT2 models. Because the submerged ALO monolayers contained both proximal airway epithelia (basal cells) and promoted AT2→AT1 differentiation, findings demonstrate that mixed cellular monolayers can mimic the host response in COVID-19. A subtractive analysis revealed that the cell type that is shared between models that showed induction of host response signatures [i.e., ALO submerged monolayers and GSE155241 (***Han et al., 2020a***; ***Figure 5F***)] but is absent in models that do not show such response (hu bronchial organoids, small airway epi, ALI-model of ALO) is AT1. We conclude that distal differentiation from AT2→AT1, a complex process that comprises distinct intermediates (***Choi et al., 2020***), is essential for modeling the host immune response in COVID-19. Further experimental evidence is needed to directly confirm if and which intermediate states during the differentiation of AT2 to AT1 are essential for the immune response to COVID19.

## Both proximal and distal airway epithelia are required to mount the overzealous host response in COVID-19

We next asked which model best simulated the overzealous host immune response that has been widely implicated in fatal COVID-19 (***Lowery et al., 2021***; ***Lucas et al., 2020***; ***Schultze and Aschenbrenner, 2021***). To this end, we relied upon a recently described artificial intelligence (AI)-guided definition of the nature of the overzealous response in fatal COVID-19 (***Sahoo et al., 2021***). Using ACE2 as a seed gene, a 166-gene signature was identified and validated as an invariant immune response that was shared among all respiratory viral pandemics, including COVID-19 (***Figure 6A***). A subset of 20 genes within the 166-gene signature was subsequently identified as a determinant of disease severity/fatality; these 20 genes represented translational arrest, senescence, and apoptosis. These two signatures, referred to as ViP (166-gene) and severe ViP (20-gene) signatures, were used as a computational framework to first vet existing SARS-CoV-2 infection models that have been commonly used for therapeutic screens (***Figure 6B–D***). Surprisingly, we found that each model fell short in one way or another. For example, the Vero E6, which is a commonly used cultured cell model, showed a completely opposite response; instead of being induced, both the 166-gene and 20-gene ViP signatures were suppressed in infected Vero E6 monolayers (***Figure 6B***). Similarly, neither ViP signature was induced in the case of SARS-CoV-2-challenged human bronchial organoids (***Suzuki, 2020***) (***Figure 6C***). Finally, in the case of the hiPSC-derived AT1/2 organoids, which recapitulated the COVID-19-lung derived immune signatures (in ***Figure 4F***), the 166-gene ViP signature was induced significantly (***Figure 6D***, top), but the 20-gene severity signature was not (***Figure 6D***, bottom). These findings show that none of the existing models capture the overzealous host immune response that has been implicated in a fatality.

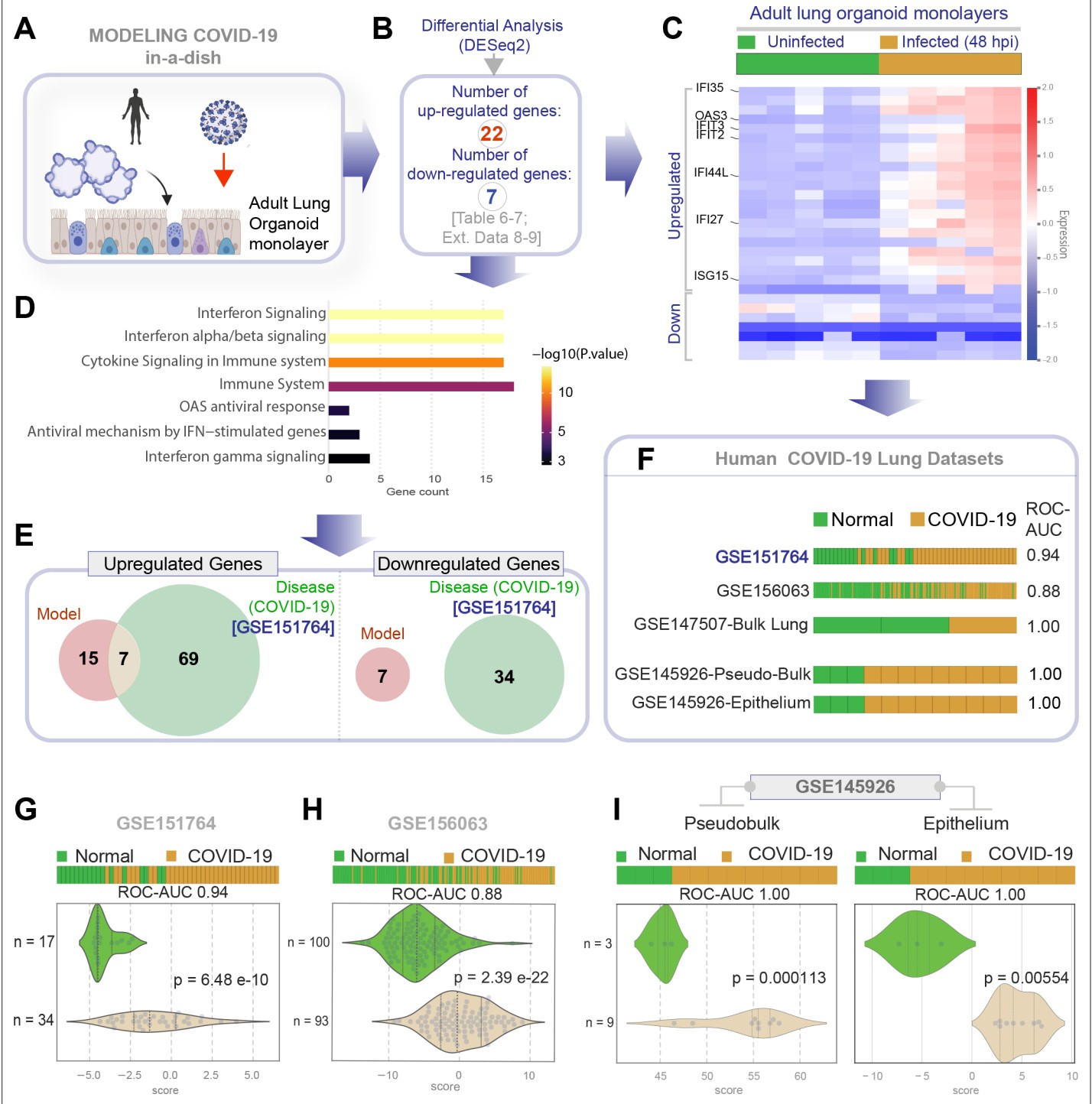

**Figure 5.** Genes and pathways induced in the SARS-CoV-2-infected lung organoid monolayers (disease model) are induced also in the lungs of COVID-19 patients (actual disease). (**A–C**) Adult lung organoid monolayers infected or not with SARS-CoV-2 were analyzed by RNA seq and differential expression analysis. Differentially expressed genes (DEGs; **B**) are displayed as a heatmap in (**C**). While only selected genes are labeled in panel (**C**) (which represent overlapping DEGs between our organoid model and publicly available COVID-19 lung dataset, GSE151764), the same heatmap is presented in *Figure 5—figure supplement 1* with all genes labeled. (**D**) Reactome-pathway analysis shows the major pathways upregulated in SARS-CoV-2-infected lung organoid monolayers. See also *Figure 5—figure supplement 2* for visualization as hierarchical ReacFoam. (**E**) A Venn diagram showing overlaps in DEGs between model (current work; **B**) and disease (COVID-19 lung dataset, GSE151764; *Figure 4*). (**F**) Bar plots display the ability of the DEGs in infected lung monolayers to classify human normal vs. COVID-19 respiratory samples from five independent cohorts. (**G–I**) Bar (top) and violin (bottom) plots compare the accuracy of disease modeling in three publicly available human lung datasets, as determined by the significant

*Figure 5 continued on next page*

*Figure 5 continued*

induction of the DEGs that were identified in the SARS-CoV-2-challenged monolayers. See also Table 6, which enlists details regarding the patient cohorts/tissue or cell types represented in each transcriptomic dataset.

The online version of this article includes the following figure supplement(s) for figure 5:

**Figure supplement 1.** Differential expression analysis of RNA seq datasets from adult lung organoid monolayers, infected or not, with SARS-CoV-2.

**Figure supplement 2.** Reactome-pathway analysis of differentially expressed genes in lung organoid monolayers infected with SARS-CoV-2.

**Figure supplement 3.** Head-to-head comparison of our adult lung organoid (ALO)-derived model of COVID-19 versus another lung organoid model in their ability to recapitulate the differentially expressed genes (DEGs) observed in lung tissues from fatal cases of COVID-19.

Our lung models showed that both the 166- and 20-gene ViP signatures were induced significantly in the submerged ALO-derived monolayers that had distal differentiation (*Figure 6E*, left), but not in the proximal-mimic ALI model (*Figure 6E*, right). Neither signatures were induced in monolayers of small airway epithelial cells (*Figure 6F*) or hiPSC-derived AT2 cells (*Figure 6G*). Finally, we analyzed a recently published lung organoid model that supports robust SARS-CoV-2 infection; this model was generated using multipotent SOX2+ SOX9+ lung bud tip (LBT) progenitor cells that were isolated from the canalicular stage of human fetal lungs (~16–17 weeks post-conception) (*Lamers et al., 2021*). Despite mixed cellularity (proximal and distal), this fetal lung organoid model failed to induce the ViP signatures (*Sahoo et al., 2021*) (*Figure 6H*). These findings indicate that despite having mixed cellular composition and the added advantage of being able to support robust viral replication (achieving ~5 log-fold increase in titers), the model lacks the signature host response that is seen in all human samples of COVID-19.

Taken together with our infectivity analyses, these findings demonstrate that although the proximal airway epithelia and AT2 cells may be infected, and as described by others (*Dye et al., 2015*; *Hou et al., 2020*), may be vital for mounting a viral response and for disease transmission, these cells alone cannot mount the overzealous host immune response that is associated with the fatal disease. Similarly, even though the alveolar pneumocytes, AT1 and AT2 cells, are sufficient to mount the host immune response, in the absence of proximal airway components, they too are insufficient to recapitulate the severe ViP signature that is characterized by cellular senescence and apoptosis. However, when both proximal and distal components are present, that is, basal, ciliated, and AT1 cells, the model mimicked the overzealous host immune response in COVID-19 (*Figure 6I*).

## Discussion

The most important discovery we report here is the creation of adult lung organoids that are complete with both proximal airway and distal alveolar epithelia; these organoids can not only be stably propagated and expanded in 3D cultures but also used as monolayers of mixed cellularity for modeling viral and host immune responses during respiratory viral pandemics. Furthermore, an objective analysis of this model and other existing SARS-CoV-2-infected lung models against patient-lung-derived transcriptomes showed that the model that most closely emulates the elements of viral infectivity, lung injury, and inflammation in COVID-19 is one that contained both proximal and distal alveolar signatures (*Figure 6H*), whereas the presence of just one or the other fell short.

There are three important impacts of this work. First, the successful creation of adult human lung organoids that are complete with both proximal and distal signatures has not been accomplished before. Previous works show the successful use of airway basal cells for organoid creationtion, but ensuring the completeness of the model with all other lung cells has been challenging to create (*Nikolić and Rawlins, 2017*). The multicellularity of the lung has been a daunting challenge that many experts have tried to recreate in vitro; in fact, the demand for perfecting such a model has always remained high, not just in the current context of the COVID-19 pandemic but also with the potential of future pandemics. We have provided evidence that the organoids that were created using our methodology retain proximal and distal cellularity throughout multiple passages and even within the same organoid. Although a systematic design of experiment (DoE) approach (*Bukys et al., 2020*) was not involved in getting to this desirable goal, a rationalized approach was taken. For example, a Wnt/R-spondin/Noggin-containing conditioned media was used as a source of the so-called 'niche factors' for any organoid growth (*Sato and Clevers, 2015*). This was supplemented with recombinant

**Table 4.** Upregulated genes and pathways: healthy vs COVID-19 lung (GSE151764).

**Genes**

| | | | | |
|---|---|---|---|---|
| BRCA2 | XAGE1B | CDK1 | SNAI2 | CXCL11 |
| CYBB | CCR5 | GBP1 | IFITM1 | IFI27 |
| KRT5 | CCR2 | HLA-G | GZMB | IFI35 |
| C1QB | ALOX15B | IDO1 | CD163 | TDO2 |
| FCGR1A | CMKLR1 | ISG20 | CD38 | GZMA |
| IL10 | MX1 | LAG3 | BST2 | OAS3 |
| IL6 | TNFRSF17 | MAD2L1 | BUB1 | POU2AF1 |
| CD44 | CCR1 | CXCL9 | CCL20 | CXCL13 |
| CD276 | CXCR3 | MKI67 | CCNB2 | GNLY |
| DMBT1 | SLAMF8 | IFIT2 | TNFSF18 | IFIT3 |
| DDX58 | IL21 | IFIT1 | ISG15 | TOP2A |
| TNFAIP8 | FOXM1 | CXCL10 | CDKN3 | LILRB1 |
| LAMP3 | IFIH1 | IRF4 | C1QA | HERC6 |
| KIAA0101 | IFI6 | PSMB9 | OAS1 | TNFSF13B |
| MELK | PDCD1LG2 | CCL18 | OAS2 | IFI44L |
| | | | | STAT1 |

**Pathways**

| Name | p-value | FDR |
|---|---|---|
| Interferon signaling | 1.11E-16 | 1.11E-14 |
| Interferon alpha/beta signaling | 1.11E-16 | 1.11E-14 |
| Cytokine signaling in immune system | 1.11E-16 | 1.11E-14 |
| Immune ssystem | 1.11E-16 | 1.11E-14 |
| Interleukin-10 signaling | 9.85E-13 | 7.88E-11 |
| Interferon gamma signaling | 9.26E-12 | 6.11E-10 |
| Chemokine receptors bind chemokines | 1.08E-10 | 6.17E-09 |
| Signaling by interleukins | 6.81E-09 | 3.41E-07 |
| Insulin-like growth factor-2 mRNA binding proteins (IGF2BPs/IMPs/VICKZs) bind RNA | 1.27E-07 | 0.000005581122619 |
| Antiviral mechanism by IFN-stimulated genes | 0.000001933058349 | 0.00007732233398 |
| CD163 mediating an anti-inflammatory response | 0.000007798676169 | 0.0002807523421 |
| OAS antiviral response | 0.00001020870997 | 0.0003368874291 |
| Peptide ligand-binding receptors | 0.00001714057687 | 0.0005142173062 |
| Interleukin-4 and Interleukin-13 signaling | 0.0001014948661 | 0.002841856252 |
| Cyclin A/B1/B2-associated events during G2/M transition | 0.0001887816465 | 0.00490832281 |
| G0 and early G1 | 0.0003607121838 | 0.009017804596 |
| Interleukin-6 signaling | 0.0004656678444 | 0.01071036042 |
| ISG15 antiviral mechanism | 0.0008313991988 | 0.01745938317 |
| Regulation of APC/C activators between G1/S and early anaphase | 0.0008313991988 | 0.01745938317 |

*Table 4 continued on next page*

*Table 4 continued*

**Genes**

| | | |
|---|---|---|
| Polo-like kinase-mediated events | 0.001110506513 | 0.02221013026 |
| APC/C-mediated degradation of cell cycle proteins | 0.001308103581 | 0.02354586446 |
| Regulation of mitotic cell cycle | 0.001308103581 | 0.02354586446 |
| G2/M DNA replication checkpoint | 0.001750156332 | 0.02975265764 |
| Class A/1 (rhodopsin-like receptors) | 0.002355063045 | 0.03537666782 |
| Interleukin-6 family signaling | 0.002358444521 | 0.03537666782 |
| TNFs bind their physiological receptors | 0.002358444521 | 0.03537666782 |

FGF7/10; FGF7 is known to help cell proliferation and differentiation and is required for normal branching morphogenesis (*Padela et al., 2008*), whereas FGF10 helps in cell maturation (*Rabata et al., 2020*) and in alveolar regeneration upon injury (*Yuan et al., 2019*). Together, they are likely to have directed the differentiation toward distal lung lineages (hence, the preservation of alveolar signatures). The presence of both distal alveolar and proximal ciliated cells was critical: proximal cells were required to recreate sustained viral infectivity, and the distal alveolar pneumocytes, in particular, the ability of AT2 cells to differentiate into AT1 pneumocytes was essential to recreate the host response. It is possible that the response is mediated by a distinct AT2-lineage population, that is, damage-associated transient progenitors (DATPs), which arise as intermediates during AT2→AT1 differentiation upon injury-induced alveolar regeneration (*Choi et al., 2020*). Although somewhat unexpected, the role of AT1 pneumocytes in mounting innate immune responses has been documented before in the context of bacterial pneumonia (*Yamamoto et al., 2012*; *Wong and Johnson, 2013*). In work (*Huang et al., 2020*) that was published during the preparation of this article, authors used long-term ALI models of hiPSC-derived AT2 monolayers (in growth conditions that inhibit AT2→AT1 differentiation, as we did here for our AT2 model) and showed that SARS-CoV-2 induces iAT2-intrinsic cytotoxicity and inflammatory response, but failed to induce type 1 interferon pathways (IFN α/β). It is possible that prolonged culture of iAT2 pneumocytes gives rise to some DATPs but cannot robustly do so in the presence of inhibitors of AT1 differentiation. This (spatially segregated viral and host immune response) is a common theme across many lung infections (including bacterial pneumonia and other viral pandemics (*Hou et al., 2020*; *Taubenberger and Morens, 2008*; *Weinheimer et al., 2012*; *Chan et al., 2013*) and hence, this mixed cellularity model is appropriate for use in modeling diverse lung infections and respiratory pandemics to come.

Second, among all the established lung models so far, ours features four key properties that are desirable whenever disease models are being considered for their use in HTP modes for rapid screening of candidate therapeutics and vaccines: (i) reproducibility, propagability, and scalability; (ii) cost-effectiveness; (iii) personalization; and (iv) modularity, with the potential to add other immune and nonimmune cell types to our multicellular complex lung model. We showed that the protocol we have optimized supports isolation, expansion, and propagability at least up to 12–15 passages (at the time of submission of this work), with documented retention of proximal and distal airway components up to P8 (by RNA seq). We noted some variability of cell types between patient to patient, and between early and late passages of ALOs, which is probably because of the heterogeneity of organoids isolated from patient's lung specimens. Feasibility has also been established for scaling up and optimizing the conditions for them to be used in miniaturized 384-well infectivity assays. We also showed that the protocols for generating lung organoids could be reproduced in both genders, and regardless of the donor's smoking status, consistency in outcome and growth characteristics was observed across all isolation attempts. The ALOs are also cost-effective; the need for exclusive reliance on recombinant growth factors was replaced at least in part with conditioned media from a commonly used cell line (L-WRN/ ATCC CRL-2647 cells). Such media has a batch-to-batch stable cocktail of Wnt, R-Spondin, and Noggin, and has been shown to improve reproducibility in the context of GI organoids in independent laboratories (*VanDussen et al., 2019*). By that token, our culture conditions may have also improved reproducibility. The major disadvantage, however, remains

Table 5. Downregulated genes and pathways: healthy vs COVID-19 lung (GSE151764).

**Genes**

| CX3CR1 | JAML | KLRB1 | GRAP2 | CD226 |
|---|---|---|---|---|
| ARG1 | CX3CR1 | LY9 | MMP9 | CD160 |
| MPO | HLA-DQB2 | CCL17 | RORC | FOXP3 |
| IL2 | TNFRSF9 | CCL22 | CCR4 | CRTAM |
| BCL2 | CXCR5 | TCF7 | IRS1 | CCR6 |
| CA4 | CD1C | CXCR4 | ITK | CEACAM8 |
| IGF1R | CD69 | CD83 | KLRG1 | PTGS2 |

**Pathways**

| Name | p-value | FDR |
|---|---|---|
| Chemokine receptors bind chemokines | 2.85E-11 | 4.98E-09 |
| Immune system | 1.25E-10 | 1.09E-08 |
| Interleukin-4 and interleukin-13 signaling | 2.82E-09 | 1.64E-07 |
| RUNX1 and FOXP3 control the development of regulatory T lymphocytes (Tregs) | 4.31E-07 | 0.00001853717999 |
| Peptide ligand-binding receptors | 6.71E-07 | 0.00002348305743 |
| Signaling by Interleukins | 0.000001503658493 | 0.0000436060963 |
| Cytokine signaling in Immune system | 0.00002606505855 | 0.0006516264636 |
| Dectin-1-mediated noncanonical NF-kB signaling | 0.00008640543215 | 0.001814514075 |
| Immunoregulatory interactions between a lymphoid and a non-lymphoid cell | 0.0001083388675 | 0.002058438482 |
| Class A/1 (rhodopsin-like receptors) | 0.0001833048828 | 0.003116183008 |
| Interleukin-10 signaling | 0.0002366961934 | 0.0035504429 |
| RUNX3 regulates immune response and cell migration | 0.0005791814113 | 0.007747184934 |
| Extra-nuclear estrogen signaling | 0.0005959373026 | 0.007747184934 |
| BH3-only proteins associate with and inactivate anti-apoptotic BCL-2 members | 0.0006992547523 | 0.008391057028 |
| CLEC7A (Dectin-1) signaling | 0.0008228035145 | 0.00905083866 |
| Generation of second messenger molecules | 0.001171991908 | 0.01171991908 |
| Innate immune system | 0.001676404092 | 0.01572360367 |
| GPCR ligand binding | 0.001747067074 | 0.01572360367 |
| Adaptive immune system | 0.002059835991 | 0.01853852391 |
| Estrogen-dependent nuclear events downstream of ESR-membrane signaling | 0.00467005583 | 0.03736044664 |
| C-type lectin receptors (CLRs) | 0.00545804495 | 0.0436643596 |
| Transcriptional regulation by RUNX3 | 0.008124332599 | 0.05687032819 |
| BMAL1:CLOCK, NPAS2 activates circadian gene expression | 0.009518272709 | 0.06662790896 |
| ESR-mediated signaling | 0.01207376237 | 0.08451633662 |
| Transcriptional regulation by RUNX1 | 0.01288156371 | 0.08786708747 |
| TCR signaling | 0.01464451458 | 0.08786708747 |

**Table 6.** The list of GSE numbers used in the figures.

| GSE# | Cell type/tissue | References | Figure |
|------|------------------|------------|--------|
| GSE132914 | Tissue from idiopathic pulmonary fibrosis subjects and donor controls | PMID:32991815 | *Figure 1A* |
| GSE151764 | COVID-19 and normal lung tissue post-mortem | PMID:33033248 | *Figure 4A–E*, *Figure 5E–G* |
| GSE155241 | hPSC lung organoids and colon organoids infected with SARS-CoV-2 | PMID:33116299 | *Figure 4E,F*, *Figure 6D* |
| GSE156063 | Upper airway of COVID-19 patients and other acute respiratory illnesses | PMID:33203890 | *Figure 4E*, *Figure 5F,H* |
| GSE147507 | A549 cells and bulk lung | PMID:32416070; PMID:33782412 | *Figure 4E,F*, *Figure 5F* |
| GSE145926 | Bronchoalveolar lavage fluid (BALF) immune cells from COVID-19 and healthy subjects | PMID:32398875 | *Figure 4E*, *Figure 5F,I* |
| GSE150819 | Human bronchial organoids | From commercially available HBEpC | *Figure 4F*, *Figure 6C* |
| GSE149312 | Intestinal organoids infected with SARS-CoV or SARS-CoV-2 | PMID:32358202 | *Figure 4F* |
| GSE151803 | hPSC-derived pancreatic and lung organoids infected with SARS-CoV-2 | No publication yet | *Figure 4F* |
| GSE153940 | Vero E6 control or SARS-CoV-2-infected cells | PMID:32707573 | *Figure 6B* |
| GSE153218 | SARS-CoV-2-infected bronchoalveolar cells derived from organoids grown using progenitor cells from human fetal lung but tip (LBT). | PMID:33283287 | *Figure 6H* |

that the composition of the media is undefined. Because the model is propagable, repeated iPSC-reprogramming (another expensive step) is also eliminated, further cutting costs compared to many other models. As for personalization, our model is derived from adult lung stem cells from deep lung biopsies; each organoid line was established from one patient. By avoiding iPSCs or expanded potential stem cells (EPSCs), this model not only captures genetics but also retains organ-specific epigenetic programming in the lung, and hence, potentially additional programming that may occur in disease (such as in the setting of chronic infection, injury, inflammation, somatic mutations, etc.). The ability to replicate donor phenotype and genotype in vitro allows for potential use as preclinical human models for phase '0' clinical trials. As for modularity, by showing that the 3D lung organoids could be used as polarized monolayers on transwells to allow infectious agents to access the apical surface (in this case, SARS-CoV-2), we demonstrate that the organoids have the potential to be reverse-engineered with additional components in a physiologically relevant spatially segregated manner: for example, immune and stromal cells can be placed in the lower chamber to model complex lung diseases that are yet to be modeled and have no cure (e.g., idiopathic pulmonary fibrosis, etc.).

Third, the value of the ALO models is further enhanced due to the availability of companion readouts/biomarkers (e.g., ViP signatures in the case of respiratory viral pandemics, or monitoring the E gene, or viral shedding, etc.) that can rapidly and objectively vet treatment efficacy based on set therapeutic goals. Of these readouts, the host response, as assessed by ViP signatures, is a key vantage point because an overzealous host response is what is known to cause fatality. Recently, a systematic review of the existing preclinical animal models revealed that most of the animal models of COVID-19 recapitulated mild patterns of human COVID-19; no severe illness associated with mortality was observed, suggesting a wide gap between COVID-19 in humans (*Spagnolo et al., 2020*) and animal models (*Ehaideb et al., 2020*). It is noteworthy that alternative models that effectively support viral replication, such as the proximal airway epithelium or iPSC-derived AT2 cells (analyzed in this work) or a fetal lung bud tip-derived organoid model recently described by others (*Lamers et al., 2021*), do not recapitulate the host response in COVID-19. The lung model we present here is distinct from all currently available other models (see *Table 1*) because of the confirmed presence of both proximal and distal airway cell types over successive passages, which is yet to be accomplished for adult lung organoid models. Another distinguishing feature of our model is the way we rigorously validated its usefulness in modeling COVID-19 via computational approaches. We confirmed, based on the gene expression changes upon SARS-CoV-2-challenge, that our model most closely recapitulates the

**Table 7.** Upregulated genes and pathways: uninfected vs infected (48 hpi) lung organoid monolayers.

**Genes**

| | | | |
|---|---|---|---|
| IFI35 | EPSTI1 | AMIGO2 | IFITM2 |
| SLC4A11 | CMPK2 | WARS1 | FAAP100 |
| APOL1 | OASL | IFI27 | ISG15 |
| OAS3 | IFI44L | CD14 | SLC35F6 |
| IFIT3 | IFI44 | SAMD9L | |
| IFIT2 | PARP9 | SRP9P1 | |

**Pathways**

| Name | p-value | FDR |
|---|---|---|
| Interferon signaling | 1.11E-16 | 4.22E-15 |
| Interferon alpha/beta signaling | 1.11E-16 | 4.22E-15 |
| Cytokine signaling in Immune system | 1.15E-10 | 2.89E-09 |
| Immune system | 0.000002540114879 | 0.00004826218271 |
| OAS antiviral response | 0.0004764545663 | 0.007146818495 |
| Antiviral mechanism by IFN-stimulated genes | 0.001033347261 | 0.01240016713 |
| Interferon gamma signaling | 0.001889694619 | 0.02078664081 |
| Transfer of LPS from LBP carrier to CD14 | 0.006318772245 | 0.05686895021 |
| TRIF-mediated programmed cell death | 0.02091267586 | 0.1656073329 |
| MyD88 deficiency (TLR2/4) | 0.03733748271 | 0.1656073329 |
| IRAK2-mediated activation of TAK1 complex upon TLR7/8 or 9 stimulation | 0.03733748271 | 0.1656073329 |
| TRAF6-mediated induction of TAK1 complex within TLR4 complex | 0.03937173812 | 0.1656073329 |
| IRAK4 deficiency (TLR2/4) | 0.03937173812 | 0.1656073329 |
| Activation of IRF3/IRF7 mediated by TBK1/IKK epsilon | 0.04140183322 | 0.1656073329 |
| Caspase activation via death receptors in the presence of ligand | 0.04140183322 | 0.1656073329 |
| IKK complex recruitment mediated by RIP1 | 0.04948077476 | 0.1855013265 |

human disease, that is, COVID-19. Analyses also pinpointed the importance of two factors that were

**Table 8.** Downregulated genes and pathways: uninfected vs. infected (48 hpi) lung organoid monolayers.

| | | | |
|---|---|---|---|
| AC093392.1 | | ARHGAP19 | HLA-V | RN7SL718P |
| MT-TV | | AC138969.3 | AC016766.1 | |

Pathways

| Name | p-value | FDR |
|---|---|---|
| rRNA processing in the mitochondrion | 0.01892731246 | 0.08366120773 |
| tRNA processing in the mitochondrion | 0.02127149105 | 0.08366120773 |
| Mitochondrial translation termination | 0.04399155446 | 0.08366120773 |
| Mitochondrial translation elongation | 0.04399155446 | 0.08366120773 |
| Mitochondrial translation initiation | 0.04490921762 | 0.08366120773 |
| Mitochondrial translation | 0.04765767844 | 0.08366120773 |

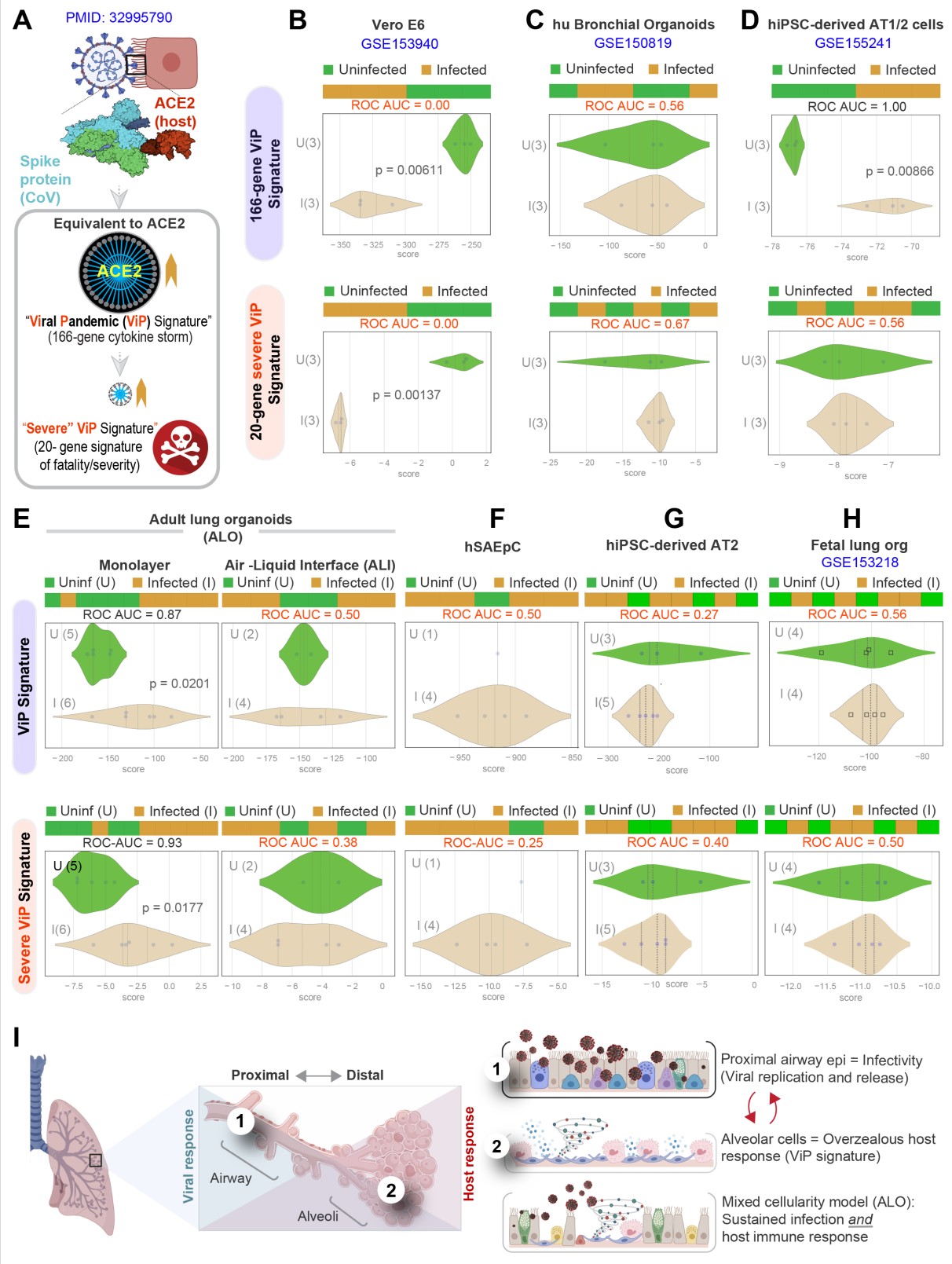

**Figure 6.** Both proximal and distal airway components are required to model the overzealous host response in COVID-19. (**A**) Schematic summarizing the immune signatures identified based on ACE2-equivalent gene induction observed invariably in any respiratory viral pandemic. The 166-gene ViP signature captures the cytokine storm in COVID-19, whereas the 20-gene subset severe ViP signature is indicative of disease severity/fatality. (**B–D**) Publicly available RNA seq datasets from commonly used lung models, Vero E6 (**B**), human bronchial organoids (**C**), and hPSC-derived AT1/2 cell-

*Figure 6 continued on next page*

*Figure 6 continued*

predominant lung organoids are classified using the 166-gene ViP signature (top row) and 20-gene severity signature (bottom row). (**E–G**) RNA seq datasets generated in this work using either human small airway epithelial cells (**E**), adult lung organoids as submerged or air-liquid interphase (ALI) models (left and right, respectively, in **F**) and hiPSC-derived AT2 cells (**G**) were analyzed and visualized as in (**B–D**). (**H**) Publicly available RNA seq datasets from fetal lung organoid monolayers (*Lamers et al., 2021*) infected or not with SARS-CoV-2 were analyzed as in (**B–D**) for the ability of ViP signatures to classify infected (I) from uninfected (U) samples. Receiver operating characteristics area under the curve (ROC AUC) in all figure panels indicate the performance of a classification model using the ViP signatures. (**I**) Summary of findings in this work, its relationship to the observed clinical phases in COVID-19, and key aspects of modeling the same. Table 6 lists details regarding the patient cohorts/tissue or cell types represented in each transcriptomic dataset.

critical in modeling COVID-19: (i) adult source and (ii) model completeness, with both proximal and distal airway cells. We conclude that the model revealed here, in conjunction with the ViP signatures described earlier (*Sahoo et al., 2021*), could serve as a preclinical model with companion diagnostics to identify drugs that target both the viral and host response in pandemics.

## Limitations of the study

Our adult stem-cell-derived lung organoids, complete with all epithelial cell types, can model COVID-19, but remains a simplified/rudimentary version compared to the adult human organ. While we successfully demonstrated the proximo-distal mixed cellular composition of the ALOs using four different approaches (flow cytometry, RNA seq, confocal immunofluorescence, and targeted qPCR) and showed that such mixed cellularity is preserved during prolonged culture, the exact cellular proportion was not assessed here. Single-cell sequencing and multiplexed profiling by flow cytometry are some of the approaches that can provide such in-depth characterization to assess cellular composition at baseline and track how such composition changes upon infection and injury. For instance, although the epithelial contributions to the host response are important, they alone cannot account for the response of the immune cells and the nonimmune stromal cells, and their crosstalk with the epithelium. Given that epithelial inflammation and damage is propagated by vicious forward-feedback loops of multicellular crosstalk, it is entirely possible that the epithelial signatures induced in infected ALO-derived monolayers are also only a fraction of the actual epithelial response mounted in vivo. Regardless of the missing components, what appears to be the case is that we already have a model that recapitulates approximately a quarter to half of the genes that are induced across diverse COVID-19-infected patient samples. This model can be further improved by the simultaneous addition of endothelial cells and immune cells to better understand the pathophysiological basis for DAD, microangiopathy, and even organizing fibrosis with loss of lung capacity that has been observed in many patients (*Spagnolo et al., 2020*); these insights should be valuable to fight some of the long-term sequelae of COVID-19. Future work with flow cytometry and cell sorting of our lung organoids would help us understand each cell type's role in viral pathogenesis. Larger living biobanks of genotyped and phenotyped ALOs, representing donors of different age, ethnicity, predisposing conditions, and coexisting comorbidities, will advance our understanding of why SARS-CoV-2 and possibly other infectious agents may trigger different disease course in different hosts. Although we provide proof-of-concept studies in low-throughput mode demonstrating the usefulness of the ALOs as human preclinical models for screening therapeutics in phase '0' trials, optimization for the same to be adapted in HTP mode was not attempted here.

## Materials and methods

**Key resources table**

| Reagent type (species) or resource | Designation | Source or reference | Identifiers | Additional information |
|---|---|---|---|---|
| Antibody | Anti-ACE2 (mouse monoclonal) | Santa Cruz | Cat# sc390851 RRID::AB_2861379 | IF (1:100) |
| Antibody | Anti-human ACE2 (rat monoclonal) | BioLegend | Cat# 375802 RRID::AB_2860959 | IF (1:50) |

*Continued on next page*

*Continued*

| Reagent type (species) or resource | Designation | Source or reference | Identifiers | Additional information |
|---|---|---|---|---|
| Antibody | Anti-acetylated α tubulin (mouse monoclonal) | Santa Cruz | Cat# sc23950 RRID::AB_628409 | IF (1:500) FC (1:8000) |
| Antibody | Anti-AQP5 (mouse monoclonal) | Santa Cruz | Cat# sc514022 RRID::AB_2891066 | IF (1:100) FC (1:800) |
| Antibody | Anti-CC10 (mouse monoclonal) | Santa Cruz | Cat# sc365992 RRID::AB_10915481 | IF (1:100) |
| Other | DAPI | Invitrogen | Cat# D1306 RRID::AB_2629482 | IF (1:500) |
| Antibody | Recombinant anti-cytokeratin 5 (rabbit monoclonal) | Abcam | Cat# ab52635 RRID::AB_869890 | IF (1:100) FC (1:8000) |
| Antibody | Recombinant anti-mucin 5AC (rabbit monoclonal) | Abcam | Cat# ab229451 RRID::AB_2891067 | IF (1:150) FC (1:800) |
| Antibody | Anti-sodium potassium ATPase (rabbit monoclonal) | Abcam | Cat# ab76020 RRID::AB_1310695 | IF (1:400) |
| Antibody | Anti-occludin (mouse monoclonal) | Thermo Fisher | Cat# OC-3F10 RRID::AB_2533101 | IF (1:500) |
| Other | Phalloidin, Alexa Fluor 594 | Invitrogen | Cat# A12381 RRID:AB_2315633 | IF (1:500) |
| Other | Propidium iodide | Invitrogen | V13241 | FC (1:100) |
| Antibody | SARS-CoV/SARS-CoV-2 nucleocapsid antibody (mouse monoclonal) | Sino Biological | Cat# 40143-MM05 RRID::AB_2827977 | IF (1:250) IHC (1:500) |
| Antibody | Anti-SARS spike glycoprotein (mouse monoclonal) | Abcam | Cat# ab273433 RRID::AB_2891068 | IHC (1:250) |
| Antibody | anti-SP-B (mouse monoclonal) | Santa Cruz | Cat# sc133143 RRID::AB_2285686 | IF (1:100) FC (1:8000) |
| Antibody | Anti-prosurfactant protein C (rabbit polyclonal) | Abcam | Cat# ab90716 RRID::AB_10674024 | IF (1:150) |
| Antibody | Goat anti-rat IgG H&L secondary antibody, Alexa Flour 594 | Invitrogen | Cat# A-11007 RRID:AB_10561522 | IF (1:500) |
| Antibody | Goat anti-rabbit IgG H&L secondary antibody, Alexa Fluor 594 | Invitrogen | Cat# A-11012 RRID:AB_2534079 | IF (1:500) |
| Antibody | Goat anti-mouse IgG H&L secondary antibody, Alexa Fluor 488 | Invitrogen | Cat# A-11011 RRID:AB_143157 | IF (1:500) FC (1:1000) |
| Antibody | Goat anti-rabbit IgG H&L secondary antibody, Alexa Fluor 488 | Abcam | Cat# ab150077 RRID:AB_2630356 | FC (1:1000) |
| Other | Countess II Automated Cell Counter | Thermo Fisher Scientific | AMQAX1000 | Section 'The preparation of lung organoid-derived monolayers' |
| Other | Epithelial Volt-Ohm (TEER) Meter | Millipore | MERS00002 | Section 'Permeability of lung monolayer using FITC-dextran' |
| Other | Leica TCS SPE Confocal | Leica Microsystems | TCS SPE | Section 'Immunofluorescence' |
| Other | Power Pressure Cooker XL | Tristar Products | | Section 'Immunohistochemistry' |
| Other | Canon Rebel XS DLSR | Canon | | *Figure 2—figure supplement 1* |

*Continued on next page*

*Continued*

| Reagent type (species) or resource | Designation | Source or reference | Identifiers | Additional information |
|---|---|---|---|---|
| Other | MiniAmp Plus Thermal Cycler | Applied Biosystems | Cat# A37835 | Section 'Quantitative (q)RT-PCR' |
| Other | QuantStudio5 | Applied Biosystems | Cat# A28140 RRID:SCR_020240 | Section 'Quantitative (q)RT-PCR' |
| Other | Light Microscope (brightfield images) | Carl Zeiss LLC | Axio Observer, Inverted; 491917-0001-000 | *Figure 2—figure supplement 1* |
| Other | Spark 20 M Multimode Microplate Reader | Tecan | | Section 'Permeability of lung monolayer using FITC-dextran' |
| Other | Guava easyCyte Benchtop Flow Cytometer | Millipore | Guava easyCyte 62L | Section 'The characterization of lung cell types using flow cytometry' |
| Software, algorithm | ImageJ | ImageJ | RRID:SCR_003070 | |
| Software, algorithm | GraphPad Prism | GraphPad Prism | RRID:SCR_002798 | |
| Software, algorithm | LAS AF Software | LAS AF Software | | |
| Software, algorithm | QuantStudio Design & Analysis Software | QuantStudio Design & Analysis Software | | |
| Software, algorithm | CIBERSORTx | CIBERSORTx | | |
| Software, algorithm | FlowJo | FlowJo V10, BD BioSciences | RRID:SCR_008520 | |
| Chemical compound, drug | Zinc formalin | Fisher Scientific | Cat# 23-313096 | |
| Chemical compound, drug | Xylene | VWR | Cat# XX0060-4 | |
| Chemical compound, drug | Hematoxylin | Sigma-Aldrich Inc | Cat# MHS1 | |
| Chemical compound, drug | Ethanol | Koptec | Cat# UN1170 | |
| Chemical compound, drug | Sodium citrate | Sigma-Aldrich | Cat# W302600 | |
| Chemical compound, drug | DAB (10×) | Thermo Fisher | Cat# 1855920 | (1:10) |
| Chemical compound, drug | Stable peroxidase substrate buffer (10×) | Thermo Fisher | Cat# 34062 | (1:10) |
| Chemical compound, drug | 3% hydrogen peroxide | Target | Cat# 245-07-3628 | |
| Chemical compound, drug | Horse serum | Vector Labs | Cat# 30022 | |
| Commercial assay or kit | HRP Horse Anti-Rabbit IgG Polymer Detection Kit | Vector Laboratories | Cat# MP-7401 | |
| Chemical compound, drug | Paraformaldehyde 16% Solution, EM Grade | Electron Microscopy Sciences | Cat# 15710 | |
| Chemical compound, drug | 100% methanol | Supelco | Cat# MX0485 | |
| Chemical compound, drug | Glycine | Fisher Scientific | Cat# BP381-5 | |
| Chemical compound, drug | Bovine serum albumin | Sigma-Aldrich | Cat# A9647-100G | |

*Continued on next page*

*Continued*

| Reagent type (species) or resource | Designation | Source or reference | Identifiers | Additional information |
|---|---|---|---|---|
| Chemical compound, drug | Triton-X 100 | Sigma-Aldrich | Cat# X100-500ML | |
| Chemical compound, drug | ProLong Glass | Invitrogen | Cat# P36984 | |
| Chemical compound, drug | Nail Polish (Rapid Dry) | Electron Microscopy Sciences | Cat# 72180 | |
| Chemical compound, drug | Gill Modified Hematoxylin (Solution II) | Millipore Sigma | Cat# 65066-85 | |
| Chemical compound, drug | *HistoGel* | Thermo Scientific | Cat# HG4000012 | |
| Chemical compound, drug | TrypLE Select | Thermo Scientific | Cat# 12563-011 | |
| Chemical compound, drug | Advanced DMEM/F-12 | Thermo Scientific | Cat# 12634-010 | |
| Chemical compound, drug | HEPES buffer | Life Technologies | Cat# 15630080 | |
| Chemical compound, drug | Glutamax | Thermo Scientific | Cat# 35050-061 | |
| Chemical compound, drug | Penicillin-streptomycin | Thermo Scientific | Cat# 15140-122 | |
| Chemical compound, drug | Collagenase type I | Thermo Scientific | Cat# 17100-017 | |
| Chemical compound, drug | Matrigel | Corning | Cat# 354234 | |
| Chemical compound, drug | B-27 | Thermo Scientific | Cat# 17504044 | |
| Chemical compound, drug | N-acetyl-L-cysteine | Sigma-Aldrich | Cat# A9165 | |
| Chemical compound, drug | Nicotinamide | Sigma-Aldrich | Cat# N0636 | |
| Chemical compound, drug | FGF-7 (KGF) | PeproTech | Cat# 100-19-50ug | |
| Chemical compound, drug | FGF10 | PeproTech | Cat# 100-26-50ug | |
| Chemical compound, drug | A-83-01 | Bio-Techne Sales Corp. | Cat# 2939/50 | |
| Chemical compound, drug | SB202190 | Sigma-Aldrich | Cat# S7067-25MG | |
| Chemical compound, drug | Y-27632 | R&D Systems | Cat# 1254/50 | |
| Chemical compound, drug | DPBS | Thermo Scientific | Cat# 14190-144 | |
| Chemical compound, drug | Ultrapure Water | Invitrogen | Cat# 10977-015 | |
| Chemical compound, drug | EDTA | Thermo Scientific | Cat# AM9260G | |
| Chemical compound, drug | Hydrocortisone | STEMCELL Technologies | Cat# 7925 | |

*Continued on next page*

*Continued*

| Reagent type (species) or resource | Designation | Source or reference | Identifiers | Additional information |
|---|---|---|---|---|
| Chemical compound, drug | Heparin | Sigma-Aldrich | Cat# H3149 | |
| Other | PneumaCult Ex-Plus Medium | STEMCELL Technologies | Cat# 5040 | Section'The preparation of lung organoid-derived monolayers' |
| Other | PneumaCult ALI Medium | STEMCELL Technologies | Cat# 5001 | Section'ALImodel of lung organoids' |
| Chemical compound, drug | Goat serum | Vector Laboratories | Cat# MP-7401 | |
| Chemical compound, drug | Fetal bovine serum | Sigma-Aldrich | Cat# F2442-500ML | |
| Chemical compound, drug | Animal Component-Free Cell Dissociation Kit | STEMCELL Technologies | Cat# 5426 | |
| Chemical compound, drug | Red Blood Cell Lysis Buffer | Invitrogen | Cat# 00-4333-57 | |
| Chemical compound, drug | Cell Recovery Solution | Corning | Cat# 354253 | |
| Chemical compound, drug | Sodium azide | Fisher Scientific | Cat# S227I-100 | |
| Chemical compound, drug | Cyto-Fast Fix/Perm Buffer Set | BioLegend | Cat# 426803 | |
| Chemical compound, drug | FITC-dextran | Sigma-Aldrich | Cat# FD10S | |
| Commercial assay or kit | Quick-RNA MicroPrep Kit | Zymo Research | Cat# R1051 | |
| Commercial assay or kit | Quick-RNA MiniPrep Kit | Zymo Research | Cat#R1054 | |
| Chemical compound, drug | Ethyl alcohol, pure | Sigma-Aldrich | Cat# E7023 | |
| Chemical compound, drug | TRI Reagent | Zymo Research | Cat# R2050-1-200 | |
| Sequence-based reagent | 2x SYBR Green qPCR Master Mix | Bimake | Cat# B21203 | |
| Sequence-based reagent | qScript cDNA SuperMix | Quanta Biosciences | Cat# 95048 | |
| Sequence-based reagent | Applied Biosystems TaqMan Fast Advanced Master Mix | Thermo Scientific | Cat# 4444557 | |
| Sequence-based reagent | 18S , Hs99999901_s1 | Thermo Scientific | Cat# 4331182 | |
| Sequence-based reagent | E_Sarbeco_F1 Forward Primer | IDT | Cat# 10006888 | |
| Sequence-based reagent | E_Sarbeco_R2 Reverse Primer | IDT | Cat# 10006890 | |
| Sequence-based reagent | E_Sarbeco_P1 Probe | IDT | Cat# 10006892 | |
| Other | 12-well Tissue Culture Plate | CytoOne | Cat# CC7682-7512 | Section'Isolation and culture of human whole lung-derived organoids' |
| Other | Transwell Inserts (6.5 mm, 0.4 μm pore size) | Corning | Cat# 3470 | Section'The preparation of lung organoid-derived monolayers' |

*Continued on next page*

*Continued*

| Reagent type (species) or resource | Designation | Source or reference | Identifiers | Additional information |
|---|---|---|---|---|
| Other | Microscope Cover Glass (#1 Thickness) 24 × 50 mm | VWR | Cat# 16004-098 | Section'Immunofluorescence' |
| Other | Microscope Cover Glass (#1 Thickness) 25 mm diameter | Chemglass Life Sciences | Cat# CLS-1760-025 | Section'Immunofluorescence' |
| Other | Millicell EZ Slide 8-Well Chamber | Millipore Sigma | Cat# PEZGS0816 | Section'Immunofluorescence' |
| Other | Trypan Blue Stain | Invitrogen | Cat# T10282 | (1:2) |
| Other | 70 µm Cell Strainer | Thermo Fisher Scientific | Cat# 22-363-548 | Section'The preparation of lung organoid-derived monolayers' |
| Other | 100 µm Cell Strainer | Corning | Cat# 352360 | Section'Isolation and culture of human whole lung-derived organoids' |
| Other | Noyes Spring Scissors – Angled | Fine Science Tools | Cat# 15013-12 | Section'Isolation and culture of human whole lung-derived organoids' |

## Detailed methods

### Collection of human lung specimens for organoid isolation

To generate adult healthy lung organoids, fresh biopsy bites were prospectively collected after surgical resection of the lung by the cardiothoracic surgeon. Before collection of the lung specimens, each tissue was sent to a gross anatomy room where a pathologist cataloged the area of focus, and the extra specimens were routed to the research lab in Human Transport Media (HTM, Advanced DMEM/F-12, 10 mM HEPES, 1× Glutamax, 1× penicillin-streptomycin, 5 µM Y-27632) for cell isolation. Deidentified lung tissues obtained during surgical resection, which were deemed excess by clinical pathologists, were collected using an approved human research protocol (IRB# 101590; PI: Thistlethwaite). Isolation and biobanking of organoids from these lung tissues were carried out using an approved human research protocol (IRB# 190105: PI Ghosh and Das) that covers human subject research at the UC San Diego HUMANOID Center of Research Excellence (CoRE). For all the deidentified human subjects, information, including age, gender, and previous history of the disease, was collected from the chart following the rules of HIPAA and described in *Table 3*.

A portion of the same lung tissue specimen was fixed in 10% zinc-formalin for at least 24 hr followed by submersion in 70% EtOH until embedding in FFPE blocks.

### Autopsy procedures for lung tissue collection from COVID-19-positive human subjects

The lung specimens from COVID-19-positive human subjects were collected through autopsy (the study was IRB exempt). All donations to this trial were obtained after telephone consent followed by written email confirmation by the next of kin/power of attorney per California state law (no in-person visitation could be allowed into our COVID-19 ICU during the pandemic). The team member followed the CDC guidelines for COVID19 and the autopsy procedures (*CAP, 2020*; *CDC, 2020*). Lung specimens were collected in 10% zinc-formalin and stored for 72 hr before processing for histology. Patient characteristics are listed in *Table 3*.

Autopsy #2 was a standard autopsy performed by anatomical pathology in the BSL3 autopsy suite. The patient expired and his family consented for autopsy. After 48 hr, the lungs were removed and immersion fixed whole in 10% formalin for 48 hr and then processed further. Lungs were only partially fixed at this time (about 50% fixed in thicker segments) and were sectioned into small 2–4 cm chunks and immersed in 10% formalin for further investigation.

Autopsy #4 and #5 were collected from rapid postmortem lung biopsies. The procedure was performed in the Jacobs Medical Center ICU (all of the ICU rooms have a pressure-negative environment, with air exhausted through HEPA filters [Biosafety Level 3 (BSL3)] for isolation of SARS-CoV-2 virus). Biopsies were performed 2–4 hr after patient expiration. The ventilator was shut off to reduce the aerosolization of viral particles at least 1 hr after the loss of pulse and before sample collection. Every team member had personal protective equipment in accordance with the university policies for

procedures on patients with COVID-19 (N95 mask+ surgical mask, hairnet, full face shield, surgical gowns, double surgical gloves, booties). Lung biopsies were obtained after L-thoracotomy in the fifth intercostal space by the cardiothoracic surgery team. Samples were taken from the left upper lobe (LUL) and left lower lobe (LLL) and then sectioned further.

## Isolation and culture of human whole lung-derived organoids

A previously published protocol was modified to isolate lung organoids from three human subjects (*Sachs et al., 2019*; *Zhou et al., 2018*). Briefly, normal human lung specimens were washed with PBS/4× penicillin-streptomycin and minced with surgical scissors. Tissue fragments were resuspended in 10 ml of wash buffer (Advanced DMEM/F-12, 10 mM HEPES, 1× Glutamax, 1× penicillin-streptomycin) containing 2 mg/ml Collagenase Type I (Thermo Fisher, USA) and incubated at 37 °C for approximately 1 hr. During incubation, tissue pieces were sheared every 10 min with a 10 ml serological pipette and examined under a light microscope to monitor the progress of digestion. When 80–100% of single cells were released from connective tissue, the digestion buffer was neutralized with 10 ml wash buffer with added 2% fetal bovine serum; the suspension was passed through a 100 µm cell strainer and centrifuged at 200 rcf. Remaining erythrocytes were lysed in 2 ml red blood cell lysis buffer (Invitrogen) at room temperature for 5 min, followed by the addition of 10 ml of wash buffer and centrifugation at 200 rcf. Cell pellets were resuspended in cold Matrigel (Corning, USA) and seeded in 25 µl droplets on a 12-well tissue culture plate. The plate was inverted and incubated at 37 °C for 10 min to allow complete polymerization of the Matrigel before the addition of 1 ml Lung Expansion Medium per well. Lung expansion media was prepared by modifying a media that was optimized previously for growing GI-organoids (50% conditioned media, prepared from L-WRN cells with Wnt3a, R-spondin, and Noggin, ATCC-CRL-3276) (*Sayed et al., 2020c*; *Ghosh et al., 2020*; *Sayed et al., 2020a*; *Sayed et al., 2020b*) with a proprietary cocktail from the HUMANOID CoRE containing B27, TGF-β receptor inhibitor, antioxidants, p38 MAPK inhibitor, FGF 7, FGF 10, and ROCK inhibitor. The lung expansion media was compared to alveolosphere media I (IMDM and F12 as the basal medium with B27, low concentration of KGF, monothioglycerol, GSK3 inhibitor, ascorbic acid, dexamethasone, IBMX, cAMP, and ROCK inhibitor) and II (F12 as the basal medium with added $CaCl_2$, B27, low concentration of KGF, GSK3 inhibitor, TGF-β receptor inhibitor dexamethasone, IBMX, cAMP, and ROCK inhibitor) modified from previously published literature (*Jacob et al., 2019*; *Yamamoto et al., 2017*). Neither alvelosphere media contain any added Wnt3a, R-spondin, and Noggin. The composition of these media was developed either by fundamentals of adult stem cell-derived mixed epithelial cellularity in other organs (like the GI tract [*Miyoshi and Stappenbeck, 2013*; *Sato et al., 2009*; *Sayed et al., 2020c*]) or rationalized based on published growth conditions for proximal and distal airway components (*Gotoh et al., 2014*; *Sachs et al., 2019*; *van der Vaart and Clevers, 2021*). Organoids were maintained in a humidified incubator at 37 °C/5 % $CO_2$, with a complete media change performed every 3 days. After the organoids reached confluency between 7 and 10 days, organoids were collected in PBS/0.5 mM EDTA and centrifuged at 200 rcf for 5 min. Organoids were dissociated in 1 ml trypLE Select (Gibco, USA) per well at 37 °C for 4–5 min and mechanically sheared. Wash buffer was added at a 1:5, trypLE to wash buffer ratio. The cell suspension was subsequently centrifuged, resuspended in Matrigel, and seeded at a 1:5 ratio. Lung organoids were biobanked and passage 3–8 cells were used for experiments. Subculture was performed every 7–10 days.

## The preparation of lung organoid-derived monolayers

Lung organoid-derived monolayers were prepared using a modified protocol of GI organoid-derived monolayers (*Sayed et al., 2020c*; *Ghosh et al., 2020*; *Sayed et al., 2020a*; *Sayed et al., 2020b*). Briefly, transwell inserts (6.5 mm diameter, 0.4 µm pore size, Corning) were coated in Matrigel diluted in cold PBS at a 1:40 ratio and incubated for 1 hr at room temperature. Confluent organoids were collected in PBS/EDTA on day 7 and dissociated into single cells in trypLE for 6–7 min at 37 °C. Following enzymatic digestion, the cell suspension was mechanically sheared through vigorous pipetting with a 1000 µl pipette and neutralized with wash buffer. The suspension was centrifuged, resuspended in PneumaCult Ex-Plus Medium (StemCell, Canada), and passed through a 70 µm cell strainer. The coating solution was aspirated, and cells were seeded onto the apical membrane at 1.8E5 cells per transwell with 200 µl PneumaCult Ex-Plus media. 700 µl of PneumaCult Ex-Plus was added to the

basal chamber. Cells were cultured over the course of 2–4 days. A media change of both the apical and basal chambers was performed every 24 hr.

## ALI model of lung organoids

Organoids were dissociated into single cells and expanded in T-75 flasks in PneumaCult Ex-Plus Medium until confluency was reached. Cells were dissociated in ACF Enzymatic Dissociation Solution (StemCell) for 6–7 min at 37 °C and neutralized in equal volume ACF Enzyme Inhibition Solution (Stem-Cell). Cells were seeded in the apical chamber of transwells at 3.3E4 cells per transwell in 200 µl of PneumaCult Ex-Plus Medium. 500 µl of PneumaCult Ex-Plus was added to the basal chamber. Media in both chambers was changed every other day until confluency was reached (~4 days). The media was completely removed from the apical chamber, and media in the basal chamber was replaced with ALI Maintenance Medium (StemCell). The media in the basal chamber was changed every 2 days. The apical chamber was washed with warm PBS every 5–7 days to remove accumulated mucus. Cells were cultured under ALI conditions for 21+ days until they completed differentiation into a pseudostratified mucociliary epithelium. To assess the integrity of the epithelial barrier, TEER was measured with an Epithelial Volt-Ohm Meter (Millicell, USA). The media was removed from the basal chamber, and wash media was added to both chambers. Cultures were equilibrated to 37 °C before TEER values were measured. Final values were expressed as $\Omega \cdot cm^2$ units and were calculated by multiplying the growth area of the membrane by the raw TEER value.

## The culture of primary airway epithelial cells and iPSC-derived alveolar epithelial cells

Primary NHBE cells were obtained from Lonza and grown according to instructions. NHBE cells were cultured in T25 cell culture tissue flasks with PneumaCult-Ex Plus media (StemCell). Cells were seeded at ~100,000 cells/T25 flask and incubated at 37 °C, 5% $CO_2$. Once cells reached 70–80% confluency, they were dissociated using 0.25% Trypsin in dissociation media and plated in 24-well transwells (Corning). Primary human bronchial epithelial cells (HBEpC) and small airway epithelial cells (HSAEpC) were obtained from Cell Applications Inc The HBEpC and HSAEpC were cultured in human bronchial/tracheal epithelial cell media and small airway epithelial cell media, respectively, following the instructions of Cell Application.

Human iPSC-derived alveolar epithelial type 2 cells (iHAEpC2) were obtained from Cell Applications Inc and cultured in growth media (i536K-05, Cell Applications Inc) according to the manufacturer's instructions. All the primary cells were used within early passages (5–6) to avoid any gradual disintegration of the airway epithelium with columnar epithelial structure and epithelial integrity.

## The infection with SARS-Cov2

Lung organoid-derived monolayers or primary airway epithelial cells either in 384-well plates or in transwells were washed twice with antibiotic-free lung wash media. 1E5 PFU of SARS-CoV-2 strain USA-WA1/2020 (BEI Resources NR-52281) in complete DMEM was added to the apical side of the transwell and allowed to incubate for 24, 48, 72, and 96 hr at 34 °C and 5% $CO_2$. After incubation, the media was removed from the basal side of the transwell. The apical side of the transwells was then washed twice with (antibiotic-free lung wash media) and then twice with PBS. TRIzol Reagent (Thermo Fisher 15596026) was added to the well and incubated at 34 °C and 5 % $CO_2$ for 10 min. The TRIzol Reagent was removed and stored at –80 °C for RNA analysis.

## RNA isolation

Organoids and monolayers used for lung cell-type studies were lysed using RNA lysis buffer followed by RNA extraction per Zymo Research Quick-RNA MicroPrep Kit instructions. Tissue samples and monolayers in SARS-CoV2 studies were lysed in TRI-Reagent and RNA was extracted using Zymo Research Direct-zol RNA Miniprep.

## Quantitative (q)RT-PCR

Organoid and monolayer cell-type gene expression was measured by qRT-PCR using 2x SYBR Green qPCR Master Mix. cDNA was amplified with gene-specific primer/probe set for the lung cell type markers and qScript cDNA SuperMix (5 ×). qRT-PCR was performed with the Applied Biosystems

QuantStudio 5 Real-Time PCR System. Cycling parameters were as follows: 95 °C for 20 s, followed by 40 cycles of 1 s at 95 °C and 20 s at 60 °C. All samples were assayed in triplicate and eukaryotic 18S ribosomal RNA was used as a reference.

| Cell types | Marker | Primer sequence |
|---|---|---|
| Basal cells | ITGA6, NGFR, TP63 | ITGA6 F 'CGAAACCAAGGTTCTGAGCCC' ITGA6 R 'CTTGGATCTCCACTGAGGCAGT' NGFR F' CCTCATCCCTGTCTATTGCTCC NGFR R' GTTGGCTCCTTGCTTGTTCTGC TP63 F' CAGGAAGACAGAGTGTGCTGGT TP63 R' AATTGGACGGCGGTTCATCCCT |
| Goblet | Muc5AC | Muc5AC F 'GGAACTGTGGGGACAGCTCTT' Muc5AC R 'GTCACATTCCTCAGCGAGGTC' |
| Cilia | FoxJ1 | FoxJ1 F 'ACTCGTATGCCACGCTCATCTG'' FoxJ1 R 'GAGACAGGTTGTGGCGGATTGA' |
| Club cell | SCGB1A1 | SCGB1A1 F 'CAAAAGCCCAGAGAAAGCATC' SCGB1A1 R 'CAGTTGGGGATCTTCAGCTTC' |
| Alveolar type 1 | AQP5, PDPN | AQP5 F 'TACGGTGTGGCACCGCTCAATG' AQP5 R 'AGTCAGTGGAGGCGAAGATGCA' PDPN F 'GTGCCGAAGATGATGTGGTGAC' PDPN R 'GGACTGTGCTTTCTGAAGTTGGC' |
| Alveolar type 2 | SFTPA1, SFTPC | SFTPA1 F 'CACCTGGAGAAATGCCATGTCC' SFTPA1 R 'AAGTCGTGGAGTGTGGCTTGGA' SFTPC F 'GTCCTCATCGTCGTGGTGATTG' SFTPC R 'AGAAGGTGGCAGTGGTAACCAG' |

## Assessment of SARS-CoV-2 infectivity test

Assessment of SARS-CoV-2 infectivity test was determined by qPCR using TaqMan assays and TaqMan Universal PCR Master Mix as done before (*Corman et al., 2020*; *Lamers et al., 2020*). cDNA was amplified with gene-specific primer/probe set for the E gene and QPCR was performed with the Applied Biosystems QuantStudio 3 Real-Time PCR System. The specific TaqMan primer/probe set for E gene are as follows: forward 5'-ACAGGTACGTTAATAGTTAATAGCGT-3' (IDT, Cat# 10006888); reverse 5'-ATATTGCAGCAGTACGCACACA-3'; probe 5'-FAM-ACACTAGCCATCCTTACTGCGCTTCG-BBQ-3' and 18S rRNA. Cycling parameters were as follows: 95 °C for 20 s, followed by 40 cycles of 1 s at 95 °C and 20 s at 60 °C. All samples were assayed in triplicate, and gene eukaryotic 18S ribosomal RNA was used as a reference.

## Immunofluorescence

Organoids and lung organoid-derived monolayers in plates or in 8-well chamber slides were fixed by either (i) 4% PFA at room temperature for 30 min and quenched with 30 mM glycine for 5 min, (ii) ice-cold 100 % methanol at –20 °C for 20 min, and (iii) ice-cold 100 % methanol on ice for 20 min. Subsequently, samples were permeabilized and blocked for 2 hr using an in-house blocking buffer (2 mg/ml BSA and 0.1 % Triton X-100 in PBS); as described previously (*Lopez-Sanchez et al., 2014*). Primary antibodies were diluted in blocking buffer and allowed to incubate overnight at 4 °C; secondary antibodies were diluted in blocking buffer and allowed to incubate for 2 hr in the dark. Antibody dilutions are listed in the key resources table. ProLong Glass was used as a mounting medium. #1 Thick Coverslips were applied to slides and sealed. Samples were stored at 4 °C until imaged. FFPE-embedded organoid and lung tissue sections underwent antigen retrieval as previously described in Materials and methods for immunohistochemistry staining. After antigen retrieval and cooling in DI water, samples were permeabilized and blocked in blocking buffer and treated as mentioned above for immunofluorescence. Images were acquired at room temperature with Leica TCS SPE confocal and with DMI4000 B microscope using the Leica LAS-AF Software. Images were taken with a 40× oil-immersion objective using 405-, 488-, 561 nm laser lines for excitation. z-stack images were acquired by successive z-slices of 1 μm in the desired confocal channels. Fields of view that were representative and/or of interest were determined by randomly imaging three different fields. z-slices of a z-stack were overlaid to create maximum intensity projection images; all images were processed using FIJI (ImageJ) software.

## Embedding of organoids in *HistoGel*

Organoids were seeded on a layer of Matrigel in 6-well plates and grown for 7–8 days. Once mature, organoids were fixed in 4% PFA at room temperature for 30 min and quenched with 30 mM glycine for 5 min. Organoids were gently washed with PBS and harvested using a cell scraper. Organoids were resuspended in PBS using wide-bore 1000 µl pipette tips. Organoids were stained using Gill's hematoxylin for 5 min for easier FFPE embedding and sectioning visualization. Hematoxylin-stained organoids were gently washed in PBS and centrifuged and excess hematoxylin was aspirated. Organoids were resuspended in 65 °C HistoGel and centrifuged at 65 °C for 5 min. HistoGel-embedded organoid pellets were allowed to cool to room temperature and stored in 70% ethanol at 4 °C until ready for FFPE embedding by LJI Histology Core. Successive FFPE-embedded organoid sections were cut at a 4 µm thickness and fixed on to microscope slides.

## Immunohistochemistry

For SARS CoV-nucleoprotein (np) antigen retrieval, slides were immersed in Tris-EDTA buffer (pH 9.0) and boiled for 10 min at 100 °C inside a pressure cooker. Endogenous peroxidase activity was blocked by incubation with 3% $H_2O_2$ for 10 min. To block nonspecific protein binding, 2.5 % goat serum was added. Tissues were then incubated with a rabbit SARS CoV-NP antibody (Sino Biological, see key resource table) for 1.5 hr at room temperature in a humidified chamber and then rinsed with TBS or PBS three times, for 5 min each. Sections were incubated with horse anti-rabbit IgG secondary antibodies for 30 min at room temperature and then washed with TBS or PBS three times for 5 min each. Sections were incubated with DAB and counterstained with hematoxylin for 30 s.

## Permeability of lung monolayer using FITC-dextran

Adult lung monolayers were grown for 2 days in PneumaCult Ex-Plus media on transwell inserts (6.5 mm diameter, 0.4 µm pore size, Corning). TEER was monitored with an Epithelial Volt-Ohm Meter (Millicell). On the second day of growth, FITC-dextran (10 kD) was added at a 1:50 dilution in lung wash media. The basolateral side of the insert was changed to lung wash media only. After 30 min of incubation with FITC-dextran, 50 µl of the basolateral supernatant was transferred to an opaque welled 96-well plate. Fluorescence was measured using a TECAN plate reader.

## The characterization of lung cell types using flow cytometry

Lung organoids were dissociated into single cells via trypLE digestion and strained with a 30 µm filter (Miltenyi Biotec, Germany). Approximately 2.5E5 cells for each sample were fixed and permeabilized at room temperature in Cyto-Fast Fix Perm buffer (BioLegend, USA) for 20 min. The samples were subsequently washed with Cyto-Fast Perm Wash solution (BioLegend) and incubated with lung epithelial cell-type markers for 30 min. Following primary antibody incubation, the samples were washed and incubated with propidium iodide (Invitrogen) and Alexa Flour 488 secondary antibodies (Invitrogen) for 30 min. Samples were resuspended in FACS buffer (PBS, 5 % FBS, 2 mM sodium azide). Flow cytometry was performed using Guava easyCyte Benchtop Flow Cytometer (Millipore) and data was analyzed using InCyte (version 3.3) and FlowJo X v10 software.

## RNA sequencing

RNA sequencing libraries were generated using the Illumina TruSeq Stranded Total RNA Library Prep Gold with TruSeq Unique Dual Indexes (Illumina, San Diego, CA). Samples were processed following the manufacturer's instructions, except modifying RNA shear time to 5 min. The resulting libraries were multiplexed and sequenced with 100 basepair (bp) paired-end (PE100) to a depth of approximately 25–40 million reads per sample on an Illumina NovaSeq 6000 by the Institute of Genomic Medicine (IGM) at the University of California San Diego. Samples were demultiplexed using bcl2fastq v2.20 Conversion Software (Illumina). RNAseq data was processed using kallisto (version 0.45.0) and human genome GRCh38 Ensembl version 94 annotation (Homo_sapiens GRCh38.94 chr_patch_hapl_scaff.gtf). Gene-level transcripts per million (TPM) values and gene annotations were computed using tximport and biomaRt R package. A custom Python script was used to organize the data and log reduced using log2(TPM +1). The raw data and processed data are deposited in Gene Expression Omnibus under accession nos. GSE157055 and GSE157057.

### Data collection and annotation

Publicly available COVID-19 gene expression databases were downloaded from the National Center for Biotechnology Information (NCBI) Gene Expression Omnibus website (GEO) (*Edgar et al., 2002*; *Barrett et al., 2005*; *Barrett et al., 2013*). If the dataset is not normalized, RMA (Robust Multichip Average) (*Irizarry et al., 2003a*; *Irizarry et al., 2003b*) is used for microarrays and TPM (*Li and Dewey, 2011*; *Pachter, 2011*) is used for RNA seq data for normalization. We used log2(TPM +1) to compute the final log-reduced expression values for RNA seq data. Accession numbers for these crowdsourced datasets are provided in the figures and article. All of the above datasets were processed using the Hegemon data analysis framework (*Dalerba et al., 2011*; *Dalerba et al., 2016*; *Volkmer et al., 2012*).

### Analysis of RNA seq datasets

DESeq2 (*Love et al., 2014*) was applied to uninfected and infected samples to identify up- and down-regulated genes. A gene signature score is computed using both the up- and downregulated genes that are used to order the sample. To compute the gene signature score, first, the genes present in this list were normalized according to a modified Z-score approach centered around StepMiner threshold (formula = (expr -SThr)/3*stddev). The normalized expression values for every probeset for all the genes were added or subtracted (depending on up and downregulated genes) together to create the final score. The samples were ordered based on the final gene signature score. The gene signature score is used to classify sample categories, and the performance of the multiclass classification is measured by ROC-AUC values. A color-coded bar plot is combined with a violin plot to visualize the gene signature-based classification. All statistical tests were performed using R version 3.2.3 (2015-12-10). Standard t-tests were performed using Python scipy.stats.ttest_ind package (version 0.19.0) with Welch's two-sample t-test (unpaired, unequal variance (equal_var = False), and unequal sample size) parameters. Multiple hypothesis correction was performed by adjusting p-values with statsmodels.stats.multitest.multipletests (fdr_bh: Benjamini/Hochberg principles). The results were independently validated with R statistical software (R version 3.6.1; 2019-07-05). Pathway analysis of gene lists was carried out via the Reactome database and algorithm (*Fabregat et al., 2018*). Reactome identifies signaling and metabolic molecules and organizes their relations into biological pathways and processes. Violin, swarm, and bubble plots were created using Python seaborn package version 0.10.1.

### Single-Cell RNA seq data analysis

Single-Cell RNA seq data from GSE145926 was downloaded from GEO in the HDF5 Feature Barcode Matrix Format. The filtered barcode data matrix was processed using Seurat v3 R package (*Stuart et al., 2019*) and Scanpy Python package (*Wolf et al., 2018*). Pseudo bulk analysis of GSE145926 data was performed by adding counts from the different cell subtypes and normalized using log2(CPM + 1). Epithelial cells were identified using SFTPA1, SFTPB, AGER, AQP4, SFTPC, SCGB3A2, KRT5, CYP2F1, CCDC153, and TPPP3 genes using SCINA algorithm (*Zhang et al., 2019*). Pseudo bulk datasets were prepared by adding counts from the selected cells and normalized using log(CPM + 1).

### Assessment of cell-type proportions

CIBERSORTx (https://cibersortx.stanford.edu/runcibersortx.php) was used for cell-type deconvolution of our dataset (which was normalized by CPM). As reference samples, we first used the single-cell RNA seq dataset (GSE132914) from Gene Expression Omnibus (GEO). Next, we analyzed the bulk RNA seq datasets for the identification of cell types of interest using relevant gene markers (see *Table 2*): AT1 cells (PDPN, AQP5, P2RX4, TIMP3, SERPINE1), AT2 cells (SFTPA1, SFTPB, SFTPC, SFTPD, SCGB1A1, ABCA3, LAMP3), BASAL cells (CD44, KRT5, KRT13, KRT14, CKAP4, NGFR, ITGA6), CLUB cells (SCGB1A1, SCGB3A2, SFTPA1, SFTPB, SFTPD, ITGA6, CYP2F1), GOBLET cells (CDX2, MUC5AC, MUC5B, TFF3), ciliated cells (ACTG2, TUBB4A, FOXA3, FOXJ1, SNTN), and generic lung lineage cells (GJA1, TTF1, EPCAM) were identified using SCINA algorithm. Then, normalized pseudo counts were obtained with the CPM normalization method. The cell-type signature matrix was derived from the single-cell RNA seq dataset, cell types, and gene markers of interest. It was constructed by taking an average from gene expression levels for each of the markers across each cell type.

## Statistical analysis

All experiments were repeated at least three times, and results were presented either as one representative experiment or as average ± SEM. Statistical significance between datasets with three or more experimental groups was determined using one-way ANOVA including a Tukey's test for multiple comparisons. For all tests, a p-value of 0.05 was used as the cutoff to determine significance (*$p < 0.05$, **$p < 0.01$, ***$p < 0.001$, and ****$p < 0.0001$). All experiments were repeated a least three times, and p-values are indicated in each figure. All statistical analyses were performed using GraphPad Prism 6.1. A part of the statistical tests was performed using R version 3.2.3 (2015-12-10). Standard t-tests were performed using Python scipy.stats.ttest_ind package (version 0.19.0).

## Acknowledgements

The authors thank Victor Pretorius, Rachel White, and Jen Bigbee (Department of Cardiothoracic Surgery, UC San Diego) who assisted with thoracotomies during rapid autopsies.

## Additional information

### Funding

| Funder | Grant reference number | Author |
|---|---|---|
| National Institute of Diabetes and Digestive and Kidney Diseases | 3R01DK107585-05S1 | Soumita Das |
| National Institute of Diabetes and Digestive and Kidney Diseases | 1R01DK107585-01A1 | Soumita Das |
| National Institute of Allergy and Infectious Diseases | R01-AI 155696 | Debashis Sahoo Pradipta Ghosh Soumita Das |
| National Institute of Allergy and Infectious Diseases | R01-AI141630 | Pradipta Ghosh |
| National Cancer Institute | CA100768 | Pradipta Ghosh |
| National Cancer Institute | CA160911 | Pradipta Ghosh |
| National Institute of General Medical Sciences | R01-GM138385 | Debashis Sahoo |
| National Heart, Lung, and Blood Institute | R01- HL32225 | Patricia A Thistlethwaite |
| University of California, San Diego | UCOP-R01RG3780 | Debashis Sahoo Pradipta Ghosh Soumita Das |
| University of California, San Diego | UCOP-R00RG2642 | Pradipta Ghosh Soumita Das |
| National Heart, Lung, and Blood Institute | R01-HL137052 | Laura E Crotty Alexander |
| Department of Veterans Affairs Merit Award | 1I01BX004767 | Laura E Crotty Alexander |

The funders had no role in study design, data collection and interpretation, or the decision to submit the work for publication.

### Author contributions

Courtney Tindle, Conceptualization, Conducted experiments on adult lung-derived 3D-organoids and 2D-monolayers, Data curation, Formal analysis, Investigation, Methodology, Writing – original draft; MacKenzie Fuller, Ayden Fonseca, Conceptualization, Data curation, Formal analysis, Investigation, Methodology, Writing – review and editing; Sahar Taheri, Data curation, Formal analysis,

Methodology, Project administration; Stella-Rita Ibeawuchi, Nathan Beutler, Gajanan Dattatray Katkar, Amanraj Claire, Vanessa Castillo, Data curation, Formal analysis, Investigation, Methodology; Moises Hernandez, Data curation, Methodology, Resources; Hana Russo, Jason Duran, Ann Tipps, Grace Lin, Thomas F Rogers, Methodology, Resources; Laura E Crotty Alexander, Patricia A Thistlethwaite, Ranajoy Chattopadhyay, Methodology, Resources, Writing – review and editing; Debashis Sahoo, Conceptualization, Funding acquisition, Investigation, Methodology, Project administration, Resources, Software, Supervision, Writing – original draft, Writing – review and editing; Pradipta Ghosh, Soumita Das, Conceptualization, Data curation, Formal analysis, Funding acquisition, Investigation, Methodology, Project administration, Resources, Supervision, Validation, Writing – original draft, Writing – review and editing

### Author ORCIDs
MacKenzie Fuller ⓘ http://orcid.org/0000-0001-6781-8710
Vanessa Castillo ⓘ http://orcid.org/0000-0002-4182-8846
Moises Hernandez ⓘ http://orcid.org/0000-0002-7651-2673
Laura E Crotty Alexander ⓘ http://orcid.org/0000-0002-5091-2660
Debashis Sahoo ⓘ http://orcid.org/0000-0003-2329-8228
Pradipta Ghosh ⓘ http://orcid.org/0000-0002-8917-3201
Soumita Das ⓘ http://orcid.org/0000-0003-3895-3643

### Ethics

Ethics Statement:The human subject research was performed following the approved protocol at University of California San Diego. Deidentified lung tissues obtained during surgical resection, that were deemed excess by clinical pathologists, were collected using an approved human research protocol (IRB# 101590; PI: Thistlethwaite). Isolation and biobanking of organoids from these lung tissues were carried out using an approved human research protocol (IRB# 190105: PI Ghosh and Das) that covers human subject research at the UC San Diego HUMANOID Center of Research Excellence (CoRE). For all the deidentified human subjects, information including age, gender, and previous history of the disease, was collected from the chart following the rules of HIPAA.

### Decision letter and Author response
Decision letter https://doi.org/10.7554/eLife.66417.sa1
Author response https://doi.org/10.7554/eLife.66417.sa2

## Additional files

### Supplementary files
• Transparent reporting form

### Data availability
Sequencing data have been deposited in GEO under accession codes GSE157055, and GSE157057.

The following dataset was generated:

| Author(s) | Year | Dataset title | Dataset URL | Database and Identifier |
|---|---|---|---|---|
| Sahoo D, Das S, Ghosh P | 2020 | Human lung organoid for modeling infection and disease conditions | https://www.ncbi.nlm.nih.gov/geo/query/acc.cgi?acc=GSE157057 | NCBI Gene Expression Omnibus, GSE157057 |

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
