## [Decision Letter]

**Acceptance summary:**

Comparing with previous systems, the new organoid culture that you describe can maintain both proximal and distal cell types in adult lungs, and this ratio appears to be relatively stable in long term cultures. These organoids can also be used to mimic infectivity and immune responses characteristic of Covid19.

**Decision letter after peer review:**

Thank you for submitting your article "Adult Stem Cell-derived Complete Lung Organoid Models Emulate Lung Disease in COVID-19" for consideration by *eLife*. Your article has been reviewed by 3 peer reviewers, including Milica Radisic as the Reviewing Editor and Reviewer #1, and the evaluation has been overseen by Jos van der Meer as the Senior Editor. The following individual involved in review of your submission has agreed to reveal their identity: Hans Clevers (Reviewer #2).

Essential revisions:

1) All reviewers agree that the cell phenotypes and stability of the organoids in culture should be better characterized with time in culture using a number of new markers that are listed in the reviewer reports, as well as additional methods such as immunostaining and flow. Reviewers have raised concerns that organoids are not stable during culture.

2) Claims should be tampered, in particular related to high throughput drug screening since only few drugs were tested.

3) Discrepancies with transcriptomic signatures of the infected human lungs need to be more carefully considered.

*Reviewer #1 (Recommendations for the authors):*

The manuscript by Tindle et al. describes generation of adult lung organoids (ALO) from human lung biopsies and their use to study the changes in gene expression as a result of SARS-CoV-2 infection. The main advantage of the use of organoids is the ability to generate many cell types that make up the lung. In this particular case the authors report the presence of AT1, AT2 cells, Basal cells, Goblet cells, Ciliated cells and Club cells. The authors were able to cultivate the cells at the air-liquid interface and to establish cultures of predominately proximal and predominately distal airway cells. The main finding is that proximal cells are more prone to viral infection, while distal cells are governing the exuberant inflammatory response, with both cells required for the exuberant response to occur. A useful information provided by the paper is the analysis gene signatures of various cellular models, in comparison to the infected human lung.

1. Although cellular complexity is notable compared to some other models, it is important to more precisely benchmark how this compares to the percentages of cells in the actual lung. Figure 3 shows percentages of cells based on RNA sequencing, however a more precise enumeration might be provided by flow cytometry provided that the cells can be accurately dissociated.

2. In general, organoid models lack functional readouts such as permeability or barrier function. Were authors able to establish and enumerate the differences in barrier function in the monolayer model in the transwell system?

3. It is slightly discouraging that the model captures only 7 upregulated genes of the 76 reported in covid19 patient lungs and no downregulated genes. It also upregulates 15 genes that are not reported in the patient lungs (Figure 5E). These discrepancies should be better discussed.

*Reviewer #2 (Recommendations for the authors):*

The manuscript "Adult Stem Cell-derived Complete Lung Organoid Models Emulate Lung Disease in COVID-19" by Das and colleagues introduces a new model system of airway epithelium derived from adult lung organoids (ALO) to be utilised for the study of COVID-19-related processes. In this manuscript two main novelties are claimed: the development of a new model system which represents both proximal as distal airway epithelium and a computationally acquired gene signature that identifies SARS-CoV-2-infected individuals. While interesting data are presented, the novelty claim is questionable and the data is not always convincing.

1. The manuscript claims a novel model system of distal and proximal airway epithelium. The authors however fail to discuss a recently published COVID-organoid study which reports similar observations. While their novelty claim could still hold in some areas, the authors do need to discuss the following manuscript (https://doi.org/10.15252/embj.2020105912).

2. Expression patterns in Figure 2B are hard to interpret when no tissue control is used. A positive control of lung tissue should be used to make valid conclusions.

3. The authors claim the presence of multiple cell types within single ALO. Data are not strong.

a. Co-staining of KRT5 and SFTPB (as shown in figure 2J) is difficult to explain, since these markers are expressed in two different cell types. The lower panel in the same figure does show two separate populations, but the images shown in the second panel do not.

b. While the authors claim the presence of proximal and distal cells in ALOs, the images show 100% of SFTPC+ cells or 100% KRT5+ cells or 100% Na/K-ATPase cells. This rather shows two different types of ALO within one culture, then mixed cell populations within a single ALO.

c. The authors claim the presence of ciliated cells by staining of acetylated α tubulin in figure 2K-J and figure 3E. The images shown however are hardly showing specific ciliated cell staining and more importantly the cilia present on the cell membrane. The authors should include much higher quality images that clearly show cilia or other markers which are exclusive for ciliated cells.

d. In figure 3H, the authors show MUC5B positive cells in their monolayer cultures. These goblet cells seem to comprise quite a significant proportion of the culture. Their graph in 3B however shows no goblet cell contribution.

The claim that ALO consists of a mix of defined cell types is essentially based on deconvoluted bulk RNAseq data. Such a claim is not conclusive.

4. Authors claim that ALO maintain cell type ratios similar to lung tissue over several passages. Their data in figure 3B however show a drastic change in the composition of the ALOs. The shift from AT2 to AT1 is explained to occur when the ALO are cultured as submerged monolayers. However, the shift is already visible at later passages in 'standard' ALO culture. Moreover, when comparing cell types between passage 1 and 8, the composition changes dramatically. The cultures have lost basal cells, ALO1 has lost goblet cells, ALO2 has lost ciliated cells. In addition, ALO1 composition differs from ALO2 at passage 2 which doesn't support the statement that the cultures are stable/robust.

5. While the authors claim that proximal airway cultures are infected and responsible for maintaining virus replication, there are no virus entry marker-annotated cells in the composition graph of figure 1B.

6. Similar to point 3, the authors claim the development of the first combined organoid culture of proximal and distal cell types, as supported by deconvoluted bulk RNAseq. The hiPSC-derived AT2 cultures however also show the presence of both distal and proximal cell types.

7. Figure 3H shows a graph presenting the infectivity relative to peak. This axis makes comparison of infectivity impossible. The authors may want to include suppl Figure 5A-B as main figure and exclude 3H from the manuscript. Moreover, viral E-gene qPCR should be complemented by viral titer measurements to verify findings for live virus. The analysis of SARS-CoV-2 infectivity is very limited when only showing a few infected cells and viral E-gene graphs. Suppl Figure 5C also only shows one or two infected cells in all samples which does convince as proof of infectivity.

8. The authors provide data for a single drug to indicate the possibility of high-throughput screening of drugs for COVID-19 treatment. This does not say much about the throughput of the assay.

9. Figure 5C compares uninfected and infected monolayer cultures for genes that are identified in the patient cohort. While the authors claim comparable patterns between the cultures and the patients, the heatmap can not be directly read. The authors should include more details in the legends and combine the heatmaps in 4C and 5C for direct comparison. Currently, the colours represent raw z-scores which do not indicate transcript read numbers but relative differences of the expression of the indicated genes in the samples analysed. These numbers could differ extensively between organoids and patients.

10. In addition to point 9, the authors show that there is very limited overlap between significantly differentially expressed genes in monolayers and patients. While the authors believe this is mostly due to the lack of mesenchymal and immune components, this indicates that the monolayer itself is not important for the observed infection signature. This makes the claim that the monolayers represent infectivity in patients questionable.

11. Description in the discussion of growth factors supplied in the medium like FGF7 and FGF10 are not novel. Sachs et al. 2019 (https://doi.org/10.15252/embj.2018100300) already described these growth factors in the culture medium of airway organoids.

12. In the discussion, the authors describe the advantages of their model system including reproducibility, retainment of genetics and use for infection studies. These claims are however not novel since previous papers from multiple groups have reported similar organoid-based models for SARS-CoV-2 research.

*Reviewer #3 (Recommendations for the authors):*

In order to further improve this manuscript we recommend that the authors address the following points:

Figure 2.

1) In Figure 2H. the authors use NGFR as a generic stem cell marker. However, previous publications have shown that NGFR is a basal cell marker (Rock et al., 2009). We suggest that the authors to clarify what is the stem cell population they are attempting to mark here.

2) In all of Figure 2, but particularly, Figure 2K and 2J, we recommend the authors specify which passage number these organoids are from and show some simple quantification about different lineages to help the readers understand if different lineage remain stable between early and late passages.

3) None of the organoid immunofluorescence images in Figure 2 demonstrate that individual organoids contain both airway and alveolar lineages. In particular, SFTPB is presented as a alveolar type 2 cell marker, but it is well-established that this protein is also expressed in club cells (see: https://hlca.ds.czbiohub.org). It is not therefore useful for co-staining with basal cell markers to establish that individual organoids contain both airway and alveolar cell lineages. Mixed airway and alveolar organoid lineage is not yet convincingly demonstrated.

4) The Ac-Tub staining is not convincing as the cilia structures are not visible at the magnification presented.

5) The authors need to comment on the fate of the CC10/SFTPC double positive cell shown in Figure 2J bottom panel. Previous efforts to find SFTPC/CC10 co-expressing cells in human lungs in vivo have been unsuccessful (See: https://hlca.ds.czbiohub.org).

6) We recommend the authors also perform some simple polarity marker gene staining, such as ECAD and ZO-1, to confirm the apical-basal polarity of the organoids. This would help to justify the need of generating monolayer of epithelium for COVID infection in following experiments.

Figure 3.

1) The authors need to discuss in Figure 3B the extent of the variation of cell lineages observed in the self-renewing 3-D organoid cultures between passages 1 and 8. These changes suggest that the organoids do not have a stable cellular phenotype over time and cast serious doubt on their ability to reproducibly model lung infections, or other diseases. Similarly, in the submerged cultures do the 5 different bars represent the results of 5 different experiments using ALO1? Which passage was transferred to 2D culture? And can the authors comment on the reproducibility of their data?

2) For the same figure, the authors need to explain what are the cell populations called the 'general lung lineage (GLL)' and the 'viral entry marker (VEM)'. Does the VEM population include AT2, club cells and ciliated cells?

3) Figure 3E and Figure 3B (middle panel) show quite contradictory results regarding different proportion of cell lineages existing in the organoid system. Figure 3B middle panel seems to suggest a dominant AT1 and airway lineage, but few AT2 lineage cells. However, in Figure 3E, the authors seemed to suggest AT2 is also quite prevailing. Is this another example of variability?

4) We recommend the authors to also check AT1 and AT2 marker genes in the ALI culture, as a recent preprint has suggested that alveolar cell fates can also be maintained in ALI culture (Abo et al., 2020).

5) The authors need to address if the 'not sustained viral release and infectivity' of iAT2 cells was due to the virus quickly infecting, and killing, the AT2 cells within the first 48 hours, which could be another explanation for the kinetics observed in Figure 3H.

Figures 4-6

The approach in Figures 4-6 of characterizing the COVID-infected lungs at a transcriptional level, deriving transcriptional signatures and testing to what extent these are replicated in the various organoid models is innovative and highly commendable. The authors state that these data strongly suggest that the mixed organoids presented are a better culture model for COVID infection than previous organoid experiments. The spread of the data, e.g. in 6E, may also reflect the variability of this culture system. However, this reviewer is not sufficiently bioinformatically-literate to comment in detail on this aspect of the manuscript.

[Editors' note: further revisions were suggested prior to acceptance, as described below.]

Thank you for resubmitting your work entitled "Adult Stem Cell-derived Complete Lung Organoid Models Emulate Lung Disease in COVID-19" for further consideration by *eLife*. Your revised article has been evaluated by Jos van der Meer (Senior Editor) and a Reviewing Editor.

The manuscript has been improved but there are some remaining issues that need to be addressed, as outlined below:

The Reviewers found that your revised manuscript is improved, however immunostaining is still not found to be completely conclusive, for example immunostaining of ciliated cells and others would help.

We suggest that the authors should alter the text in response to the points raised in the second round of reviews to tamper the claims according to the suggestions of the Reviewers listed below.

*Reviewer #2 (Recommendations for the authors):*

The revisions on the manuscript "Adult Stem Cell-derived Complete Lung Organoid Models Emulate Lung Disease in COVID-19" by the group of Prof. Das have been extensive, as is the rebuttal letter.

Overall, the authors have answered most of the questions about COVID-19 in an adequate manner. However, major issues regarding the characterisation of the organoid model remain. In general, the authors still present contradicting data about the cell type composition of the model. This also casts indirect doubt on the COVID experiments.

Below, I go through the rebuttal letter point-by-point

Comment # 1

All reviewers agree that the cell phenotypes and stability of the organoids in culture should be better characterized with time in culture using a number of new markers that are listed in the reviewer reports, as well as additional methods such as immunostaining and flow. Reviewers have raised concerns that organoids are not stable during culture.

Response to rebuttal

The authors have included additional analyses to verify the stability of their model system. These additional analyses however are not overlapping with existing data. Discussed by method:

- The authors have used flow cytometry to identify cell compositions in the organoid models in early and late passages. While the authors indeed show (Figure 2S5) that the percentage of cell types based on key markers is stable in passages, the data itself is not convincing. The authors describe a model that consists of 96% basal cells, 60% goblet or secretory cells, 90% ciliated cells, 48% AT1 cells and 90% AT2 cells. While the authors already discuss that these percentages add up to more that 100%, the sum of the cell types is even 384% over 5 cell types. The authors claim this is due to overlap of marker genes between cell types and reference a manuscript. The manuscript however does not cover this overlap but actually shows that these markers can be used to separate the cell types. Apart from this, the same antibodies when used in immunostaining (Figure 3E-F) do not show these high percentages of cell positivity for the given markers. While in the FACS analysis the monolayers consist of 92% SFTPB+ cells, the immunostaining shows 10-20% positivity. Both methods are based on protein levels and should therefore show similar phenotypes. Secondly, the authors' claim of lowly expressed marker genes in all cell types this could be resolved by using a more stringent gate in the flow cytometer, only gating the real positives or by addition of a second marker in another channel. The first approach seems to already work for AQP5+ population in which there is a second population that is more positive and therefore potentially true AT1 cells. This result is also in line with the immunostaining in Figure 3E.

Anyway, the markers used are known to be highly specific to each of the cell types. Thus, this part of the manuscript remains very confusing.

– While the qRT-PCR data of the comparison between tissue and ALOs is convincing (Figure 2S3) and indeed underlines the representative nature of the model, the comparison between passages is slightly discouraging (Figure 2S4). While the authors state that cell type composition is stable during culture based on the qRT-PCR results, the data show some large differences between early and late passages. Loss of SCGB1A1 and increase in FOXJ1 in culture in all ALOs is visible as well as some varying levels of other markers in one or more lines does not comply with the stability stated in the text. The authors should still state this variation somewhere in the text.

– Immunostaining of ciliated cells and others would really help. The authors however have not included immunostainings of cell types in different passages to verify the qRT-PCR results. These immunostainings could be quantified to show cell compositions in the ALOs in 3D and 2D.

In all, the authors should use qRT-PCR to determine presence of all cell types, and combine this with immunostainings to determine cell compositions of the models. The authors still show the CIBERSORTx method which confuses the reader since it contradicts data shown in other figures. The authors explain in their rebuttal letter that the method was only used as corroborative, yet the data contradicts their own data.

Comment #2

Claims should be tampered, in particular related to high throughput (HTP) drug screening since only few drugs were tested.

Response to rebuttal

The authors explain in great depth that they did not imply that HTP were performed. This can however be inferred from the text even after changing the text as given in the rebuttal. The finding that the ALOs can be used in a 384-well format is important but the authors should add some more in-depth detailed description that this only generates a possibility for potential HTP screens instead of inferring HTP can be done on these models. The description of a "proof-of-concept for HTP" confuses. This term implies that the authors have used an initial HTP drug screen to validate. This can be changed to "initial protocol for building a platform on which HTP drug screens could be performed"?

Comment #3

Discrepancies with transcriptomic signatures of the infected human lungs need to be more carefully considered.

Response to rebuttal

The authors have included more clinical datasets and comparisons between other published model systems. It remains important to mention the discrepancies between clinical samples and the model. This is altered in the text sufficiently. This comment is therefore satisfactorily answered in the revised version of the manuscript.

Personal reviewer comments

Comment 1

The manuscript claims a novel model system of distal and proximal airway epithelium. The authors however fail to discuss a recently published COVID-organoid study which reports similar observations. While their novelty claim could still hold in some areas, the authors do need to discuss the recent manuscripts.

Response to rebuttal

The authors have extended their table including all the existing models with the recent published model systems. I agree that keeping up with COVID-19 model systems is hard in these times. The inclusion into the tables has, however, not led to the authors changing their claim of a novel model system of bronchiolar and alveolar potential. In line 102-104 the authors still write "…what is particularly noteworthy is that none recapitulate the heterogeneous epithelial cellularity of both proximal and distal airways…" which should be altered since Lamers et al. did exactly this. Moreover, in the rebuttal letter, the authors also mention that iPSC-derived AT2 cultures can contain both proximal and distal cell types.

The comparison with the currently published method of Lamers et al. 2021 is now in the manuscript and could be highlighted in the discussion or at the end of the paragraph named "Creation of a lung organoid model, complete with both proximal and distal airway epithelia" by adding a few lines comparing the cell types present in this model (ciliated and goblet) which was not shown in Lamers et al. 2021 (of at least this is convincingly demonstrated)

Comment 2.

Expression patterns in Figure 2B are hard to interpret when no tissue control is used. A positive control of lung tissue should be used to make valid conclusions.

Response to rebuttal

The authors have answered this comment by adding Figure 2S3

Comment 3.

The authors claim the presence of multiple cell types within single ALO. Data are not strong.

3a. Co-staining of KRT5 and SFTPB (as shown in figure 2J) is difficult to explain, since these markers are expressed in two different cell types. The lower panel in the same figure does show two separate populations, but the images shown in the second panel do not.

and

3b. While the authors claim the presence of proximal and distal cells in ALOs, the images show 100% of SFTPC+ cells or 100% KRT5+ cells or 100% Na/K-ATPase cells. This rather shows two different types of ALO within one culture, then mixed cell populations within a single ALO.

Response to rebuttal

By changing the images and the text, the authors made clear that the ALOs are either composed of multiple cell types or of a singular cell type. Therefore, this comment is sufficiently answered.

3c. The authors claim the presence of ciliated cells by staining of acetylated α tubulin in figure 2K-J and figure 3E. The images shown however are hardly showing specific ciliated cell staining and more importantly the cilia as present on the cell membrane.

Response to rebuttal

By addition of the higher resolution images, the authors answered this comment sufficiently.

3d. In figure 3H, the authors show MUC5B positive cells in their monolayer cultures. These goblet cells seem to comprise quite a significant proportion of the culture. Their graph in 3B however shows no goblet cell contribution. The claim that ALO consists of a mix of defined cell types is essentially based on deconvoluted bulk RNAseq data. Such a claim is not conclusive.

Response to rebuttal

This comment comes back to earlier described methods in the editor comment section. To shortly summarise, the CYBERSORTx data confuses the reader and is contradictory to the immunostainings, qRT-PCR and flow cytometry data.

Comment 4.

Authors claim that ALO maintain cell type ratios similar to lung tissue over several passages. Their data in figure 3B however show a drastic change in the composition of the ALOs. The shift from AT2 to AT1 is explained to occur when the ALO are cultured as submerged monolayers. However, the shift is already visible at later passages in 'standard' ALO culture. Moreover, when comparing cell types between passage 1 and 8, the composition changes dramatically. The cultures have lost basal cells, ALO1 has lost goblet cells, ALO2 has lost ciliated cells. In addition, ALO1 composition differs from ALO2 at passage 2 which doesn't support the statement that the cultures are stable/robust.

Response to rebuttal

See earlier comments about using CYBERSORTx and flow cytometry data.

Comment 5.

While the authors claim that proximal airway cultures are infected and responsible for maintaining virus replication, there are no virus entry marker-annotated cells in the composition graph of figure 1B.

Response to rebuttal

The comment was aimed at the proximal ALI culture cell composition graph in original figure 3B. This graph did not indicate a VEM population. Still this culture was infected by the virus, without showing a VEM population. The authors have removed the viral entry marker cells from their data representation. This was due to bioinformatic difficulty and VEM cells being a separate cell type in their analyses.

Comment 6

Similar to point 3, the authors claim the development of the first combined organoid culture of proximal and distal cell types, as supported by deconvoluted bulk RNAseq. The hiPSC-derived AT2 cultures however also show the presence of both distal and proximal cell types.

Response to rebuttal

In the rebuttal letter the authors mention the manuscripts describing the presence of proximal cell types in a AT2 differentiation protocol of iPSCs. This once more underlines that their statement of the first culture of proximal and distal cell types is incorrect.

Comment 7

Figure 3H shows a graph presenting the infectivity relative to peak. This axis makes comparison of infectivity impossible. The authors may want to include suppl Figure 5A-B as main figure and exclude 3H from the manuscript. Moreover, viral E-gene qPCR should be complemented by viral titer measurements to verify findings for live virus. The analysis of SARS-CoV-2 infectivity is very limited when only showing a few infected cells and viral E-gene graphs. Suppl Figure 5C also only shows one or two infected cells in all samples which does convince as proof of infectivity.

Response to rebuttal

The authors provide an extensive reasoning behind the use of the legend in figure 3H. The sustained infectivity the authors want to bring across is sufficiently depicted in figure 3H. Figure 3H might confuse the readers that the submerged culture are more heavily infected than ALI cultures. Moreover, this would allow the authors to already claim their permissiveness for infectivity in proximal airways without having to only look at 48-72h timeframe. "Permissive" in general refers to infection at an early stage which is shown in figure 3S3B and not figure 3H.

Comment 8

The authors provide data for a single drug to indicate the possibility of high-throughput screening of drugs for COVID-19 treatment. This does not say much about the throughput of the assay.

Response to rebuttal

This comment has been answered and once more commented in editor comments.

Comment 9

Figure 5C compares uninfected and infected monolayer cultures for genes that are identified in the patient cohort. While the authors claim comparable patterns between the cultures and the patients, the heatmap can not be directly read. The authors should include more details in the legends and combine the heatmaps in 4C and 5C for direct comparison. Currently, the colors represent raw z-scores which do not indicate transcript read numbers but relative differences of the expression of the indicated genes in the samples analysed. These numbers could differ extensively between organoids and patients.

Response to rebuttal

The authors have added some extra lines on the method used to generate these heatmaps. It now becomes clear that these datasets can not be directly compared within one heatmap.

Comment 10

In addition to point 9, the authors show that there is very limited overlap between significantly differentially expressed genes in monolayers and patients. While the authors believe this is mostly due to the lack of mesenchymal and immune components, this indicates that the monolayer itself is not important for the observed infection signature. This makes the claim that the monolayers represent infectivity in patients questionable.

Response to rebuttal

This has been discussed in the editor's section of the revision above

Comment 11.

Description in the discussion of growth factors supplied in the medium like FGF7 and FGF10 are not novel. Sachs et al. 2019 (https://doi.org/10.15252/embj.2018100300) already described these growth factors in the culture medium of airway organoids.

Response to rebuttal

The authors have sufficiently answered this comment.

Comment 12.

In the discussion, the authors describe the advantages of their model system including

reproducibility, retainment of genetics and use for infection studies. These claims are however not novel since previous papers from multiple groups have reported similar organoid-based models for SARS-CoV-2 research.

Response to rebuttal

With the comments above and the answers from the authors, this comment will be answered. The authors should better define their novelty by explaining the origin of the organoids model (adult stem cells) and their improvement in COVID-19 modelling.

*Reviewer #3 (Recommendations for the authors):*

The authors established a new 3D adult lung organoid system. Comparing with previous systems, the new organoid culture can maintain both proximal and distal cell types in adult lungs, and this ratio appears to be relatively stable in long term cultures. The 3D organoids can subsequently be dissociated and re-plated into 2D submerge culture, which exhibited promising features for modelling COVID19 infection. The authors further used bioinformatics analysis to show that the COVID19 infection using the 2D submerge culture was able to recapitulate both the infectivity and the immune responses in COVID patients.

In the revision processes, the authors have provided more supporting data to show the co-existence of proximal and distal lung cell types in the long-term culture and addressed most of the previous reviewers' comments. The few comments that we recommend the authors to further address are as follows:

1) The author added the Act-TUB staining for 2D culture in Figure 3, which looks convincing, however, for the 3D organoids, Act-TUB staining in Figure 2J still doesn't look like any cilia structure.

2) In Page 6 text line 189, the author mentioned a 'stem cell' population, which is TP63 positive. This is separate to the basal cells on line 188. Whereas in Figure 2-Supp4 the authors have acknowledged TP63 as a basal cell marker. We recommend the authors to make this consistent as basal cells which are well-established to be the airway stem cells (Rock et al., 2009).

3) The authors still make a strong claim about the co-existence of proximal (airway) and distal (alveolar) cell populations in a single 3D organoid (line 198). However, the staining shown in Figure 2 can only infer the existence of either proximal or distal cell populations in a single 3D organoid, given SFTPB is not a specific marker for alveolar lineage. Additionally, this strong claim wasn't adding much value to the manuscript as the authors only need to show proximal-distal cells exist in the overall population as 3D culture, given only the 2D submerged culture was used for COVID infection. We recommend the author not to mention this claim as even the revised data do not support it.

In addition, the same paragraph describes BASCs in the organoids. We cannot cite a reference to refute the existence of BASC cells in human lungs. However, in now more than 10 years of people looking for them a convincing demonstration that BASCs exist in human lungs is still missing. Human distal airways have a very different organisation to those of mice (respiratory bronchioles). BASC cells have not been found in them. We recommend that you do not highlight human BASCs in the organoids given the lack of credible evidence that they exist in vivo.

4) The authors need to clarify how the 5 replicates for submerged culture and 2 replicates for ALI culture were done in Figure 2B middle panel. Were they from the same ALO line or from different ALO lines? What was the passage number used here? These will help the readers to have an idea about how reproducible the system is.

---

## [Author Response]

Essential revisions:1) All reviewers agree that the cell phenotypes and stability of the organoids in culture should be better characterized with time in culture using a number of new markers that are listed in the reviewer reports, as well as additional methods such as immunostaining and flow. Reviewers have raised concerns that organoids are not stable during culture.

We also agree that cell phenotypes and stability of organoids in culture are two very claims that should be rigorously backed up with evidence. We recognize that stability in culture is especially important from a QC standpoint if this particular lung organoid model was going to be used in other studies, by us and others, to unravel disease mechanism(s) and/or for the preclinical drug screening.

Action(s) taken: To address these important issues, we have performed the following new experiments:

A) Immunostaining and Flow cytometry with early and late passaged lung organoids from three different patients (Figure 2—figure supplement 5). The complete lung organoid showed the presence of distal and proximal cell markers and there is not much difference in the percentage of positive cells in these cultures between an early and late passage (shown in the table in Figure 2-figure supplement 5D). For these studies, we have carefully dissociated the lung organoid to single cells and each antibody for the cell markers are optimized following the staining index and using proper isotype control antibody to rule out the non-specific binding.

The strategy for gating of the % positive cells and the corresponding isotype IgG controls are also shown in Figure 2-figure supplement 5.

One issue that is worth noting is that individual markers do not add up to 100% and show a higher percentage which is because it is well known that lung cell types share markers, as shown by others by FACS^1^. What we can report is that during immunostaining, we tested a serial dilution of antibodies to ensure that only minimal conc. of each antibody were used to detect the cell types (in addition to the isotype controls). Thus, the rigor in the analyses (alongside the IF data showcases earlier in the intact organoids) gives us confidence that what we see is believable.

B) qRT-PCR: We performed qPCR on the organoids and the lung tissue specimens from which they were derived to compare their cell type compositions (Figure 2-figure supplement 3). Our analyses showed that the tissue specimens and the organoids have a comparable amount of cell type markers.

We also performed qPCR studies on all three adult lung organoid lines (ALO1-3) from early (below passage 8) and late (above passage 8) passages and determined that the cell types remain stable in the culture. These data are included as a new figure (Figure 2-figure supplement 4).

C) Immunostaining: We added new staining data as requested by Reviewer #1 (higher resolution images of the Acetylated Tubulin-stained structures indicative of the presence of abundant ciliated cells (Figure 3F; maxprojected Z stack and orthogonal views).

The altered text reads as follows:

“Finally, using qRT-PCR of various cell-type markers as a measure, we confirmed that the ALO models overall recapitulated the cell type composition in the adult lung tissues from which they were derived (Figure 2—figure supplement 3) and retained such composition in later passages without significant notable changes in any particular cell type (Figure 2—figure supplement 4). The mixed proximal and distal cellular composition of the ALO models and their degree of stability during in vitro culture was confirmed also by flow cytometry (Figure 2—figure supplement 5).”

2) Claims should be tampered, in particular related to high throughput drug screening since only few drugs were tested.

It was never our intention to claim that the model’s usefulness was somehow validated for use in HTP drug screens. What we had intended to state was that we successfully optimized the use of ALO monolayers (cell #, plating conditions to achieve an intact monolayer, matrigel coating, timing of infection, cell viability, viral E gene detection and IF and gene expression analyses; etc) for infectivity in 384-well miniaturized format. This itself is an important step that sets us up for conducting HTP screens. It is unfortunate that the way it was written it may have come across that we have somehow done such screen already. We regret that.

Action(s) taken: We found 4 places within the text where the term ‘HTP’ was used. Here is how we have made changes that to tamper down the claim.

On Page 10:

Original sentence: “Findings also provide proof-of-concept that ALO monolayers may serve as effective models for use in HTP therapeutic screen”.

Changed to read: “Findings also provide proof-of-concept that ALO monolayers may be adapted in miniatured formats for use in 384-well plates for high-throughput (HTP) drug screens”.

On Page 14:

Original sentence: “Second, among all the established lung models so far, ours features 4 key properties that are desirable whenever disease models are being considered for their use in HTP modes for rapid screening of candidate therapeutics and vaccines……”

Changes made: None, because the use of HTP in this sentence was referring to desirable properties of any model.

On Page 14:

Original sentence: “Feasibility has also been established for scaling up for use in 384-well HTP assays.”

Changes made: “Feasibility has also been established for scaling up and optimizing the conditions for them to be used in miniaturized 384-well infectivity assays.”

On Page 15:

Original sentence: “Although we provide proof-of-concept studies in low throughput mode demonstrating the usefulness of the ALOs as human pre-clinical models for screening therapeutics in Phase ‘0’ trials, optimization for the same to be adapted in HTP mode was not attempted here.”

Changes made: None. This sentence was meant to accurately describe the study limitation, i.e., that HTP studies were not attempted here.

3) Discrepancies with transcriptomic signatures of the infected human lungs need to be more carefully considered.

In this comment, the Editor has summarized what appears to be a common criticism from both the Reviewers, i.e., the lower number of overlapping genes between human COVID-19-affected lung tissue and our lung organoid monolayers infected with SARS CoV2. To recap, this is specifically referring to the figure panel 5E in the original submission, in which we used a venn diagram to show that UP-regulated DEGs in our model overlap a ~one third of the time (7/22 genes) with UP-regulated DEGs in the infected patient lung, whereas there are no overlaps in DOWN-regulated DEGs.

We never addressed this apparent discrepancy in our original submission, and retrospectively, as regret such an error of omission.

Actions taken:

In this revised submission, we have performed several new analyses of 3 publicly available lung models of COVID-19 (ours, GSE160435, and GSE153218) against the following new patient-derived COVID19 datasets:

1) GSE151764- post-mortem COVID-19 and normal lung tissues

2) GSE156063- upper airway samples from patients with COVID-19

3) GSE145926- epi -- bronchoalveolar lavage fluid (BALF) cells from patients with varying degrees of COVID-19 severity

4) GSE157526-tracheal-bronchial cells infected with SARS-Cov2.

(a new Table of datasets (Table 8) has been included for the convenience of the reviewers and readers)

What did we do?: We set out to compare the DEGs from each of the 3 models against patient-derived datasets. GSE160435 (PMID: 32637946-preprint) is a model in which differentiated air-liquid interface from 3D organoid cultures of the alveolar epithelium were infected with SARS-CoV2. GSE153218 (PMID: 33283287; EMBO J) is 3D lung organoids derived from fetal (16-17 wk) lung bud tips.

What did we find?: As displayed in Figure 5—figure supplement 3:

– Our model: showed 22-54% overlaps in UP-regulated DEGs, and no overlaps with DOWN-regulated DEGs. (Panel A)

– GSE160435: showed 10-25% overlaps in UP-regulated DEGs, and no overlaps with DOWN-regulated DEGs. (Panel B)

– Our model and GSE160435 had a 50% overlap among the UP-regulated DEGs (Panel C)

– GSE153218: showed only 3 DEGs even for p-adj=0.5; lfc=0.0 cut-off values. Out of 17347 with nonzero total read count

adjusted p-value < 0.5

LFC > 0 (up) : 1, 0.0058% == CYP4A11__chr1

LFC < 0 (down) : 2, 0.012% == SARS-CoV-2, SARS1-HKU-39849 outliers [1] : 0, 0% low counts [2] : 0, 0%

(mean count < 0)

In the absence of DEGs our ability compares this third model against human samples was impaired. We also carried out (as requested by Reviewer #1) further analyses on GSE153218, which we have showcased later in the rebuttal (see Response to Reviewer #2, Comment #1). Briefly, this third model did not show the telltale signatures of host immune response to viral infection and hence, in the absence of those signature and absence of significant DEGs, this third model was excluded from the analysis in Figure 5—figure supplement 3.

What do we make of these new findings? We believe that although the epithelial contributions to the host response are important, it alone cannot account for the complete host response because the response of the immune and non-immune stromal cells, and their crosstalk with the epitheliual are missing from these minimalistic single component models. Given that inflammation is propagated by forward feedback loops of multi-compartment crosstalk, we believe that the epithelial signatures induced in vitro are only partially capturing the response that is likely to exist in vivo. Regardless of the missing components, what appears to be the case is that we have a model that recapitulates a one quarter to one half of the UP-regulated genes in COVID-19 despite cohort heterogeneity.

How have we modified the text to reflect these analyses?

In this version of the manuscript we have edited two sections.

On page 11: Results and Discussion section:

“Next, we analyzed the datasets from our ALO monolayers for differentially expressed genes (DEGs) when challenged with SARS-COV-2 (Figure 5A-B). Genes and pathways upregulated in the infected lung organoid-derived monolayer models (Figure 5—figure supplement 1-2) overlapped significantly with those that were upregulated in the COVID-19 lung signature (compare Figure 4C-D with 5C-D, Table 6-7). We observed only a partial overlap (ranging from ~22-55% across various human datasets; Figure 5—figure supplement 3) in upregulated genes and no overlaps among downregulated genes between model and disease (COVID-19) (Figure 5E). Because the degree of overlap was even lesser (ranging from ~10-25% across various human datasets; Figure 5—figure supplement 3) in the case of another publicly released model (GSE160435)2, these discrepancies between model and the actual disease likely reflects the missing stromal and immune components in our organoid monolayers.”

On page 15: Study limitations.

“Limitations of the study

Our adult stem-cell-derived lung organoids, complete with all epithelial cell types, can model COVID-19, but still remains a simplified/rudimentary version compared to the adult human organ. For instance, although the epithelial contributions to the host response are important, it alone cannot account for the response of the immune cells and of the non-immune stromal cells, and their crosstalk with the epithelium. Given that epithelial inflammation and damage is propagated by vicious forward-feedback loops of multicellular crosstalk, it is entirely possible that the epithelial signatures induced in infected ALO-derived monolayers are also only a fraction of the actual epithelial response mounted in vivo. Regardless of the missing components, what appears to be the case is that we already have a model that recapitulates a ¼ th to ½ of the genes that are induced across diverse COVID-19 infected patient samples. This model can be further improved by the simultaneous addition of endothelial cells and immune cells to better understand the pathophysiologic basis for DAD, microangiopathy, and even organizing fibrosis with loss of lung capacity that has been observed in many patients3; these insights should be valuable to fight some of the long-term sequelae of COVID-19.”

Reviewer #1 (Recommendations for the authors):The manuscript by Tindle et al. describes generation of adult lung organoids (ALO) from human lung biopsies and their use to study the changes in gene expression as a result of SARS-CoV-2 infection. The main advantage of the use of organoids is the ability to generate many cell types that make up the lung. In this particular case the authors report the presence of AT1, AT2 cells, Basal cells, Goblet cells, Ciliated cells and Club cells. The authors were able to cultivate the cells at the air-liquid interface and to establish cultures of predominately proximal and predominately distal airway cells. The main finding is that proximal cells are more prone to viral infection, while distal cells are governing the exuberant inflammatory response, with both cells required for the exuberant response to occur. A useful information provided by the paper is the analysis gene signatures of various cellular models, in comparison to the infected human lung.

We appreciate the accurate recap of the seminal findings and that the reviewer felt the computational analysis of gene signatures between models and disease as an useful information.

1. Although cellular complexity is notable compared to some other models, it is important to more precisely benchmark how this compares to the percentages of cells in the actual lung. Figure 3 shows percentages of cells based on RNA sequencing, however a more precise enumeration might be provided by flow cytometry provided that the cells can be accurately dissociated.

This comment has two parts:

i) In the first part, the reviewer asks for a comparison between ALO models and the parent tissue of origin.

ii) In the second part, the reviewer asks for a better enumeration of cell types and confirmation of mixed cellularity in ALO models after dissociating the organoids and assessing by immunostaining and flow cytometry (if such careful dissociation is feasible). In Figure 3B of the original manuscript we used CIBERSORTX, a machine learning method that extends this framework and infer cell-type-specific gene expression profiles without physical cell isolation. This method depends on the feeding of the data with specific markers that used to designate the cell types. As the lung markers are shared between different cell populations, we expected this computational method will be more corroborative. Also it is well known that lung is complex model and cellular plasticity is a major features^4,5^. We agree with the reviewer that beyond the computational approach, experimental validation is needed to confirm the mixed cellularity of the organoids.

Actions taken:

Please see response #1 in the Essential revisions.

Briefly, we have carried out what was asked of us, and have added new findings (in a total of 3 figures, Figure 2—figure supplement 3, 4, and 5). These new data (described in detail above on Page #9) show 6 different cell types by qRT-PCR and immunostaining/FACS. For the convenience of the reviewer, the edited text in the revised manuscript is also presented in within the response #1 in this rebuttal document.

2. In general, organoid models lack functional readouts such as permeability or barrier function. Were authors able to establish and enumerate the differences in barrier function in the monolayer model in the transwell system?

We agree that this was an important point, one that we only addressed partially earlier. In the original submission, we had only shown Transepithelial Electrical Resistance (TEER) in ALI monolayers derived from ALOs (Figure S7 in the original submission; currently Figure 3—figure supplement 1E). Hence, epithelial barrier in transwells was not evaluated, which we have now rectified in this revised submission.

Actions taken:

Expts conducted: We have dissociated the organoids to single cells and added to transwells. Following differentiation, we have performed the functional readout of barrier function as measured by the Transepithelial Electrical Resistance (TEER) and the permeability using the FITC-Dextran (10 kD). We have added the new TEER data from two different lung monolayer models in Figure 3-figure supplement 1B and the permeability using FITC-dextran in the same monolayers in Figure 3-figure supplement 1C. The impact of LPS on the TEER was also studied (Figure 3- figure supplement 1K-L).

Key Findings:

– Our data with TEER and permeability correlate with each other, and both show that the ALO-derived submerged monolayers can form an epithelial barrier.

– But these submerged monolayers were leakier than ALI models derived from the same organoids (which was expected) and compared to previously published TEER (400-1000 ohm-cm^2^) of human bronchial epithelial cells (NHBE), as shown in the published literature^6^.

– Occludin was visualized in patches, despite intactness of the monolayer (as determined by Phalloidin), indicative of leaky areas. Figure 3—figure supplement 1F-G. We chose to stain for Occludin because it is an important regulator of tight junction stability and function, is under the transcriptional control of TTF1/NKX2.1^7^. While there are numerous types of Claudins in the lung epithelial cells (Claudins 1-3-4-5-78-10), Occludin, however, is a more shared and constant marker throughout the airway whose role is to stabilize claudins and regulate their turnover^8^.

Interpretation: The submerged ALO-derived monolayers were able to form an epithelial barrier; it is however, leaky. We believe that this leakiness is likely due to the dynamic changes in proximal-distal cell ratios as ALO3D organoids and differentiated into ALO-monolayers and the persistence of progenitors at various stages of differentiation to AT1 cells. Our findings are in keeping with prior work demonstrating that as columnar wedge-shaped progenitors flatten to become AT1 cells, apical tight junctions (TJ) are maintained whereas lateral junctions are lost^9^. As alveolar differentiation takes place in the submerged monolayers, we not only expect shifting cellular ratios, but also changes in the types of Claudins. For example, some Claudins make the barrier tighter (Claudin-18) and others than make the barrier leakier (Claudins-3/4)^10^, Some of the claudins are alv specific (Claudin -18) whereas others are present in the bronchial passage (Claudin-1) and some that are present throughout (Claudin-4/7). We speculate that claudin-3 being more abundant on AT2 than AT1 cells and increases the alveolar permeability, and the permeability of our ALO-derived submerged monolayers.

Changes made to the manuscript text: To succinctly described the findings, and yet demonstrate restraint in avoiding speculative statements and/or not destroy the flow of the manuscript or distract readers with extensive review of the literature, we have made the following edits in the manuscript:

The epithelial barrier was leakier, as determined by relatively lower trans-epithelial electrical resistance (TEER; Figure 3—figure supplement 1B) and the flux of FITC-dextran from apical to basolateral chambers (Figure 3—figure supplement 1C), and corroborated by morphological assessment by confocal immunofluorescence of localization of occludin, a bona-fide TJ marker. We chose occludin because it is a shared and constant marker throughout the airway that stabilizes claudins and regulates their turnover8 and plays an important role in maintaining the integrity of the lung epithelial barrier11. Junction-localized occludin was patchy in the monolayer, despite the fact that the monolayer was otherwise intact, as determined by phalloidin staining (Figure 3- figure supplement 1H-I). Our finding, that ALO 3D organoids differentiating into monolayers in submerged cultures (where alveolar differentiation and cell-flattening happens dynamically as progenitor cells give rise to AT_1/2_ cells) are leaky is in keeping with prior work demonstrating that the TJs are rapidly remodeled as alveolar cells mature9,10. By contrast, and as expected6, the ALI-monolayers formed a more effective epithelial barrier, as determined by TEER (Figure 3—figure supplement 1F) and appeared to be progressively hazier with time after air-lift, likely due to the accumulation of secreted mucin (Figure 3—figure supplement 1G).”

3. It is slightly discouraging that the model captures only 7 upregulated genes of the 76 reported in covid19 patient lungs and no downregulated genes. It also upregulates 15 genes that are not reported in the patient lungs (Figure 5E). These discrepancies should be better discussed.

We agree that this is an important point. Because this was a major “Essential Revisions” request from the Editors, to avoid duplication of text, we kindly refer the reviewer to our Response # 3 in the Essential revisons.

Reviewer #2 (Recommendations for the authors):The manuscript "Adult Stem Cell-derived Complete Lung Organoid Models Emulate Lung Disease in COVID-19" by Das and colleagues introduces a new model system of airway epithelium derived from adult lung organoids (ALO) to be utilised for the study of COVID-19-related processes. In this manuscript two main novelties are claimed: the development of a new model system which represents both proximal as distal airway epithelium and a computationally acquired gene signature that identifies SARS-CoV-2-infected individuals. While interesting data are presented, the novelty claim is questionable and the data is not always convincing.1. The manuscript claims a novel model system of distal and proximal airway epithelium. The authors however fail to discuss a recently published COVID-organoid study which reports similar observations. While their novelty claim could still hold in some areas, the authors do need to discuss the following manuscript (https://doi.org/10.15252/embj.2020105912).

Within just weeks of submission of our paper, we ourselves recognized that the Table 1 (which listed all existing models and compared ours against them) was incomplete, because a new paper describing new models emerged every almost every other week. Consequently, we had no way of discussing these newer publications/citing them in our paper. The manuscript in question^12^ (Lamers et al., EMBO J, 2021) is one of them because it was published after our work was submitted, and hence, we had missed citing it.

Actions taken:

In this revised version of our manuscript, we have now cited this work and also added it (and 3 other models, as requested by Reviewer #3) to an updated Table 1.

As for the major differences between ours and the model described by Lamers et al., EMBO J (2021)40:e105912 are the following:

i) In their system, culture conditions were established to support long-term self-renewal of multipotent *sox2*^+^*SOX9*+ lung bud tip progenitor cells, which in vivo differentiate into both airway and alveolar cells. These lung bud tip organoids (LBT) are from canalicular stage human fetal lungs 16–17 wk postconception weeks and two such lines were used to establish the model. By contrast, our lung organoid model is from the adult lung specimens collected from 3 different donors that represent both the genders, smokers and non-smokers as illustrated in Figure 2 Figure supplement 1A. These fetal vs adult tissue source of stem cells is a key distinguishing feature that we have listed in a revised Table 1.

ii) The bronchio-alveolar like model presented here shows a robust increase in infectious virus titers over time of ~ 5 logs, whereas other recently developed adult-derived 3D alveolar organoid models show a more limited increase in viral titers (~ 1–2 logs). Our model, consistent with other adult models, has a limited increase in viral titers.

iii) As mentioned in our Response # 3 to the “Essential revisions” of this rebuttal, we did not see significant DEGs induced/repressed in the brochoalveolar lung datasets infected or not with SARS-CoV-2. The raw metrics for this analysis is presented above. This is unusual given that all other publicly released datasets show gene expression changes that have a varying degree of overlaps with numerous patient-derived samples. We addressed this issue in great detail in our Response #3. To avoid duplication of entire text, we kindly refer the reviewer to that section.

iv) In the absence of DEGs, which impaired our ability to proceed with cross-comparison against human samples, we went ahead and conducted additional analyses to see if the fetal lung derived model that is proficient in viral infectivity and permissive to massive replication may do so because it lacks significant host immune response. In fact, that is exactly what we found. Analyses of the dataset showed that the feta lung organoid-derived bronchoalveolar monolayers failed to mount the host immune response. This is rather unusual, because this signature has been widely validated prospectively in all adult COVID-19 human samples that have emerged to date, i.e., a total of 727 samples (Sahoo et al., eBioMedicine, 2021)^13^. We have included these new analyses in Figure 6H and edited the legend to reflect this update.

It is important for us to clarify that in the same work (Lamers et al., EMBO J (2021) 40:e105912), small airway epithelial cells were infected (as positive controls). We analyzed these datasets and found that they showed an induction of the ViP signatures. This data is presented Author response image 1, and not included in the manuscript.

**Author response image 1. sa1fig1:** Publicly available RNA seq datasets (GSE153218) from Small Airway Epi (SAEp) monolayers12 infected or not with SARS-CoV-2 were analyzed for the ability of ViP signatures to classify infected (Inf) from uninfected (Uninf) samples. ROC AUC indicate the performance of a classification model using the ViP signatures. Unlike the brochoalveolar monolayers (see Figure 6H in the revised manuscript) derived from fetal lung organoids in the same work, SAEp monolayers successfully induced the ViP signatures because the signatures were induced in infected monolayers.

Interpretation: From these new analyses we conclude that the fetal lung organoids are more permissive to viral infection and replication, but do not mount the host immune response that is seen in COVID-19, which largely affects adults.

Changes in the revised manuscript: several changes were made. For the convenience of the reviewer we have copied and pasted the text (annotated with page #).

New Figure 6H and legend: New panel added showing the 166-gene ViP signature and 20-gene sViP signature based classification of infected vs uninfected samples.

On Page #12-13:

“Our lung models showed that both the 166- and 20-gene ViP signatures were induced significantly in the submerged ALO-derived monolayers that had distal differentiation (Figure 6E; left), but not in the proximal-mimic ALI model (Figure 6E; right). Neither signatures were induced in monolayers of small airway epithelial cells (Figure 6F) or hiPSC-derived AT2 cells (Figure 6G). Finally, we analyzed a recently published lung organoid model that supports robust SARS-CoV-2 infection; this model was generated using multipotent sox2^+^SOX9+ lung bud tip (LBT) progenitor cells that were isolated from the canalicular stage of human fetal lungs (~16–17 wk post-conception)12. Despite mixed cellularity (proximal and distal), this fetal lung organoid model failed to induce the ViP signatures (Fig 6H). These findings indicate that despite having mixed cellular composition and the added advantage of being permissive to robust viral replication (achieving ~ 5 log-fold increase in titers), the model lacks the signature host response that is seen in all human samples of COVID-1913.”

On Page #15:

“Third, the value of the ALO models is further enhanced due to the availability of companion readouts/ biomarkers (e.g., ViP signatures in the case of respiratory viral pandemics, or monitoring the E gene, or viral shedding, etc.) that can rapidly and objectively vet treatment efficacy based on set therapeutic goals. Of these readouts, the host response, as assessed by ViP signatures, is a key vantage point because an overzealous host response is what is known to cause fatality. Recently, a systematic review of the existing pre-clinical animal models revealed that most of the animal models of COVID-19 recapitulated mild patterns of human COVID-19; no severe illness associated with mortality was observed, suggesting a wide gap between COVID-19 in humans 3 and animal models 14. It is noteworthy that alternative models that effectively support viral replication, such as the proximal airway epithelium or iPSC-derived AT2 cells (analyzed in this work) or a fetal lung bud tipderived organoid model recently described by others12, do not recapitulate the host response in COVID-19. The model revealed here, in conjunction with the ViP signatures described earlier 15, could serve as a pre-clinical model with companion diagnostics to identify drugs that target both the viral and host response in pandemics.”

In conclusion, we hope that we have discussed the two models accurately and presented concrete evidence to show how the adult lung organoid model described here is novel or different from the fetal model described by Lamers et al. EMBO J, 2021.

2. Expression patterns in Figure 2B are hard to interpret when no tissue control is used. A positive control of lung tissue should be used to make valid conclusions.

We agree.

Actions taken: In this revised submission we performed qPCR on the organoids and the lung tissue specimens from which they were derived to compare their cell type compositions (Figure 2-figure supplement 3). Our analyses showed that the tissue specimens and the organoids have a comparable amount of cell type markers.

3. The authors claim the presence of multiple cell types within single ALO. Data are not strong.a. Co-staining of KRT5 and SFTPB (as shown in figure 2J) is difficult to explain, since these markers are expressed in two different cell types. The lower panel in the same figure does show two separate populations, but the images shown in the second panel do not.

We agree that the way the images are presented, it can be confusing. For the convenience of the reviewer(s) and the editors, we have pasted the original figure in question within this rebuttal (please see Author response image 2). While the lower panel was meant to show more cell type specific organization within a single organoid structure, the upper panel was meant to show that the cell types mixed/interleaved with each other. We agree that although there are distinct red and green nonoverlapping cell types, in some areas (due to thickness of the cut and max projected zstacks), the merged panel shows ‘yellow’ pixels that make it look like KRT5 and SFTPB are overlapping with each other. The second interpretation can be a very unfortunate conclusion because it raises concerns about antibody specificity, controls, technical rigor, and overall concerns regarding one of the central claims (i.e., mixed cellularity of the organoids).

Within this rebuttal we have provided additional examples (Author response image 3), one of which was used to replace the upper panel:

**Author response image 3. sa3fig3:** 

b. While the authors claim the presence of proximal and distal cells in ALOs, the images show 100% of SFTPC+ cells or 100% KRT5+ cells or 100% Na/K-ATPase cells. This rather shows two different types of ALO within one culture, then mixed cell populations within a single ALO.

This is indeed an important point, which we addressed, but incompletely. To be clear, we never intended to claim that each organoid structure in each ALO line has all cell components. That was simply not our intention, and if that is how it came across, we regret the description presented in the original submission. We simply wanted to state that both proximal and distal airway components are present in the same line of organoids in culture; at times the structures were heterogeneous, at times they were homogeneous in their cell composition. For example, in this particular field of SFTPB/KRT5 stained structures, we see a single large organoid with heterogeneous cellularity, and two smaller structures that are almost exclusively and homogeneously comprised of one or the other (arrows in Author response image 4).

**Author response image 4. sa4fig4:** 

Figure 2J**:** the example showcased here was usedOn page #7: The text was updated to explicitly discuss/clarify this issue. For the convenience of the reviewer we have copied and pasted the text.

“The presence of all cell types was also confirmed by assessing protein expression of various cell types within organoids grown in 3D cultures. Two different approaches were used—(i) slices cut from FFPE cell blocks of HistoGel-embedded ALO lines (Figure 2I-J) or (ii) ALO lines grown in 8-well chamber slides were fixed in matrigel (Figure 2K), stained, and assessed by confocal microscopy. Such staining not only confirmed the presence of all cell types in each ALO line but also demonstrated the presence of more than one cell type (i.e., mixed cellularity) of proximal (basal-KRT5) and distal (AT1/AT2 markers) within the same organoid structure. For example, AT2 and basal cells, marked by SFTPB and KRT5, respectively, were found in the same 3D-structure (Figure 2J, interrupted curved lines). Similarly, ciliated cells and goblet cells stained by Ac-Tub and Muc5, respectively, were found to coexist within the same structure (Figure 2J, interrupted box; Figure 2K, arrow). Besides the organoids with heterogeneous makeup, each ALO line also showed homotypic organoid structures that were relatively enriched in one cell type (Figure 2J, arrowheads pointing to two adjacent structures that are either KRT5- or SFTPB-positive). Regardless of their homotypic or heterotypic cellular organization into 3D-structures, the presence of mixed cellularity was documented in all three ALO lines (see multiple additional examples in Figure 2—figure supplement 2I).”

c. The authors claim the presence of ciliated cells by staining of acetylated α tubulin in figure 2K-J and figure 3E. The images shown however are hardly showing specific ciliated cell staining and more importantly the cilia present on the cell membrane. The authors should include much higher quality images that clearly show cilia or other markers which are exclusive for ciliated cells.

We agree that it is incredibly difficult to appreciate finer structures when the original images are reduced to post-stamp sized panels to assemble into figures, and the figures had to be reduced in size for uploading.

Actions taken: We have performed new experiments and images of higher quality image of the cilia is now added in Figure 3F.

d. In figure 3H, the authors show MUC5B positive cells in their monolayer cultures. These goblet cells seem to comprise quite a significant proportion of the culture. Their graph in 3B however shows no goblet cell contribution.The claim that ALO consists of a mix of defined cell types is essentially based on deconvoluted bulk RNAseq data. Such a claim is not conclusive.

The reviewer brings up an important point that there is an apparent discrepancy between Figure 3B (which is RNA seq-based data) and Figure 3E (we believe that the reviewer meant 3E, because 3H is infectivity graphs), which is immunofluorescence staining. While the RNA Seq data did not show much “goblet” cell fraction, the immunostaining approach showed significant amount. This is an unfortunate outcome of trying to provide multiple lines of evidence through diverse approaches, but not clarifying what might be the strengths and weaknesses of each approach.

Limitations of RNA seq-based claims in Figure 3B**:** CIBERSORTx is a machine learning method that extends this framework and infer cell-type-specific gene expression profiles without physical cell isolation. This method depends on the feeding of the data with specific markers that are used to designate the cell types. Because many lung cell markers are shared between different cell types, we expect this computational method will be more of a corroborative evidence and not the overriding evidence. For example, CIBERSORTX has indicated that % cellularity in submerged has mostly AT_1/2_, and that corroborates with the IF panels.

Alternative explanations: Because 3B and 3E compare mRNA and protein markers of goblet cells, it is possible that the mRNA and protein abundance do not track each other well when it comes to those markers. Alternatively, it is possible that the expression level of the markers associated with goblet cells are low compared to the other cell types; therefore the goblet cell fraction is falsely underrepresented in the CYBERSORTx analyses.

Actions taken: We understand that rigorous and complementary approaches are better in both transcriptional and translational level. In this revised submission we added additional evidence (qRT-PCR, IF and FACS), as discussed in the Response #1 of “Essential Revisions” to determine different cellular proportion present in the lung organoid.

4. Authors claim that ALO maintain cell type ratios similar to lung tissue over several passages. Their data in figure 3B however show a drastic change in the composition of the ALOs. The shift from AT2 to AT1 is explained to occur when the ALO are cultured as submerged monolayers. However, the shift is already visible at later passages in 'standard' ALO culture. Moreover, when comparing cell types between passage 1 and 8, the composition changes dramatically. The cultures have lost basal cells, ALO1 has lost goblet cells, ALO2 has lost ciliated cells. In addition, ALO1 composition differs from ALO2 at passage 2 which doesn't support the statement that the cultures are stable/robust.

We agree. There are two things worth mentioning in this regard. In the original submission, viral entry markers (VEMs; e.g., ACE2, TMPRSS2, etc) were used as a part of the CYBERSPRTx analysis, which has been shown to be present on many cell types. This caused the cellular proportion analysis by CYBERSORTx less useful or interpretable. Second, as we have highlighted above, CYBERSORTx is imperfect; it was meant to be used as a corroborating evidence, but not the evidence.

Actions taken: The reason is, as the reviewers have themselves pointed out, this is a poor-man’s way to obtain cellularity information in the bulk transcriptome. The evidence that there is mixed cellularity in each line is drawn from – IF and qPCR studies. In this revised submission, we have now provided additional evidence about the nature and extent of drift for ALO 1-2-3 from early passages (1-8) and late passages (above 8) in Figure 2-figure supplement 3. We also compared the organoids with the tissue specimens to compare the cell types (Figure 2-figure supplement 3).

5. While the authors claim that proximal airway cultures are infected and responsible for maintaining virus replication, there are no virus entry marker-annotated cells in the composition graph of figure 1B.

We are unclear on what the reviewer is referring to in this comment and what he asks to see as part of revisions. Figure 1A is a box plot of the abundance of ACE2 (the viral entry marker) among various lung epithelial cell types. Because AT2 cells had high ACE2 transcripts, we then went on to show in human lungs (Figure 1B) the presence of SFTPC+vs AT2 cells in healthy lungs (top) and SARS-CoV-2 laden AT2 cells in the lung tissue of a patient with fatal COVID-19. If by viral entry marker in 1B the reviewer asks to see ACE2, we did not do that because there have been far too many groups who have shown ACE2+ve AT2 cells.

Alternatively, it is possible that the reviewer meant to refer to Figure 3B, but by mistake typed Figure 1B. If that is so, he might be wondering why viral entry marker (VEM)-annotated cells were not seen in the CYBERSORTx graphs. We can clarify why. As shown in Table 2 (that accompanied Figure 3B and listed the cell type markers used in the RNA Seq analyses), we used many more markers to ‘gate’ AT2 and other cell types, all of which carry VEMs. Hence, the proportion of VEM-containing cell types was all accounted for within the % distribution of all other cell types.

Actions taken: During this revised submission, we have corrected an error that we made in our judgement in the initial submission. We gated the RNA se data with lung cell type markers, and also included VEM as another ‘cell type’, which, retrospectively was not the right thing to do. We have re-done the CYBERSORTx analysis and replaced Figure 3B after excluding VEMs. For the convenience of the reviewers, we present here the original and the new versions side by side.

6. Similar to point 3, the authors claim the development of the first combined organoid culture of proximal and distal cell types, as supported by deconvoluted bulk RNAseq. The hiPSC-derived AT2 cultures however also show the presence of both distal and proximal cell types.

The fact that iPSC-derived AT2 cells were found to have other cell types markers by CYBERSORTx is not unexpected because it is well known that iPSC-derived AT2 be differentiated to other cell types, including AT1 and ciliated and club cells^16-18^. and/or express other markers that are commonly found on clara/club cells^19^.

7. Figure 3H shows a graph presenting the infectivity relative to peak. This axis makes comparison of infectivity impossible. The authors may want to include suppl Figure 5A-B as main figure and exclude 3H from the manuscript. Moreover, viral E-gene qPCR should be complemented by viral titer measurements to verify findings for live virus. The analysis of SARS-CoV-2 infectivity is very limited when only showing a few infected cells and viral E-gene graphs. Suppl Figure 5C also only shows one or two infected cells in all samples which does convince as proof of infectivity.

This comment has several parts, all of which relate to the veracity of the proof of infectivity, which in turn reflects the utility of the model to serve as a platform for screening drugs. We agree that this is an important point.

In the original submission, we provided the following proofs of infectivity and appropriate host response to the same:

1) Direct visualization of infectivity and examples of cellular cytopathic changes, evidence of intracellular packaging of viral particles (two montages of IF images in Figure 3G and Figure 3—figure supplement 2A).

2) E gene amplification by qPCR, at various time points after infection: Figure 3H and Figure 3—figure supplement 2B-C.

3) Impact of directly acting a previously confirmed anti-viral agent with anti-SARS-CoV-2 activity: Figure 3I.

4) Recapitulation of host response (gene signature induction) upon SARS-CoV-2 infection, where most other models described to date, fail or underperform: Figures 4-6.

The reviewer asks us to remove Figure 3H from the manuscript. His rationale for that suggestion is that this figure does not allow one to compare infectivity. We agree.

The reviewer believes that the Figure 3—figure supplement 3B-C is instead much more informative for comparing infectivity. We Agree.

We chose to show the sustained nature of the It was not our intention to highlight the degree of infectivity, rather the sustained nature of infectivity because one of the major distinguishing features (and, hence, in our mind, a novelty) of our model is that it recapitulated the nature of the host response to infection that is observed in human airway/lung samples and in numerous cohorts of patients with COVID-19 (> 700 datasets, which were used to prospectively validate the ViP signatures).

As mentioned above (Response #1 to this reviewer), when we compared our adult lung organoid model against the fetal lung organoid model (Lamers et al., EMBO J. Mar 2021), modeling COVID-19 not just requires permissiveness to viral infection, but infectivity with proportionate host immune response.

Actions taken:

We would like to retain Figure 3H in the main figure because we believe that the sustained infectivity of the model is a major aspect of modeling COVID-19. If the reviewer feels very strongly, and insists we change it, we can reconsider. We have, however, modified the legend for Figure 3H and refer readers to Figure 3—figure supplement 3B-C for comparing the peak viral amplification across various models. Because *eLife* allows supplementary figures to be interleaved within the main figures, should this manuscript be accepted for publication, we believe that the readers will readily have access to all data, regardless of where we insert it.

8. The authors provide data for a single drug to indicate the possibility of high-throughput screening of drugs for COVID-19 treatment. This does not say much about the throughput of the assay.

We agree. It was never our intention to claim that the model’s usefulness was somehow validated for use in HTP drug screens. What we had intended to state was that we successfully optimized the use of ALO monolayers (cell #, plating conditions to achieve an intact monolayer, matrigel coating, timing of infection, cell viability, viral E gene detection and IF and gene expression analyses; etc) for infectivity in 384 well miniaturized format. This itself is an important step that sets us up for conducting HTP screens. It is unfortunate that the way it was written it may have come across that we have somehow done such screen already. We regret that.

Actions taken: There were a total of 4 places in the manuscript where we used the term HTP. As part of Response #2 to “Essential Revisions”, we have eliminated and/or clarified each sentence. To avoid duplication of text, we kindly refer the reviewer to refer to this section in our Response #2 to Editors.

9. Figure 5C compares uninfected and infected monolayer cultures for genes that are identified in the patient cohort. While the authors claim comparable patterns between the cultures and the patients, the heatmap can not be directly read. The authors should include more details in the legends and combine the heatmaps in 4C and 5C for direct comparison. Currently, the colours represent raw z-scores which do not indicate transcript read numbers but relative differences of the expression of the indicated genes in the samples analysed. These numbers could differ extensively between organoids and patients.

We found this comment/suggestion difficult to mitigate, and perhaps reflects our inability to clearly explain what was done and why. Figure 4 and Figure 5 are dedicated to cross-validation of the actual COVID19 disease versus various SARS-CoV-2-infected in vitro models of the same (including our newly developed ALO model). As is displayed through workflow schematics in Figure 4, we first extract DEGs from the actual human disease (lungs of healthy vs COVID-19 patients; 4A-C) and tests the conservation of those gene sets in other human datasets (4E) before moving on to testing their ability to classify infected vs uninfected in vitro models (4F-G). In Figure 5, we extract DEGs from the infected vs uninfected ALO-model (5A-C) prior to their use to classify numerous human diseased samples (5F-I).

Thus, Figure 4C and Figure 5C represent heatmaps of DEGs in two different datasets from different technological platforms (Ion Torrent and Illumina NovaSeq). Also, read numbers are not comparable across these two datasets because of different tissue types and different RNASeq processing pipelines ( e.g., two different genome builds were used as a reference for each dataset, our organoid Model used hg38 while the other one used hg19). Therefore, we cannot combine the heatmap of Figure 4C and 5C.

Actions taken: We have inserted an additional sentence on Page #10 in “results” section to clarify what was being done. For the convenience of the reviewer, we have copied and pasted that edited piece of text below and indicated with highlight the newly added sentence. We have also edited the legends for figure 4 and 5 to improve clarity. We hope the reviewer finds it a bit more accessible

“Next, we asked if the newly generated lung models accurately recapitulate the host immune response in COVID-19. To this end, we analyzed the infected ALO monolayers (both the submerged and ALI variants) as well as the airway epithelial (HSAEpC) and AT2 monolayers by RNA seq and compared them all against the transcriptome profile of lungs from deceased COVID-19 patients. We did this analysis in two steps of reciprocal comparisons: (i) First, the actual human disease-derived gene signature was assessed for its ability to distinguish infected from uninfected disease models (in Figure 4). (ii) Second, the ALO model-derived gene signature was assessed for its ability to distinguish healthy from diseased patient samples (in Figure 5).”

10. In addition to point 9, the authors show that there is very limited overlap between significantly differentially expressed genes in monolayers and patients. While the authors believe this is mostly due to the lack of mesenchymal and immune components, this indicates that the monolayer itself is not important for the observed infection signature. This makes the claim that the monolayers represent infectivity in patients questionable.

We agree that this is a very important point, raised also by other reviewers, and was one of the important “Essential Revisions” suggested by the Editors.

Action taken: We have now extensively addressed this very important issue with the addition of new analyses and numerous figure panels. To avoid duplication of large passage of text with the rebuttal, we kindly refer the reviewer to Response #3 to Editor’s list of “Essential Revisions”.

11. Description in the discussion of growth factors supplied in the medium like FGF7 and FGF10 are not novel. Sachs et al. 2019 (https://doi.org/10.15252/embj.2018100300) already described these growth factors in the culture medium of airway organoids.

We agree. It was never our intention to stake claims that our media composition is novel. We have cited the *Sachs et al. 2019 paper*. Our media is a modified version of the media mentioned here and all the specifics are already added in “Methods”.

12. In the discussion, the authors describe the advantages of their model system including reproducibility, retainment of genetics and use for infection studies. These claims are however not novel since previous papers from multiple groups have reported similar organoid-based models for SARS-CoV-2 research.

In our updated Table 1 of this revised submission, we have tried to cite everything that has been tried to date and released publicly, and analyzed any transcriptomic datasets that were publicly available for each of those models. The reviewer is right that there are several papers that describe the development of organoid models, either iPSC-derived or adult stem-cell derived. We highlight in the Table and in the Discussion section (which has been modified to reflect the same) that the current model is unique and novel due to two major points.

i) The presence of the major cell types of proximal and distal region of adult lung.

ii) The infection with SARS-CoV-2 induces the expected host immune response that is observed in the actual disease.

These features, and the fact that they can be adapted easily to miniaturized 384-well formats for infection and drug Rx make them promising models for semi-HTP drug screening. As showcased in our response to Editors’ general comments, others groups have already independently used the model and reproduced its ability to serve as disease model.

Reviewer #3 (Recommendations for the authors):In order to further improve this manuscript we recommend that the authors address the following points:Figure 2.1) In Figure 2H. the authors use NGFR as a generic stem cell marker. However, previous publications have shown that NGFR is a basal cell marker (Rock et al., 2009). We suggest that the authors to clarify what is the stem cell population they are attempting to mark here.

We understand the concern that NGFR is a generic stem cell marker and specially for human oral keratinocyte stem/progenitor cell/epidermal stem cell and reported in the human cornea and epidermis. Previous report in lung has shown that following cell damage by chemical agents (naphthalene) or viral infection there are rapid changes in the proliferation of the basal cells and these cells quickly regenerate the epithelium and restore barrier function. Some of the Club secretory cells can undergo reprogramming to become Krt5+ Trp63+ basal cells and can function as stem cells in vivo^26,27^.

Another report has stated that, a small subpopulation of Scgb1a1^+^ Club cells in the distal bronchioles can also express Sftpc^28^ (known as dual-positive, bronchioalveolar stem cells or BASCs) using factors made by lung endothelial cells^29,30^. BASCs have the potential to differentiate into both AEC2s and airway cells.

Actions taken: In this revised version of the manuscript, we have performed new experiments with our organoid model and compared the levels of expression of Scgb1a1, Sftpc, and Trp63/TP63, added in Figure 2—figure supplement 3-4.

2) In all of Figure 2, but particularly, Figure 2K and 2J, we recommend the authors specify which passage number these organoids are from and show some simple quantification about different lineages to help the readers understand if different lineage remain stable between early and late passages.

For Figure 2K and 2J we have used cells from passages #3-6.

But, we agree with the other two reviewers and the editors that the characterization of cell composition stability of ALOs from early to late passages is something that needed to be more formally assessed and addressed in the manuscript, beyond just clarifying which figure used what passage of organoids.

Actions taken: In this revised manuscript we have performed and added new experiments assessing early and late passages of ALO1-2-3 by qRT-PCR (for cell type gene expression) and by immunostaining followed by Flowcytometry (cell marker protein expression). To avoid duplication of large text passages within this rebuttal, we kindly refer this reviewer to our detailed response to the Response #1 to Editor’s list of Essential Reviews. We conclude that lung organoids from early passages # 3-8 are not significantly different from late passages #9-15.

3) None of the organoid immunofluorescence images in Figure 2 demonstrate that individual organoids contain both airway and alveolar lineages. In particular, SFTPB is presented as a alveolar type 2 cell marker, but it is well-established that this protein is also expressed in club cells (see: https://hlca.ds.czbiohub.org). It is not therefore useful for co-staining with basal cell markers to establish that individual organoids contain both airway and alveolar cell lineages. Mixed airway and alveolar organoid lineage is not yet convincingly demonstrated.

Whether or not each individual 3D organoid structure has mixed cell types is a claim that we agree that we are not in any position to make. Part of that is because, as the reviewer states, lung epithelial cells are notoriously known to share markers among cell types. There is no good way to tell.

What we can convincing state, and have added many supporting evidence in this revised submission, is that organoids are either homotypic or heterotypic in composition (see Figure 2J, second row from the top). We have addressed this particular issue in our Response #3a-b to Reviewer #2 of this rebuttal. We have also carried out cell type composition studies (qPCR and FACS) (Figure 2—figure supplement 3-5).

Regardless of whether the 3D structures assemble into homogeneous or heterogeneous cell types, what is clear is that each ALO line overall has both proximal and distal cell types. Thus, while we agree with the premise of the critique, but respectfully disagree with the last statement.

4) The Ac-Tub staining is not convincing as the cilia structures are not visible at the magnification presented.

Agree. New Figure 3F has been added with higher magnification.

5) The authors need to comment on the fate of the CC10/SFTPC double positive cell shown in Figure 2J bottom panel. Previous efforts to find SFTPC/CC10 co-expressing cells in human lungs in vivo have been unsuccessful (See: https://hlca.ds.czbiohub.org).

We thank the reviewer for pointing out this error of omission, in that we had not addressed what these structures are. We believe that the SFTPC/CC10 double stained structure represent multipotent stem cells termed bronchioalveolar stem cells (BASCs) which have been found to be located at the bronchioalveolar-duct junctions (BADJs)^31,32^. BASCs coexpress club cell maker secretoglobin 1a1 (Scgb1a1 or CC10) and AT2 cell maker surfactant protein C (Sftpc or SPC).

The reviewer points to a github CZI link that appears to be a single cell RNA Seq based lung cell atlas. In doing so, he/she is requesting that the failure to detect CC10/SFTPC double positive (for RNA expression) cells in single cell sequencing studies of the human lung be addressed in light of our organoids showing the double-stained (for protein) 3D lung organoid structures in 2J. Besides the obvious differences in comparing such distinct approaches and samples, it is important to note that single cell studies have significant limitations too (high ‘dropouts’ due to degrees of sparsity and technical limitations). As highlighted and summarized in this leading biotechnology article single cell seq studies are still evolving and must get better.

Actions taken: We have inserted a sentence in the “results” section on Page #7 about the double-stained structures. While we appreciate that the reviewer pointed out the scSeq studies, we decline to comment on the fact that they did not find such BASCs. We felt that going into such comparison of two techniques, looking at RNA vs protein would confuse and distract the readers (or give the notion that what we see is somehow contradictory) in an otherwise descriptive study.

6) We recommend the authors also perform some simple polarity marker gene staining, such as ECAD and ZO-1, to confirm the apical-basal polarity of the organoids. This would help to justify the need of generating monolayer of epithelium for COVID infection in following experiments.

We agree. Reviewer #1 also asked a similar question. In order to avoid duplicating large chunks of text, we kindly refer the reviewer to Response to Comment #2 from Reviewer #1. We detailed in our response the experiments conducted during revision, our findings and interpretations, our choice of markers and the actions taken (revision to text and figures).

Figure 3.1) The authors need to discuss in Figure 3B the extent of the variation of cell lineages observed in the self-renewing 3-D organoid cultures between passages 1 and 8. These changes suggest that the organoids do not have a stable cellular phenotype over time and cast serious doubt on their ability to reproducibly model lung infections, or other diseases. Similarly, in the submerged cultures do the 5 different bars represent the results of 5 different experiments using ALO1? Which passage was transferred to 2D culture? And can the authors comment on the reproducibility of their data?

We agree that stability of organoids in culture conditions through later passages is an important point. To answer this reviewer’s question directly, the 5 sequenced samples in Figure 3B represent ALO1-2, passages #3-6. 2D cultures were carried out at the time of manuscript submission from passages #3-8, but since then we have characterized passages #1-15.

Actions taken: In this revised submission, we have carried out numerous studies and added them to the manuscript to address this issue squarely through dedicated figures. More specifically, the stability of cellular composition in long term culture was assessed using two methods (qPCR and FACS) spanning early (1-8) and late (9-15) passages (Figure 2—figure supplement 4-5). We hope that this additional evidence helps mitigate any concerns this reviewer has about the drift of ALOs during long term culture.

2) For the same figure, the authors need to explain what are the cell populations called the 'general lung lineage (GLL)' and the 'viral entry marker (VEM)'. Does the VEM population include AT2, club cells and ciliated cells?

We understand reviewer's concern. This was a shared concern of at least one another reviewer (Reviewer #2). We agree that including VEM in the gating strategy was a bad idea because these are not cell types and were likely skewing the overall results because the fraction would include AT2 and club cells and other cells that have some of these VEMs.

Actions taken: In the current version we have simplified the gating strategy and added only the 6 cell types (basal, AT1, AT2, cilia, club cells, goblet cells), that is the focus of the current paper and removed the GLL and VEM markers.

3) Figure 3E and Figure 3B (middle panel) show quite contradictory results regarding different proportion of cell lineages existing in the organoid system. Figure 3B middle panel seems to suggest a dominant AT1 and airway lineage, but few AT2 lineage cells. However, in Figure 3E, the authors seemed to suggest AT2 is also quite prevailing. Is this another example of variability?

We recognize (based on the comments from other reviewers and this reviewer) that the CYBRSORTx analysis from the RNA seq has limitations; it is, afterall, a machine learning approach that produces data based on our knowhow of what lung cell markers are ‘fed’ into the application as a starting point. Therefore, CYBERSORTx is, at best, more of a corroborative evidence, but not ‘the’ evidence.

Actions taken: In this revised submission, we have carried out numerous other studies and added them to the manuscript to support with evidence the following 3 claims:

1) Multicellular composition of ALO lines (qPCR and FACS)- Figure 2—figure supplement 3-5.

2) Comparison of ALO lines against the adult lung tissue from which they originated (qPCR)- Figure 2—figure supplement 3.

3) Stability of cellular composition in long term culture (qPCR and FACS)-- Figure 2—figure supplement 4-5.

4) IF stained 3D organoids with evidence of heterogeneous cellular composition in single 3D structures: Figure 2J. Figure 2—figure supplement 2.

5) IF stained 2D monolayers with evidence of heterogeneous cellular composition in monolayers: Figure 3. Figure 3—figure supplement 2.

4) We recommend the authors to also check AT1 and AT2 marker genes in the ALI culture, as a recent preprint has suggested that alveolar cell fates can also be maintained in ALI culture (Abo et al., 2020).

We agree.

Actions taken: In the revised version of this manuscript, we have carried out additional staining on ALI monolayers and included a new panel Figure 3—figure supplement 1J. For the convenience of the reviewer, we have copied and pasted that panel below. The figure legend and the text has been accordingly modified citing this additional panel.

5) The authors need to address if the 'not sustained viral release and infectivity' of iAT2 cells was due to the virus quickly infecting, and killing, the AT2 cells within the first 48 hours, which could be another explanation for the kinetics observed in Figure 3H.

We agree that this is a plausible explanation, and is certainly consistent with the diffuse alveolar damage (DAD) that is pathognomonic of the COVID-19 affected acute lung injury, which leads to ARDS. We have included a sentence to entertain this possibility in the text.

Figures 4-6The approach in Figures 4-6 of characterizing the COVID-infected lungs at a transcriptional level, deriving transcriptional signatures and testing to what extent these are replicated in the various organoid models is innovative and highly commendable. The authors state that these data strongly suggest that the mixed organoids presented are a better culture model for COVID infection than previous organoid experiments. The spread of the data, e.g. in 6E, may also reflect the variability of this culture system. However, this reviewer is not sufficiently bioinformatically-literate to comment in detail on this aspect of the manuscript.

We thank the reviewer for appreciating the degree of diligence that we demonstrated in confirming if the disease model is accurately able to not just model infectivity, but the host immune response that is observed in the actual disease, and is believed to be the cause of fatality. Because we do not want readers to feel left out if they do not find themselves as familiar with computational approaches, we have modified the results section with the intent to explain what was done and why. We also added additional datasets (prospectively) of models released since the submission of this manuscript (Fig 6H, new) and made head to head comparisons of other models versus disease (Fig 5- Figure Supplement 3).

[Editors' note: further revisions were suggested prior to acceptance, as described below.]

The manuscript has been improved but there are some remaining issues that need to be addressed, as outlined below:The Reviewers found that your revised manuscript is improved, however immunostaining is still not found to be completely conclusive, for example immunostaining of ciliated cells and others would help.We suggest that the authors should alter the text in response to the points raised in the second round of reviews to tamper the claims according to the suggestions of the Reviewers listed below.

We are pleased to see that all three reviewers found that the revised manuscript is improved and that we have adequately addressed their major concerns. As for the two major points summarized by the Editors, we have addressed those in the following way:

i) “immunostaining of ciliated cells…is not completely conclusive”: This point refers to the patterns of ciliary structures. We have addressed this with edits to the text to explain why 2D submerged monolayers but not 3D structures had prominent apical ciliary structures, which were further augmented in the 2D-ALI model. These progressive changes in staining patterns are consistent with the fact that differentiation and apicobasal polarity is minimal in 3D structures that are yet to form lumen (Figure 2J-K), is intermediate in the submerged 2D model, and maximal in the 2D ALI model (Figure 3F). Revised text now calls this pattern to attention (on Page 9). A detailed response to this point is on Page 13 of this rebuttal document (Response to Comment 1 from Reviewer #3).

ii) “alter the text …..to tamper the claims”: We have now done that throughout the text. For the convenience of the editors and the reviewers, we have copied and pasted the relevant edited text within the body of this rebuttal.

Reviewer #2 (Recommendations for the authors):The revisions on the manuscript "Adult Stem Cell-derived Complete Lung Organoid Models Emulate Lung Disease in COVID-19" by the group of Prof. Das have been extensive, as is the rebuttal letter.Overall, the authors have answered most of the questions about COVID-19 in an adequate manner. However, major issues regarding the characterisation of the organoid model remain. In general, the authors still present contradicting data about the cell type composition of the model. This also casts indirect doubt on the COVID experiments.Below, I go through the rebuttal letter point-by-pointComment # 1All reviewers agree that the cell phenotypes and stability of the organoids in culture should be better characterized with time in culture using a number of new markers that are listed in the reviewer reports, as well as additional methods such as immunostaining and flow. Reviewers have raised concerns that organoids are not stable during culture.Response to rebuttalThe authors have included additional analyses to verify the stability of their model system. These additional analyses however are not overlapping with existing data. Discussed by method:– The authors have used flow cytometry to identify cell compositions in the organoid models in early and late passages. While the authors indeed show (Figure 2S5) that the percentage of cell types based on key markers is stable in passages, the data itself is not convincing. The authors describe a model that consists of 96% basal cells, 60% goblet or secretory cells, 90% ciliated cells, 48% AT1 cells and 90% AT2 cells. While the authors already discuss that these percentages add up to more that 100%, the sum of the cell types is even 384% over 5 cell types. The authors claim this is due to overlap of marker genes between cell types and reference a manuscript. The manuscript however does not cover this overlap but actually shows that these markers can be used to separate the cell types. Apart from this, the same antibodies when used in immunostaining (Figure 3E-F) do not show these high percentages of cell positivity for the given markers. While in the FACS analysis the monolayers consist of 92% SFTPB+ cells, the immunostaining shows 10-20% positivity. Both methods are based on protein levels and should therefore show similar phenotypes. Secondly, the authors' claim of lowly expressed marker genes in all cell types this could be resolved by using a more stringent gate in the flow cytometer, only gating the real positives or by addition of a second marker in another channel. The first approach seems to already work for AQP5+ population in which there is a second population that is more positive and therefore potentially true AT1 cells. This result is also in line with the immunostaining in Figure 3E.Anyway, the markers used are known to be highly specific to each of the cell types. Thus, this part of the manuscript remains very confusing.– While the qRT-PCR data of the comparison between tissue and ALOs is convincing (Figure 2S3) and indeed underlines the representative nature of the model, the comparison between passages is slightly discouraging (Figure 2S4). While the authors state that cell type composition is stable during culture based on the qRT-PCR results, the data show some large differences between early and late passages. Loss of SCGB1A1 and increase in FOXJ1 in culture in all ALOs is visible as well as some varying levels of other markers in one or more lines does not comply with the stability stated in the text. The authors should still state this variation somewhere in the text.– Immunostaining of ciliated cells and others would really help. The authors however have not included immunostainings of cell types in different passages to verify the qRT-PCR results. These immunostainings could be quantified to show cell compositions in the ALOs in 3D and 2D.In all, the authors should use qRT-PCR to determine presence of all cell types, and combine this with immunostainings to determine cell compositions of the models. The authors still show the CIBERSORTx method which confuses the reader since it contradicts data shown in other figures. The authors explain in their rebuttal letter that the method was only used as corroborative, yet the data contradicts their own data.

We understand the reviewer’s concern that the 4 methodologies used here are not 100% in agreement with each other when it comes to the proportion of cell types in the ALO models. In fact, this particular issue is brought up by this reviewer in two more comments later within this document (Comments #3d and 4), criticizing the FACS and CYBERSORTx approaches. We have attempted to answer all of those questions in this response.

To recap for the readers, our intended use of these approaches was always to rigorously test the presence of a mixed population of proximal and distal (AT2/AT1) cells in the organoid lines (a major claim in the manuscript), despite passaging in culture. It was never our intention of using them to draw conclusions regarding the exact proportion of each cell type. In fact, we have not claimed anything regarding the relative proportions of cells anywhere in the manuscript.

The reason is that each of the 4 different approaches used here has its own set of strengths (PROS) and limitations (CONS):

**Author response table 1. sa2table1:** 

Approach	PROS	CONS	Major conclusion	Caution
FACS of dispersed cells from organoids	HighthroughputanalysisAnalyzes protein, nottranscripts	Ab-related artifactsHas the potential to introduce artifactsduring dissociation	Mixed cellularity was confirmed in all 3 ALO lines.Mixed cellularity is retained despite the passage	This methodology, *standalone*, is not appropriate to draw conclusions regarding the absolute proportions of each cell type because of shared markers, and antibody limitations
Targeted qPCR	Highly sensitive andspecific	Low-throughputanalysis that only measures transcript, but does not inform about protein translation	Mixed cellularity was confirmed in all 3 ALO lines.Mixed cellularity is retained despite passage	this methodology, *standalone*, is not sufficient to draw conclusions regarding the proportions of each cell type because of shared transcripts between cell types*.*
RNASeq>deconvolution using CYBERSORTx	highthroughput analysis	Results are as good as our collective knowledge of cell type markers, many of which are shared	Mixed cellularity was confirmed in all 3 ALO lines.Mixed cellularity is retained despite the passage	this methodology, *standalone*, is not sufficient to draw conclusions regarding the proportions of each cell type because of shared transcripts between cell types
In situ detection of protein by immunostaining of3D/2D organoids	Detection of protein (not just transcript) and with fewerartifactsbecause of *in situ* analysis	Low-throughput qualitative analysis	We prioritized this methodology over others and used two different approaches (FFPE samples after embedding in Histogel Figure 2I-J and direct fixation with PFA/methanol, Figure 2K) to reduce the fixation related aritfacts of any one particular methodology.	Although this approach was the best way to show mixed cellularity in each line, and at times, within the same 3D organoid structure, it is low throughput and qualitative (not quantitative), and hence, not suitable to be used for serial imaging of markers to estimate % cellularity/composition.

What is important to note that all methodologies used, regardless of whether they detected mRNA or protein, confirmed that both proximal and distal epithelial cells were present.

No conclusions were drawn at any point in the manuscript regarding the relative proportion of each cell type.

While we agree that it would be nicer to have confocal imaging on all lines from all passages, this is the lowest throughput methodology and doing this is not feasible.

Action taken: We have revised the “Limitations” section of the discussion (Page 17) to explicitly state that the type of evidence presented only supports the claim of mixed proximo-distal cellular composition, but does not reveal the absolute proportion of such cellularity. For the convenience of the reviewer and editor, we have copied and pasted the altered text below:

“While we successfully demonstrated the proximo-distal mixed cellular composition of the ALOs using four different approaches (flow cytometry, RNA seq, confocal immunofluorescence and targeted qPCR), and showed that such mixed cellularity is preserved during prolonged culture, the exact cellular proportion was not assessed here. Single cell sequencing and multiplexed profiling by flow cytometry are some of the approaches that can provide such in-depth characterization to assess cellular composition at baseline and track how such composition changes upon infection and injury.”

And again on Page 16:

“We noted some variability of cell types between patient to patient, and between early and late passages of ALOs, which is probably because of the heterogeneity of organoids isolated from patient’s lung specimens”.

Comment #2Claims should be tampered, in particular related to high throughput (HTP) drug screening since only few drugs were tested.Response to rebuttalThe authors explain in great depth that they did not imply that HTP were performed. This can however be inferred from the text even after changing the text as given in the rebuttal. The finding that the ALOs can be used in a 384-well format is important but the authors should add some more in-depth detailed description that this only generates a possibility for potential HTP screens instead of inferring HTP can be done on these models. The description of a "proof-of-concept for HTP" confuses. This term implies that the authors have used an initial HTP drug screen to validate. This can be changed to "initial protocol for building a platform on which HTP drug screens could be performed"?

We understand the concern. We are happy to share with the reviewer that our method of monolayers preparation for drug screenings is adapted by other groups and the publication is recently released in BioRxiv (PMID: 34159337).

Action taken: To satisfy the reviewer, we have modified the text in the result on Page 11.

“Findings also validate optimized protocols for the adaptation of ALO monolayers in miniaturized 384-well formats for use in high throughput drug screens”.

Comment #3Discrepancies with transcriptomic signatures of the infected human lungs need to be more carefully considered.Response to rebuttalThe authors have included more clinical datasets and comparisons between other published model systems. It remains important to mention the discrepancies between clinical samples and the model. This is altered in the text sufficiently. This comment is therefore satisfactorily answered in the revised version of the manuscript.

We are pleased to see that the reviewer thinks this comment was addressed satisfactorily.

Comment 1The manuscript claims a novel model system of distal and proximal airway epithelium. The authors however fail to discuss a recently published COVID-organoid study which reports similar observations. While their novelty claim could still hold in some areas, the authors do need to discuss the recent manuscripts.Response to rebuttalThe authors have extended their table including all the existing models with the recent published model systems. I agree that keeping up with COVID-19 model systems is hard in these times. The inclusion into the tables has, however, not led to the authors changing their claim of a novel model system of bronchiolar and alveolar potential. In line 102-104 the authors still write "…what is particularly noteworthy is that none recapitulate the heterogeneous epithelial cellularity of both proximal and distal airways…" which should be altered since Lamers et al. did exactly this. Moreover, in the rebuttal letter, the authors also mention that iPSC-derived AT2 cultures can contain both proximal and distal cell types.The comparison with the currently published method of Lamers et al. 2021 is now in the manuscript and could be highlighted in the discussion or at the end of the paragraph named "Creation of a lung organoid model, complete with both proximal and distal airway epithelia" by adding a few lines comparing the cell types present in this model (ciliated and goblet) which was not shown in Lamers et al. 2021 (of at least this is convincingly demonstrated)

The reviewer is right in that the model developed by Lamers et al., has mixed cellularity. We should have defined knowledge gap with more specificity.

Action taken: We have edited the following two sentences and added a new sentence in the introduction on Page 4 to sharpen the scope and context, and accurately present what has been done prior to this work.

“While a head-to-head comparison of the key characteristics of each model can be found in Table 1, what is particularly noteworthy is that most of the models lack the heterogeneous epithelial cellularity of both proximal and distal airways, i.e., airway epithelia, basal cells, secretory club cells and alveolar pneumocytes.”.

“Also, iPSC-derived AT2 cells be differentiated to proximal and distal cell types, including AT1 and ciliated and club cells13-15 but the models derived from iPSCs lack propagability and/or cannot be reproducibly generated for biobanking; nor can they be scaled up in cost-effective ways for use in drug screens”.

“More specifically, adult lung organoid models that can be grown in a sustainable mode and are complete with proximo-distal epithelia are yet to emerge.”

We also added the work from Lamers *et al.* (EMBOJ, 2021) in the result section on Page 7.

“It is noteworthy that the co-existence of proximal and distal epithelial cells in lung organoids has been achieved in one another instance prior; Lamers et al., showed such mixed cellular composition in fetal lung bud tip-derived organoids^45^. However, their model lacked ciliated and goblet cells^45^, something that we could readily detect in our 3D organoids.”

Comment 2.Expression patterns in Figure 2B are hard to interpret when no tissue control is used. A positive control of lung tissue should be used to make valid conclusions.Response to rebuttalThe authors have answered this comment by adding Figure 2S3

We are pleased to see that the reviewer thinks this comment was addressed satisfactorily.

Comment 3.The authors claim the presence of multiple cell types within single ALO. Data are not strong.3a. Co-staining of KRT5 and SFTPB (as shown in figure 2J) is difficult to explain, since these markers are expressed in two different cell types. The lower panel in the same figure does show two separate populations, but the images shown in the second panel do not.and3b. While the authors claim the presence of proximal and distal cells in ALOs, the images show 100% of SFTPC+ cells or 100% KRT5+ cells or 100% Na/K-ATPase cells. This rather shows two different types of ALO within one culture, then mixed cell populations within a single ALO.Response to rebuttalBy changing the images and the text, the authors made clear that the ALOs are either composed of multiple cell types or of a singular cell type. Therefore, this comment is sufficiently answered.

We are pleased to see that the reviewer thinks this comment was addressed satisfactorily.

3c. The authors claim the presence of ciliated cells by staining of acetylated α tubulin in figure 2K-J and figure 3E. The images shown however are hardly showing specific ciliated cell staining and more importantly the cilia as present on the cell membrane.Response to rebuttalBy addition of the higher resolution images, the authors answered this comment sufficiently.

We are pleased to see that the reviewer thinks this comment was addressed satisfactorily.

3d. In figure 3H, the authors show MUC5B positive cells in their monolayer cultures. These goblet cells seem to comprise quite a significant proportion of the culture. Their graph in 3B however shows no goblet cell contribution. The claim that ALO consists of a mix of defined cell types is essentially based on deconvoluted bulk RNAseq data. Such a claim is not conclusive.Response to rebuttalThis comment comes back to earlier described methods in the editor comment section. To shortly summarise, the CYBERSORTx data confuses the reader and is contradictory to the immunostainings, qRT-PCR and flow cytometry data.

We have provided detailed answer to this point in our response to Comment #1.

Comment 4.Authors claim that ALO maintain cell type ratios similar to lung tissue over several passages. Their data in figure 3B however show a drastic change in the composition of the ALOs. The shift from AT2 to AT1 is explained to occur when the ALO are cultured as submerged monolayers. However, the shift is already visible at later passages in 'standard' ALO culture. Moreover, when comparing cell types between passage 1 and 8, the composition changes dramatically. The cultures have lost basal cells, ALO1 has lost goblet cells, ALO2 has lost ciliated cells. In addition, ALO1 composition differs from ALO2 at passage 2 which doesn't support the statement that the cultures are stable/robust.Response to rebuttalSee earlier comments about using CYBERSORTx and flow cytometry data.

We have already addressed it in comment 1 in the Editors list of Essential Revisions and in comment 3 above. To avoid repetition, please see the response to Comment 1 above.

Comment 5.While the authors claim that proximal airway cultures are infected and responsible for maintaining virus replication, there are no virus entry marker-annotated cells in the composition graph of figure 1B.Response to rebuttalThe comment was aimed at the proximal ALI culture cell composition graph in original figure 3B. This graph did not indicate a VEM population. Still this culture was infected by the virus, without showing a VEM population. The authors have removed the viral entry marker cells from their data representation. This was due to bioinformatic difficulty and VEM cells being a separate cell type in their analyses.

This mistake was also picked up by reviewer #3. We thank the reviewer for closely reading the manuscript.

Action taken: We fixed the legend in Figure 3 by removing GLL and VEM.

Comment 6Similar to point 3, the authors claim the development of the first combined organoid culture of proximal and distal cell types, as supported by deconvoluted bulk RNAseq. The hiPSC-derived AT2 cultures however also show the presence of both distal and proximal cell types.Response to rebuttalIn the rebuttal letter the authors mention the manuscripts describing the presence of proximal cell types in a AT2 differentiation protocol of iPSCs. This once more underlines that their statement of the first culture of proximal and distal cell types is incorrect.

This sentence is an unfortunate misunderstanding. We understand that our statement on iPSCderived AT2 cells being able to differentiate also into proximal cell types can be confusing to the readers.

Action taken: Therefore, we have edited the following line to compare the difference between our model with the iPSC-derived models in the introduction on Page 10.

“Also, iPSC-derived AT2 cells be differentiated to proximal and distal cell types, including AT1 and ciliated and club cells^13-15^ but the models derived from iPSCs lack propagability and/or cannot be reproducibly generated for biobanking; nor can they be scaled up in cost-effective ways for use in drug screens”.

Comment 7Figure 3H shows a graph presenting the infectivity relative to peak. This axis makes comparison of infectivity impossible. The authors may want to include suppl Figure 5A-B as main figure and exclude 3H from the manuscript. Moreover, viral E-gene qPCR should be complemented by viral titer measurements to verify findings for live virus. The analysis of SARS-CoV-2 infectivity is very limited when only showing a few infected cells and viral E-gene graphs. Suppl Figure 5C also only shows one or two infected cells in all samples which does convince as proof of infectivity.Response to rebuttalThe authors provide an extensive reasoning behind the use of the legend in figure 3H. The sustained infectivity the authors want to bring across is sufficiently depicted in figure 3H. Figure 3H might confuse the readers that the submerged culture are more heavily infected than ALI cultures. Moreover, this would allow the authors to already claim their permissiveness for infectivity in proximal airways without having to only look at 48-72h timeframe. "Permissive" in general refers to infection at an early stage which is shown in figure 3S3B and not figure 3H.

We are pleased to see that the reviewer agrees–“The sustained infectivity the authors want to bring across is sufficiently depicted in figure 3H.” The reviewer opposes the use of the word “permissive/permissiveness” because he believes that this word suggests early events in viral infection, whereas we are investigating and drawing conclusions about sustained infectivity. We agree.

Action taken: We had used the word “permissive” in 3 places within the text. The first two times it is used in the context of the current model. The third instance is when describing the fetal lung model developed by Lamars et al., EMBO J.

We edited each of those as follows.

First (Page 10): “We first asked if ALO monolayers are permissive to SARS-CoV-2 entry and replication and support sustained viral infection.”

Second (Page 10): “When we specifically analyzed the kinetics of viral E gene expression during the late phase (48-72 hpi window), we found that proximal airway models [human Bronchial airway Epi (HBEpC)] showed high levels of sustained infectivity than distal models [human Small Airway Epi (HSAEpC) and AT2] to viral replication (Figure 3—figure supplement 3C); the ALO monolayers showed intermediate sustained infectivity (albeit with variability).

Third (Page 13): “These findings indicate that despite having mixed cellular composition and the added advantage of being able to support robust viral replication (achieving ~ 5 log-fold increase in titers), the model lacks the signature host response that is seen in all human samples of COVID-19.”

Comment 8The authors provide data for a single drug to indicate the possibility of high-throughput screening of drugs for COVID-19 treatment. This does not say much about the throughput of the assay.Response to rebuttalThis comment has been answered and once more commented in editor comments.

We are pleased to see that the reviewer thinks this comment was addressed satisfactorily.

Comment 9Figure 5C compares uninfected and infected monolayer cultures for genes that are identified in the patient cohort. While the authors claim comparable patterns between the cultures and the patients, the heatmap can not be directly read. The authors should include more details in the legends and combine the heatmaps in 4C and 5C for direct comparison. Currently, the colors represent raw z-scores which do not indicate transcript read numbers but relative differences of the expression of the indicated genes in the samples analysed. These numbers could differ extensively between organoids and patients.Response to rebuttalThe authors have added some extra lines on the method used to generate these heatmaps. It now becomes clear that these datasets can not be directly compared within one heatmap.

We are pleased to see that the reviewer now understands why the previously made request for directly comparing distinct datasets within one heatmap is not appropriate and hence, not done.

Comment 10In addition to point 9, the authors show that there is very limited overlap between significantly differentially expressed genes in monolayers and patients. While the authors believe this is mostly due to the lack of mesenchymal and immune components, this indicates that the monolayer itself is not important for the observed infection signature. This makes the claim that the monolayers represent infectivity in patients questionable.Response to rebuttalThis has been discussed in the editor's section of the revision above

We presented objective analyses of all available models (including ours) and human disease tissue. Our analyses showed that compared to all other models, our model most closely recapitulated the human disease. More specifically, despite missing immune cell and stromal cell components, the epithelial cells alone captured ~ 25-50% of the gene expression changes (Differentially upregulated genes). This is a level of rigor seldom met by any disease modeling work.

In light of this objective evidence, we respectfully disagree with the reviewer’s skepticism that the epithelial monolayers do not represent infectivity in patients.

It was never our intention to question the importance of the other cell types in any disease model (and we explicitly discussed that as a study limitation). However, we believe that understanding the epithelial cell response to infection is critical to understand how the first responders in our innate immune defense system may respond to the infection. Modeling such epithelial response is very important before we go for the complex model after adding mesenchymal and immune cells.

Action taken: We also discussed the findings in the section titled “limitations of the study” on Page 17.

“ Our adult stem-cell-derived lung organoids, complete with all epithelial cell types, can model COVID-19, but remains a simplified/rudimentary version compared to the adult human organ. For instance, although the epithelial contributions to the host response are important, it alone cannot account for the response of the immune cells and the non-immune stromal cells, and their crosstalk with the epithelium. Given that epithelial inflammation and damage is propagated by vicious forward-feedback loops of multicellular crosstalk, it is entirely possible that the epithelial signatures induced in infected ALO-derived monolayers are also only a fraction of the actual epithelial response mounted in vivo. Regardless of the missing components, what appears to be the case is that we already have a model that recapitulates approximately a quarter to half of the genes that are induced across diverse COVID-19 infected patient samples. This model can be further improved by the simultaneous addition of endothelial cells and immune cells to better understand the pathophysiologic basis for DAD, microangiopathy, and even organizing fibrosis with loss of lung capacity that has been observed in many patients40; these insights should be valuable to fight some of the long-term sequelae of COVID-19. ”.

Comment 11.Description in the discussion of growth factors supplied in the medium like FGF7 and FGF10 are not novel. Sachs et al. 2019 (https://doi.org/10.15252/embj.2018100300) already described these growth factors in the culture medium of airway organoids.Response to rebuttalThe authors have sufficiently answered this comment.

We are pleased to see that the reviewer thinks this comment was addressed satisfactorily.

Comment 12.In the discussion, the authors describe the advantages of their model system includingreproducibility, retainment of genetics and use for infection studies. These claims are however not novel since previous papers from multiple groups have reported similar organoid-based models for SARS-CoV-2 research.Response to rebuttalWith the comments above and the answers from the authors, this comment will be answered. The authors should better define their novelty by explaining the origin of the organoids model (adult stem cells) and their improvement in COVID-19 modelling.

We are pleased to see that the reviewer felt that our answers regarding the novelty of the model are satisfactorily answered. We appreciate the suggestion that we should improve that description of novelty.

Action taken: In this revised submission, we have expanded the last paragraph in the discussion on Page 17 (just before the paragraph on study limitations).

“The lung model we present here is distinct from all currently available other models (see Table 1) because of the confirmed presence of both proximal and distal airway cell types over successive passages, which is yet to be accomplished for adult lung organoid models. Another distinguishing feature of our model is the way we rigorously validated its usefulness in modeling COVID-19 via computational approaches. We confirmed, based on the gene expression changes upon SARS-CoV-2-challenge, that our model most closely recapitulates the human disease, i.e., Covid-19. Analyses also pinpointed the importance of two factors that were critical in modeling COVID-19: (1) adult source, and (2) model completeness, with both proximal and distal airway cells.”.

Reviewer #3 (Recommendations for the authors):The authors established a new 3D adult lung organoid system. Comparing with previous systems, the new organoid culture can maintain both proximal and distal cell types in adult lungs, and this ratio appears to be relatively stable in long term cultures. The 3D organoids can subsequently be dissociated and re-plated into 2D submerge culture, which exhibited promising features for modelling COVID19 infection. The authors further used bioinformatics analysis to show that the COVID19 infection using the 2D submerge culture was able to recapitulate both the infectivity and the immune responses in COVID patients.In the revision processes, the authors have provided more supporting data to show the co-existence of proximal and distal lung cell types in the long-term culture and addressed most of the previous reviewers' comments. The few comments that we recommend the authors to further address are as follows:

We are pleased to see that this reviewer found our revisions strengthened the major claim in this manuscript, i.e., co-existence of proximal and distal cell types in long term culture.

The reviewer also noted that although most of the previous reviewers’ comments were answered, he/she recommends that we address 4 major and 3 minor points. We have done that below. We hope that the changes we made address the remaining concerns adequately.

1) The author added the Act-TUB staining for 2D culture in Figure 3, which looks convincing, however, for the 3D organoids, Act-TUB staining in Figure 2J still doesn't look like any cilia structure.

In this comment, the reviewer is comparing Acetylated Tubulin staining patterns in Figure 2, which shows 3D organoids and Figure 3, which shows 2D organoids (both submerged monolayers or in the ALI model). Our claim was that Ac Tub positive ciliary structures are seen in 2D organoids, most prominently in the more differentiated ALI model. The cilia in ALI has been shown in a magnified view in Figure 3 Figure supplement 1J. We do not see such prominent ciliary structures in 3D organoids; it is possible that these 3D organoids do not show the ciliated structures simply because they are not differentiated sufficiently (e.g., no lumen, and hence, we suspect that apicobasal polarity is yet to be established, which is essential for apical cilia formation). Others have demonstrated cilia formation in ALI models of 3D growth (PMID: 27713058), where the size of the organoid structure is substantially larger and lumen formation is associated with apicobasal polarity and cilia formation.

Action taken: To ensure that our findings, figures, and description in text avoid misleading readers, we have added the following sentences on Page 9 to address this issue with greater clarity:

“Compared to the 3D organoids, the 2D organoid cultures, especially the ALI model, showed a significant increase in ciliated structures, as determined by Acetylated Tubulin (compare Ac Tub stained panels in Figure 2J-K with Figure 3E-F)”. The observed progressive prominence of ciliary structures from 3D to 2D models is in keeping with the fact that 3D ALOs that are yet to form lumen represent the least differentiated state, whereas 2D submerged monolayers are intermediate and the 2D-ALI monolayers are maximally differentiated; differentiation is known to establish apicobasal polarity, which is essential for the emergence of mature cilia on the apical surface”.

2) In Page 6 text line 189, the author mentioned a 'stem cell' population, which is TP63 positive. This is separate to the basal cells on line 188. Whereas in Figure 2-Supp4 the authors have acknowledged TP63 as a basal cell marker. We recommend the authors to make this consistent as basal cells which are well-established to be the airway stem cells (Rock et al., 2009).

We agree with the suggestion. We also believe that TP63 should be represented as a basal cell marker and hence, in the original version of the manuscript, we had indicated the same. However, upon suggestion of this reviewer, we analyzed another different marker (p75/NGFR) and had added that finding. We are glad to see that the reviewer now agrees that TP63 is a *bona fide* marker of airway stem cells.

Action taken: In this revised submission, we have edited it throughout the manuscript to indicate that TP63 is a marker of basal cells, which are well established to be airway stem cells.

3) The authors still make a strong claim about the co-existence of proximal (airway) and distal (alveolar) cell populations in a single 3D organoid (line 198). However, the staining shown in Figure 2 can only infer the existence of either proximal or distal cell populations in a single 3D organoid, given SFTPB is not a specific marker for alveolar lineage. Additionally, this strong claim wasn't adding much value to the manuscript as the authors only need to show proximal-distal cells exist in the overall population as 3D culture, given only the 2D submerged culture was used for COVID infection. We recommend the author not to mention this claim as even the revised data do not support it.In addition, the same paragraph describes BASCs in the organoids. We cannot cite a reference to refute the existence of BASC cells in human lungs. However, in now more than 10 years of people looking for them a convincing demonstration that BASCs exist in human lungs is still missing. Human distal airways have a very different organisation to those of mice (respiratory bronchioles). BASC cells have not been found in them. We recommend that you do not highlight human BASCs in the organoids given the lack of credible evidence that they exist in vivo.

This comment has two parts and we have broken down the parts below and responded to each separately:

i) First, the reviewer says that while it is convincing that the same ALO line has a mixed population of proximal and distal airway epithelial cells, such a mixed population is not there in the same 3D organoid structure does not. The reviewer acknowledges that because infections are all carried out in 2D monolayers of a mixed cell population, the claim that a single 3D structure has mixed cell types is unnecessary for the major point that we are trying to make.

We agree that the presence of mixed cellularity in a single 3D organoid structure is not essential for supporting the major claims in this manuscript which use 2D mixed cellular model for SARS-CoV-2 infection and COVID-19 modeling.

However, we respectfully disagree with the reviewer that single organoid structures do not have a mixture of cells. We believe that this is an important point in the characterization of the ALO model because the fact that ALOs can achieve both homotypic and heterotypic cellular composition may influence how these models may behave in 3D-ALI or whether they are appropriate for use in 3D models of infection with microinjections.

Action taken:

To ensure we state the findings, but not overstate it, we modified the sentence to strengthen the claim that this reviewer acknowledges, i.e., that mixed proximal and distal cellularity was there in each ALO line, and soften the claim that such cellularity was detected also in the same 3D structure.

“The presence of all cell types was also confirmed by assessing protein expression of various cell types within organoids grown in 3D cultures. Two different approaches were used—(i) slices cut from FFPE cell blocks of HistoGel-embedded ALO lines (Figure 2I-J) or (ii) ALO lines grown in 8-well chamber slides were fixed in matrigel (Figure 2K), stained, and assessed by confocal microscopy. Such staining not only confirmed the presence of more than one cell type (i.e., mixed cellularity) of proximal (basal-KRT5) and distal (AT1/AT2 markers) within the same ALO line, but also, in some instances, demonstrated the presence of mixed cellularity within the same 3D structure. For example, AT2 and basal cells, marked by SFTPB and KRT5, respectively, were found in the same 3D-structure (Figure 2J, interrupted curved lines). Similarly, ciliated cells and goblet cells stained by Ac-Tub and Muc5AC, respectively, were found to coexist within the same structure (Figure 2J, interrupted box; Figure 2K, arrow). Intriguingly, we also detected 3D structures that co-stained for CC10 and SFTPC (Figure 2J, bottom panel) indicative of mixed populations of club and AT2 cells. Besides the organoids with heterogeneous makeup, each ALO line also showed homotypic organoid structures that were relatively enriched in one cell type (Figure 2J, arrowheads pointing to two adjacent structures that are either KRT5- or SFTPB-positive). Regardless of their homotypic or heterotypic cellular organization into 3D-structures, the presence of mixed cellularity was documented in all three ALO lines (see multiple additional examples in Figure 2—figure supplement 2I)”.

ii) Second, the reviewer asks us to remove the discussion and citation surrounding BASCs, which have not been found in the human lung.

Action taken: We completely agree with the reviewer and we removed the line on Page 7 with the associated reference on BASC.

We have also edited the following line on Page 7**.**

“Intriguingly, we also detected 3D structures that co-stained for CC10 and SFTPC (Figure 2J, bottom panel) indicative of mixed populations of club and AT2 cells”.

4) The authors need to clarify how the 5 replicates for submerged culture and 2 replicates for ALI culture were done in Figure 2B middle panel. Were they from the same ALO line or from different ALO lines? What was the passage number used here? These will help the readers to have an idea about how reproducible the system is.

We believe that the reviewer meant to say Figure 3B instead of 2B because in Figure 2B there are no submerged or ALI cultures. Instead, the description of panels provided here matches the Figure panel 3B where % of the cellular composition is predicted using the CIBERSORTx.

For the submerged monolayers, ALO1 and ALO2 were used where we have the following 5 samples (i and ii) from ALO1 passage 3, (iii and iv) from ALO2 passage 3 and (v) from ALO1 passage 8. The ALI model is generated from ALO1 passage 3 and passage 8.

Action taken: In this revised submission, we have included these details also in Figure 3 panel B.